Report

# Developmental mitochondrial complex I activity determines lifespan

Rhoda Stefanatos [1,2,3,7✉], Fiona Robertson [3], Beatriz Castejon-Vega[3], Yizhou Yu [4], Alejandro Huerta Uribe[5], Kevin Myers[3], Tetsushi Kataura[6], Viktor I Korolchuk[1], Oliver D K Maddocks[5], L Miguel Martins [4] & Alberto Sanz [3,7✉]

## Abstract

**Aberrant mitochondrial function has been associated with an increasingly large number of human disease states. Observations from in vivo models where mitochondrial function is altered suggest that maladaptations to mitochondrial dysfunction may underpin disease pathology. We hypothesized that the severity of this maladaptation could be shaped by the plasticity of the system when mitochondrial dysfunction manifests. To investigate this, we have used inducible fly models of mitochondrial complex I (CI) dysfunction to reduce mitochondrial function at two stages of the fly lifecycle, from early development and adult eclosion. Here, we show that in early life (developmental) mitochondrial dysfunction results in severe reductions in survival and stress resistance in adulthood, while flies where mitochondrial function is perturbed from adulthood, are long-lived and stress resistant despite having up to a 75% reduction in CI activity. After excluding developmental defects as a cause, we went on to molecularly characterize these two populations of mitochondrially compromised flies, short- and long-lived. We find that our short-lived flies have unique transcriptomic, proteomic and metabolomic responses, which overlap significantly in discrete models of CI dysfunction. Our data demonstrate that early mitochondrial dysfunction via CI depletion elicits a maladaptive response, which severely reduces survival, while CI depletion from adulthood is insufficient to reduce survival and stress resistance.**

**Keywords** Ageing; Complex I; Drosophila; Mitochondria; Mitochondrial Disease
**Subject Categories** Development; Metabolism

## Introduction

Mitochondria, the organelles that make multicellular life possible, regulate major cellular processes such as division, differentiation, and death (Nunnari and Suomalainen, 2012). The mitochondrion serves as the site of production for energy, signaling molecules, and metabolites. At the core of this is oxidative phosphorylation, OXPHOS. OXPHOS requires a functional electron transport chain (ETC) made up of 4 multi-protein complexes (CI-CIV) to create a proton gradient that powers ATP synthase (CV). Complex I (CI) is the largest of the four ETC complexes. It not only acts as an electron entry point but also as a major source of NAD + and mitochondrial ROS (mtROS). Deregulation and mutation of genes encoding CI subunits have been associated with various disease states, including diabetes, mitochondrial disease, cancer, and neurodegeneration (Pagliarini et al, 2008). It is, therefore only right, that CI is considered a highly relevant therapeutic target that regulates not only cellular but also organismal fate (Stefanatos and Sanz, 2011).

The assembly, structure, and function of CI have been intensively studied in several model systems (Granat et al, 2023; Grba et al, 2024; McElroy et al, 2020; Pagliarini et al, 2008). From these studies, we have seen varied effects of deletion or depletion of CI on fitness, with the mechanisms underlying the consequences of dysregulated mitochondrial metabolism still unclear. In ageing, CI depletion has been shown to both reduce and extend lifespan in various model systems (Copeland et al, 2009; Dillin et al, 2002; Foriel et al, 2018; Grad and Lemire, 2004; Kruse et al, 2008; Owusu-Ansah et al, 2013; Sanz et al, 2010). Currently, we do not fully understand how alterations in mitochondrial function can have such paradoxical effects.

In the case of mitochondrial disease, primary mitochondrial CI deficiency can result in very severe disease that presents early and is often fatal, but it can also present mildly and later in life (Gorman et al, 2016; Gorman et al, 2015; Russell et al, 2020). The prevailing dogma proposes that mitochondrial dysfunction eventually results in an energy crisis that cannot be overcome. However, published works in distinct animal models have demonstrated that the effects of mitochondrial/CI dysfunction can be prevented or attenuated by interventions that do not directly target or rescue mitochondrial deficiency and, therefore, energy production (Cerutti et al, 2014; Civiletto et al, 2018; Ferrari et al, 2017; Jain et al, 2019; Johnson et al, 2013). Accordingly, during the course of our research, we have gathered substantial evidence indicating that the severity of pathology induced by mitochondrial or CI dysfunction depends on factors

[1]Biosciences Institute, Faculty of Medical Sciences, Newcastle University, Campus for Ageing and Vitality, NE4 5PL Newcastle upon Tyne, UK. [2]Wellcome Centre for Mitochondrial Research, Faculty of Medical Sciences, Newcastle University, NE2 4HH Newcastle upon Tyne, UK. [3]School of Molecular Biosciences, College of Medical, Veterinary and Life Sciences, University of Glasgow, G12 8QQ Glasgow, UK. [4]MRC Toxicology Unit, University of Cambridge, CB2 1QR Cambridge, UK. [5]School of Cancer Sciences, Wolfson Wohl Cancer Research Centre, University of Glasgow, G61 1QH Glasgow, UK. [6]Department of Neurology, Institute of Medicine, University of Tsukuba, 305-8575 Ibaraki, Japan. [7]These authors contributed equally as senior authors: Rhoda Stefanatos, Alberto Sanz. ✉E-mail: Rhoda.Stefanatos@glasgow.ac.uk; Alberto.SanzMontero@glasgow.ac.uk

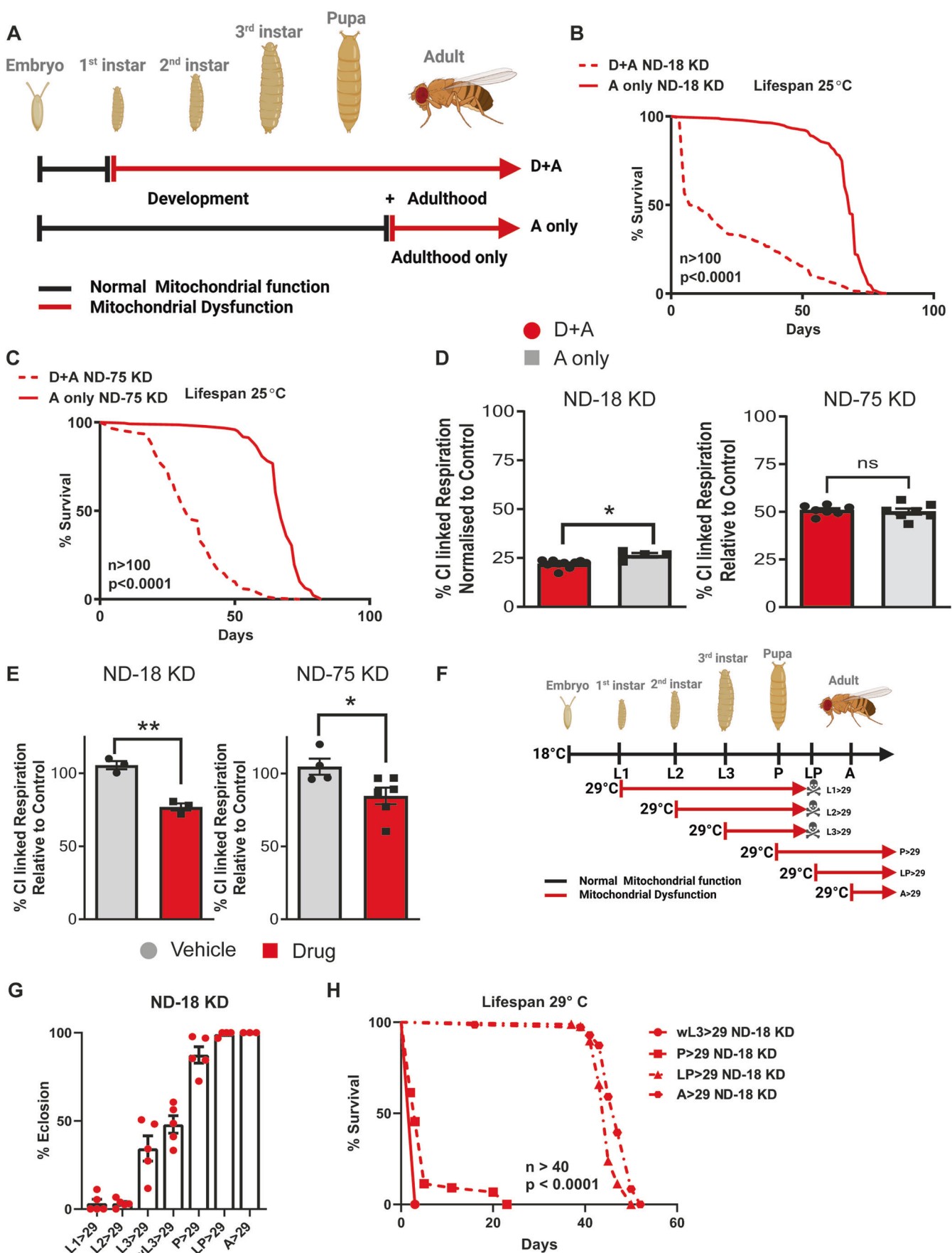

◀　**Figure 1.　Depletion of respiratory complex I from development but not from adulthood shortens lifespan.**

(A) A schematic illustrating the periods during which the GS inducible expression system was used to reduce CI function, from early development into adulthood (D + A) or from adulthood only (A-only). (B) Survival of male flies where CI subunit, ND-18, has been depleted from early development (D + A) or from adulthood (A-only). Log-rank test, $P < 0.0001$, $n = 262$–266 flies per experimental group. (C) Survival of male flies where CI subunit, ND-75, has been depleted from early development (D + A) or from adulthood (A-only). Log-rank test, $P < 0.0001$, $n = 151$–220 flies per experimental group. (D) Normalized levels of CI-linked respiration in ND-18 KD (left) and ND-75 KD (right) 5–7 day adult males. Kolmogorov–Smirnov test, $P = 0.0460$ (ND-18 KD); $T$ test, $P = 0.7131$ (ND-75 KD), $n = 4$–10 independent samples per group. (E) Normalized levels of CI-linked respiration in ND-18 KD (left) or ND-75 KD (right) male 3rd instar larvae in the presence (drug) or absence (vehicle) of the GS inducer, RU-486. $T$ test, $P = 0.0016$ (ND-18 KD) & $P = 0.0429$ (ND-75 KD), $n = 3$–6 independent samples per group. (F) A schematic illustrating the developmental stages where the UAS/GAL4 expression system, coupled with the temperature-sensitive transcriptional repressor GAL80ts, was employed to reduce CI function. (G) % eclosion of flies where depletion of CI subunit, ND-18, has been induced at distinct developmental stages. $n = 3$–5 independent vias per group. (H) Survival of male flies where depletion of CI subunit, ND-18, has been induced at distinct developmental stages. Log-rank test, $P < 0.0001$, $n = 44$–71 flies per group. Please note that in the wL3 > 29 ND-18-KD group, only four flies survived after eclosion (flies got stuck in the fly food); therefore, this group was excluded from the statistical analysis. In bar graphs, means are presented with ± SEM. Significance is indicated as follows: *$P < 0.05$, **$P < 0.01$, ***$P < 0.001$; ns not significant. Source data are available online for this figure.

beyond energy generation or the overproduction of reactive oxygen species (Scialo et al, 2016a). We hypothesized that one of these "factors", which determines the severity of the pathology, is the mitochondrial plasticity of the organism and, therefore, the developmental stage when mitochondrial dysfunction manifests.

The *Drosophila* lifecycle can be crudely divided into development and adulthood. During development, organisms are more adaptable and, therefore, more vulnerable to changes in their internal and external environments. Consequently, any maladaptations that occur can persist into adulthood (Bazopoulou et al, 2019; Elkahlah et al, 2020; Gong et al, 2015; Gyllenhammer et al, 2020; Lafuente and Beldade, 2019). To investigate whether inducing mitochondrial dysfunction at two distinct stages of the lifecycle would alter its effects, we created inducible fly models of CI dysfunction. We decreased CI function either from early development or from eclosion (adults). Flies where mitochondrial dysfunction was induced developmentally exhibited decreased lifespan and lower stress resistance, whereas flies, where mitochondrial function was perturbed from adulthood behaved similarly to control flies. The observed differences in lifespan and stress resistance cannot be attributed to differences in CI activity in adulthood, as both models were indistinguishable in this regard. Using a multiomics approach, we characterized these two populations of flies with compromised mitochondrial function but disparate adult survival. We observed that short-lived mitochondrially compromised flies mount a maladaptive biological response, characterized by reduced functionality of critical catabolic and anabolic pathways, which primarily manifest in the adult fat body but also affect other tissues. The data presented here demonstrate that the inherent plasticity of the organism when mitochondrial metabolism is challenged determines the grade of maladaptation in adults. This maladaptation manifests as several alterations in the capacity to confront stress and shortens the adult lifespan.

## Results and discussion

### Ubiquitous depletion of complex I during development, but not in adulthood, shortens lifespan

To understand whether developmental plasticity could determine the severity of pathology associated with mitochondrial dysfunction, we employed novel inducible fly models of mitochondrial dysfunction.

Using a combination of the GS expression system (Roman et al, 2001) and RNAi interference against subunits of CI, we induced whole-body mitochondrial dysfunction from either development (D + A) or adult eclosion (A-only). This allowed us to directly test the effects of mitochondrial dysfunction when plasticity is high, i.e., development and when it is low, i.e., adulthood. Experimental flies were derived from crossing a ubiquitous driver (*tubulinGS*) with either a control line (UAS-Empty/Control) or RNAi lines targeting two different subunits of CI, ND-18 (ND-18 KD) and ND-75 (ND-75 KD). ND-18, the orthologue of *Ndufs4*, is an accessory subunit of CI required for assembly and stability (Garcia et al, 2017). ND-75, the orthologue of *Ndufs1*, is an essential subunit required for assembly, stability, catalytic activity, and supercomplex formation (Garcia et al, 2017). Both subunits have been found to be altered in patients suffering from mitochondrial disease (Benit et al, 2001; van den Heuvel et al, 1998).

To induce knockdown from development (D + A), flies were crossed on fly media containing appropriate concentrations (drug–food) of the inducer (RU-486) and upon eclosion maintained in vials containing drug–food (Fig. 1A). To limit knockdown to the adult phase (A-only), flies were crossed on normal fly media and transferred to drug–food after eclosion (Fig. 1A). We confirmed the depletion of ND-18 and ND-75 at the mRNA and CI at the protein level in adult male flies (Fig. EV1A,C; Appendix Fig. S2A) and late-stage male larvae (L3) (Fig. EV1B; Appendix Fig. S2B). We saw no difference in the levels of depletion between D + A and A-only ND-18 KD, ND-75 KD, or control flies (Fig. EV1A–D). The D + A group in both ND-18 and ND-75 KD flies were severely short-lived in comparison to A-only KD flies, as well as D + A and A-only control flies (Figs. 1B,C and EV1E; Appendix Fig. S1A,B). These stark differences in survival were also observed in female D + A vs A-only ND-18 KD and ND-75 KD flies (Fig. EV1F,G). This is particularly interesting given conflicting published reports where manipulation of CI has both reduced and extended lifespan (Copeland et al, 2009; Foriel et al, 2018; Scialo et al, 2016a). Our data reconcile the previously contradictory findings that reported both lifespan shortening and extension when CI is depleted. We demonstrate that a fully functional CI is crucial during development but not necessarily in adulthood. This provides a more specific context to the differing outcomes noted in prior studies. Accordingly, mild (this study) to strong knockdown of CI subunits during development significantly reduces adult survival and, in certain cases, can cause notable structural alterations in tissues (muscle (Cho et al, 2012) or brain (Hegde et al, 2014)) and functions

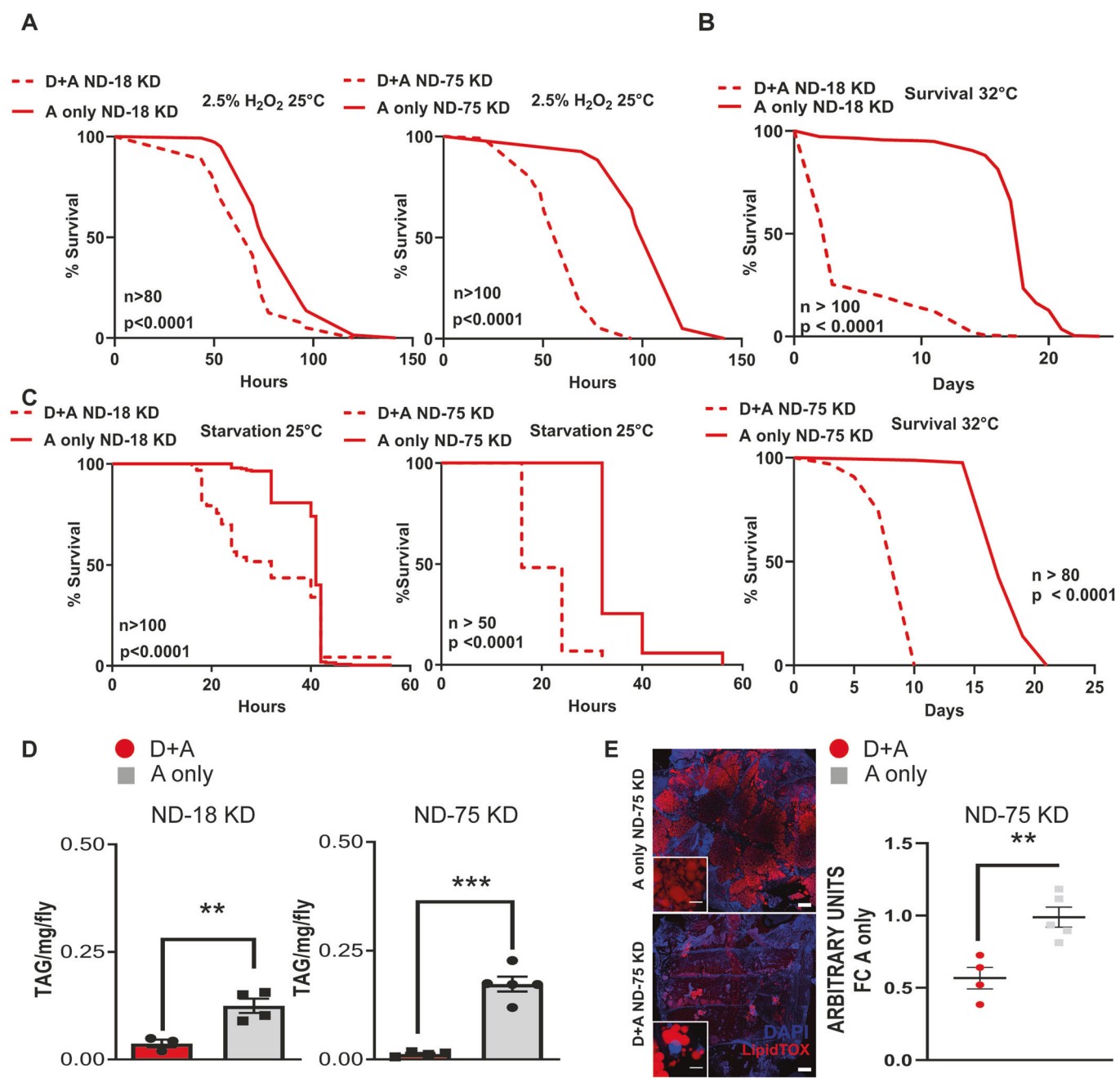

**Figure 2. Developmental but not adulthood CI depletion increases stress sensitivity and alters energy homeostasis.**

(A) Survival under oxidative stress conditions of male flies where either ND-18 (left) or ND-75 (right) has been depleted from early development (D + A) or from adulthood (A-only). Log-rank test, $P < 0.0001$, $n = 80$–140 flies per experimental group. (B) Survival under thermal stress conditions of male flies where either ND-18 (upper) or ND-75 (lower) has been depleted from early development (D + A) or from adulthood (A-only). Log-rank test, $P < 0.0001$, $n = 85$–304 flies per experimental group. (C) Survival under starvation conditions of male flies where either ND-18 (left) or ND-75 (right) has been depleted from early development (D + A) or from adulthood (A-only). Log-rank test, $P < 0.0001$, $n = 51$–254 flies per experimental group. (D) Quantification of triacylglyceride levels in male flies where either ND-18 (left) or ND-75 (right) has been depleted from early development (D + A) or from adulthood (A-only). T test, $P = 0.0090$ (ND-18 KD) & $P = 0.0006$ (ND-75 KD), $n = 3$–5 samples per experimental group. (E) Confocal imaging of fat bodies from male flies where ND-75 left have been depleted from early development (D + A) or from adulthood (A-only) stained with LipidTOX Red and DAPI, scale bar (90 μM), inset scale bar (10 μM). Quantification is displayed on the right. T test, $P = 0.0045$, $n = 4$–5 independent samples per experimental group. In bar graphs, means are presented with ± SEM. Significance is indicated as follows: *$P < 0.05$, **$P < 0.01$, ***$P < 0.001$. Source data are available online for this figure.

(feeding (Foriel et al, 2018)). However, adult depletion of CI does not manifest these complications, as evidenced in this study.

To rule out the possibility that the stark differences in survival observed between flies with CI dysfunction induced from development (D + A) and adulthood (A-only) were due to CI activity variations, we measured CI-linked respiration levels. There were minimal or no significant differences in normalized CI respiratory capacity between D + A and A-only KD flies in both ND-18 KD and ND-75 KD (Fig. 1D, data normalized to Fig. EV1H, see also Appendix Fig. S1C). When we assayed respiration in late-stage larvae (L3), we observed, as expected, decreased CI-linked respiration in ND-18 KD and ND-75 KD larvae on drug–food but not those on vehicle food (Fig. 1E; Appendix Fig. S1D). Notably, the level of remaining CI activity (~25% ND-18 KD, ~50% ND-75 KD) only correlated with lifespan in flies where CI dysfunction had been induced from development. While in flies where CI dysfunction was induced during adulthood (A-only), no difference in lifespan was observed between flies with ND-18 KD, ND-75 KD, or control flies despite this large reduction in mitochondrial function, suggesting that a decrease in CI function alone is not sufficient to reduce lifespan. These data indicate that the reported loss in CI levels and activity in aged flies may not contribute in itself to ageing (Cabre et al, 2017; Navarro and Boveris, 2004; Scialo et al, 2016a). Since similar changes are observed in rodents and humans (Cabre et al, 2017; Navarro and Boveris, 2004), the age-related reductions in CI levels and activity may represent a protective adaptation identical to the proposed mechanism for the low levels of CI observed in oocytes (Rodriguez-Nuevo et al, 2022).

We had established that development was the window during which maladaptation to mitochondrial dysfunction correlates with reduced fitness. To further narrow this down, we replaced the GS system with Gal4/Gal80[ts] (Lee and Luo, 1999). Using ubiquitously expressed daughterless Gal4 in combination with tubulinGal80[ts], we used two culture temperatures, 18 °C (Off) and 29 °C (On), to control the expression of the RNAis and, therefore, induction of CI dysfunction more precisely (Fig. 1F). Developing cultures were moved from 18 °C to 29 °C at different stages of development (Fig. 1F). Eclosion data from these cultures clearly shows that lethality of CI depletion correlates with developmental timing (Figs. 1G and EV1I; Appendix Fig. S1E), where the earlier in development that ND-18 KD or ND-75 KD flies are shifted to 29 °C, the greater the level of lethality. This also correlated with survival; the lifespan of flies that survived eclosion was severely reduced proportionally, and only in those flies where CI dysfunction was induced during development (Figs. 1H and EV1J; Appendix Fig. S1F). When cultures were moved to 29 °C at the late pupal (LP) or eclosed adult stage, no effect on lifespan was observed. The lifespan of control flies subjected to the same treatments was unaffected, allowing us to discount culture temperature (Fig. EV1K,L).

## Developmental CI depletion increases stress sensitivity and alters metabolic homeostasis

Given the reduced survival of D + A CI KD flies compared with A-only CI KD flies and control flies of both groups, we wanted to ask if D + A CI KD flies were short-lived due to inherent stress sensitivity. We looked at three distinct types of stress: oxidative,

heat, and metabolic stress. All these stresses should be affected by differences in mitochondrial function as they require mitochondria to adapt adequately (Graham et al, 2022; Jorgensen et al, 2021; Lewis et al, 2021) i.e., we would expect to see reduced survival in KD flies of both groups (D + A and A-only) compared to control flies as they both have a 50–75% reduction in CI function. D + A and A-only control flies showed no difference in stress resistance in any of the conditions, suggesting that treatment with the inducer itself did not sensitize the flies to stress (Fig. EV2A–C, see also Appendix Fig. S3A–C).

Induction of oxidative stress via $H_2O_2$ treatment demonstrated that A-only KD flies (ND-18 and ND-75) were as resilient to $H_2O_2$ compared to control D + A and A-only flies (Figs. 2A and EV2A). However, D + A CI KD flies (ND-18 and ND-75) were significantly more sensitive, with ND-75 KD D + A flies being less resistant than ND-18 D + A flies (Fig. 2A, compare dashed lines with solid lines). Heat stress at 32 °C resulted in a reduced lifespan in groups compared to the standard culture temperature of 25 °C. However, only flies with CI dysfunction from development (D + A KD flies ND-18 and ND-75) were short-lived compared to their A-only counterparts (Fig. 2B, compare dashed with solid lines). Finally, we subjected all groups to starvation at 25 °C. As with oxidative and heat stress, starvation resistance in A-only KD flies was in accordance with what we observed for control D + A and A-only flies, while D + A ND-18 and ND-75 KD flies were significantly more sensitive (Fig. 2C, compare dashed with solid lines). In summary, only flies with disrupted mitochondrial function during development exhibited increased sensitivity to stress, decoupling levels of CI during adulthood from stress resistance.

Starvation sensitivity in flies correlates with alterations in the adult fat body (Lei et al, 2023); therefore, we opted to conduct a comprehensive characterization of this tissue. The fat body is the major lipid storage tissue, plays a crucial role in the cellular immune response in *Drosophila* and is analogous to the mammalian adipose tissue and liver (Baker and Thummel, 2007). We started by measuring levels of whole-fly triglycerides (TAG) in all groups (Figs. 2D and EV2D; Appendix Fig. S3D). D + A and A-only control flies had similar levels of TAG, while both ND-18 KD and ND-75 KD D + A flies had significantly less TAG than the corresponding A-only flies, confirming a specific decrease in flies where CI has been depleted during development. TAG levels in D + A ND-75 KD flies were almost undetectable despite higher levels of CI function than in D + A ND-18 KD flies. Dissection and staining of adult abdominal fat bodies with a neutral lipid stain, LipidTOX, further corroborated the TAG data (Figs. 2E and EV2E,F; Appendix Fig. S3E). We observed decreased levels of stored lipids in D + A ND-75 KD flies compared with A-only ND-75 KD flies (Fig. 2E). ND-75 KD D + A flies had almost no discernible adult fat body but remnants of the larval fat body attached to the trachea, which would normally be fully degraded within days of adult eclosion, implying a delay in the metabolic transition to adulthood (Fig. 2E). This reduction in stored lipids in D + A CI-depleted flies could underly reduced starvation resistance in comparison to A-only CI-depleted flies (Fig. 2C).

Given the levels of CI dysfunction we were inducing, it was important to understand the effect on the NAD + :NADH ratio. The NAD + :NADH ratio has been implicated in the pathology of several diseases with mitochondrial dysfunction (Liu et al, 2021; Santidrian et al, 2013; Titov et al, 2016). We detected no effect of

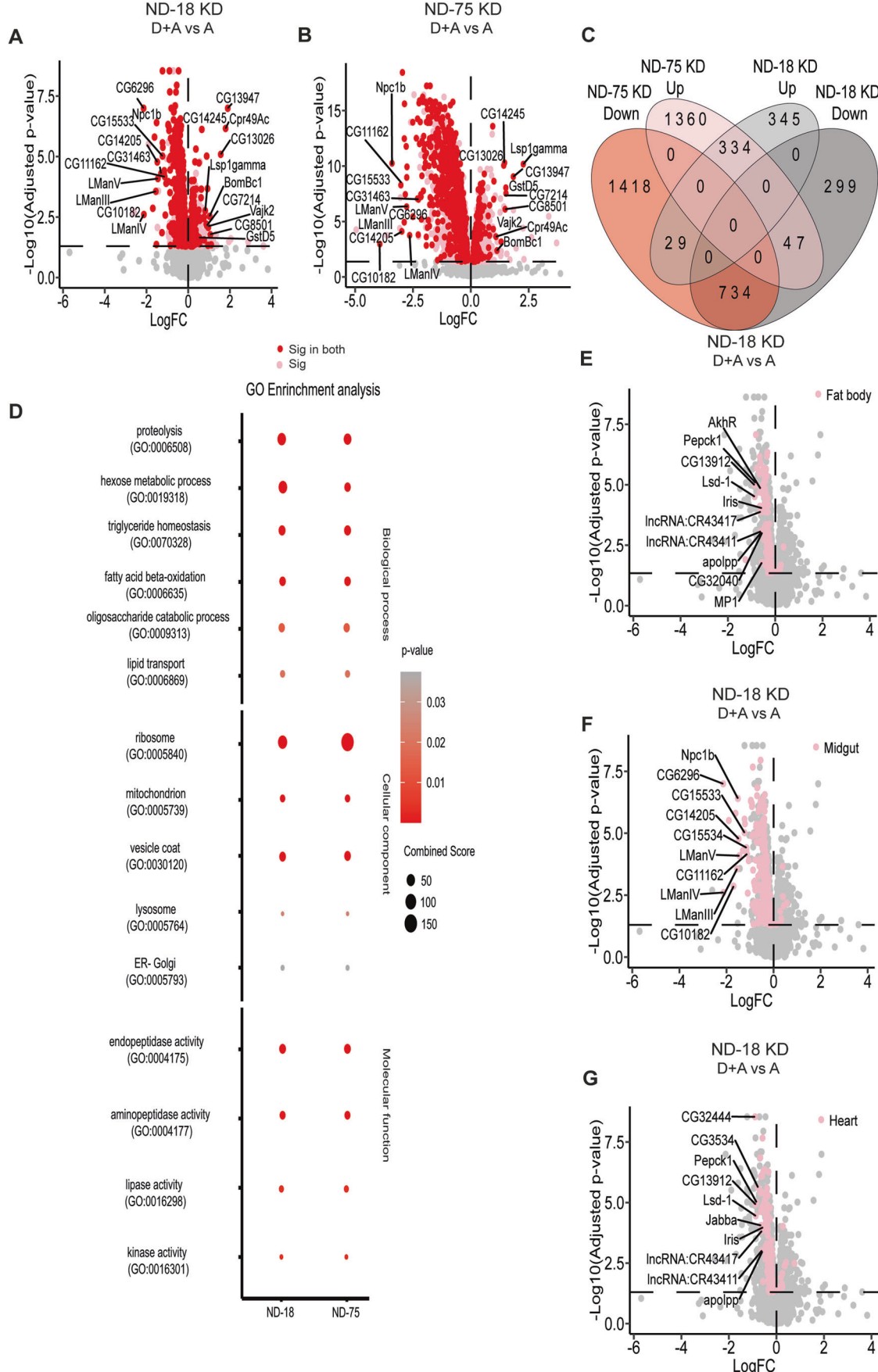

◄ **Figure 3.  Developmental CI depletion results in a specific transcriptomic signature.**

(A, B) Volcano plot of genes differentially expressed between flies where ND-18 (A) or ND-75 (B) was depleted from early development (D + A) or from adulthood only (A). Each dot represents a single gene. In pink are those genes that are significantly altered; in red are those which are significantly altered in both ND-18 and ND-75 D + A KD flies. In gray are genes which were not significantly altered. FDR, P < 0.05, n = 4 independent samples per experimental group. (C) Venn diagram illustrating the relationship between the differential expression seen in ND-18 D + A vs A-only and ND-75 D + A vs A-only, n = 4 independent samples per experimental group. (D) GO Enrichment Analysis performed on those genes significantly altered in both ND-18 D + A vs A and ND-75 D + A vs A reveals significant enrichment in the following classifications. FDR, P < 0.05, n = 4 independent samples per experimental group. (E–G) Volcano plot of genes significantly differentially expressed between ND-18 D + A and ND-18 A-only (gray), highlighted in pink are those genes whose expression is known to be highly enriched in the fat body (E), midgut (F) and heart (G). FDR, P < 0.05, n = 4 independent samples per experimental group. Source data are available online for this figure.

drug treatment on levels of NAD + , NADH or the NAD + :NADH ratio in control D + A or A-only flies (Fig. EV2G). D + A ND-75 KD flies had a significant decrease in NAD+ and NADH in comparison to A-only ND-75 (Fig. EV2H). However, differences between D + A and A-only were not mirrored in ND-18 KD flies (Fig. EV2I; Appendix Fig. S4A,B), indicating that the NAD + : NADH ratio does not contribute to the differential longevity of these flies.

These data demonstrate that CI dysfunction alone is not sufficient to reduce lifespan and increase stress sensitivity, as flies with CI dysfunction induced from adulthood are indistinguishable from control flies. We hypothesized that the induction of CI dysfunction during development, when adaptive plasticity is at its highest, causes maladaptation, which results in reduced survival and stress resistance.

## Transcriptomic analysis indicates tissue-specific alterations affecting the fat body, heart, and midgut in CI-depleted short-lived flies

We established that developmental mitochondrial dysfunction resulted in decreased lifespan and increased stress sensitivity (Figs. 1 and 2), while inducing the same level of CI depletion in adult flies did not elicit any of these phenotypes. We reasoned that the response to CI dysfunction during development must underlie differences in the physiological effects of CI dysfunction. We hypothesized that the inherent plasticity of development, reduced or absent in adulthood, could allow for maladaptive responses to CI dysfunction.

To test whether CI dysfunction in development vs adulthood was provoking a differential, adaptive response, we first assessed the transcriptional response using RNAseq analysis of whole adult flies from all groups. We opted to compare short-lived (D + A) and long-lived (A) flies with mitochondrial dysfunction under the premise that transcriptomic differences between these two groups could potentially explain the observed differences in lifespan. Conversely, if specific genes or pathways show no difference between D + A and A, it is extremely unlikely that they are responsible for the striking differences in adult survival. Principal component analysis implied the expression profiles of D + A vs A-only KD flies were distinct, while for control flies there was significantly more overlap (Fig. EV3A,B). Comparing gene expression in control D + A vs A-only flies identified only 12 significantly altered genes (Fig. EV3C). Treatment with various concentrations of RU-486 has been reported to alter gene expression in several fly models (Andjelkovic et al, 2016; Sommer et al, 2020). To remove any effect on gene expression that might result from the RU-486

treatment regime used to induce CI dysfunction, we filtered out those genes differentially regulated in control D + A vs A-only from our KD expression data. CI dysfunction induced from development vs adulthood (D + A vs A) with KD of either ND-18 or ND-75 resulted in the significant differential expression of 1800 and 3934 genes, respectively (Fig. 3A,B). ND-75 KD flies showed a greater response than ND-18 KD flies. Although some genes were upregulated (708 in ND-18 KD and 1742 in ND-75 KD), more genes were downregulated (1092 in ND-18 KD and 2192 in ND-75 KD). Importantly, comparative analysis of gene expression changes between ND-18 KD D + A vs A-only and ND-75 D + A vs A-only, demonstrated a high level of concordance (Fig. 3A–C, significant genes in pink, significant genes in both comparisons in red). Establishing that the response to CI dysfunction in both models overlaps highly.

GeneOntology (GO) Enrichment Analysis of the genes significantly differentially expressed in both ND-18 and ND-75 D + A KD flies revealed a transcriptional signature characterized by dysregulated lipid homeostasis, transport and metabolism (GO:0070328, GO:0006869, GO:0006635, GO:0016298), reduced protein and sugar metabolism (GO:0006508, GO:0019318, GO:0009313, GO:0004175, GO:0004177, GO:0005793, GO:0005764, GO:0030120) and translation (GO:0005840) (Fig. 3D). We conclude that the distinct transcriptomic signature observed—characterized by the dysregulation of energy metabolism and translation—results from maladaptive processes occurring during development in flies with CI deficiency rather than being a direct consequence of reduced CI levels.

We used clustergrams to look at the expression of all genes associated with fatty acid beta-oxidation, triglyceride homeostasis, mitochondria, and protein synthesis (Fig. EV4A,B). We found that most genes associated with each of the GO terms were differentially regulated in our D + A CI KD flies. This common signature in D + A CI KD flies reinforced a suppression of both catabolic and anabolic metabolism in response to developmental CI dysfunction. KEGG pathway analysis further confirmed this (Fig. EV3D), with sugar, lipid and amino acid metabolism most significantly affected. These data also correlate with our findings of reduced lifespan, increased stress sensitivity and dysregulation of lipid homeostasis. In another fly model with neuronal-specific knockdown of ND-75, there was similarly a decrease in genes involved in translation and hexose metabolic (Granat et al, 2023) processes, supporting their pathogenic role in CI deficiency. This decrease in genes associated with translation is also noted in a brain-specific mouse model of Leigh Syndrome (McElroy et al, 2020), indicating that this response may be conserved across insects and mammals. However, the study by Granat et al, did not observe changes in the additional pathways we identified in ND-75 KD and ND-18 KD models when

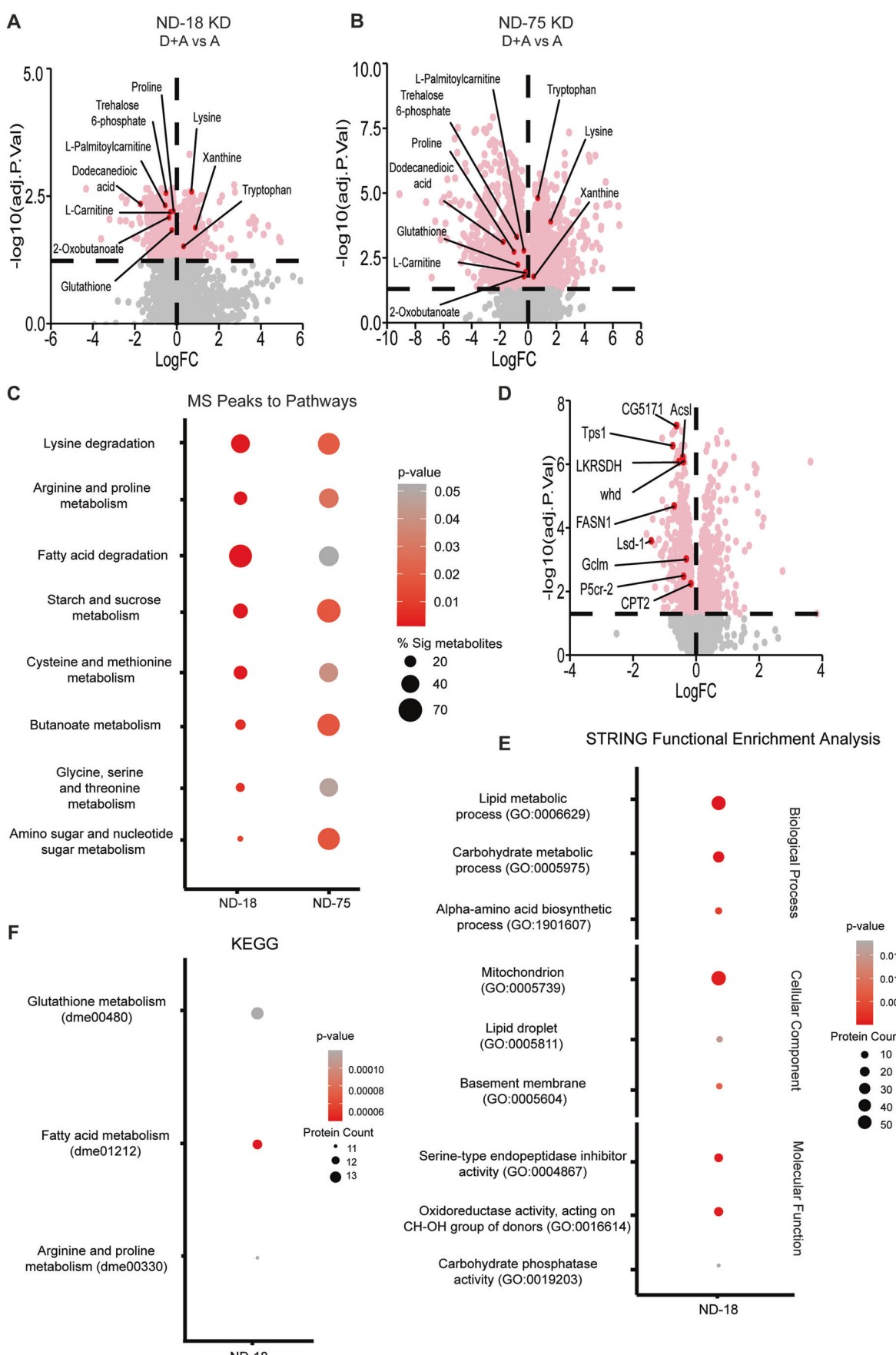

**Figure 4. Developmental CI depletion results in a specific metabolomic and proteomic signature.**

(A, B) Volcano plots of metabolites with significantly different abundance between flies where ND-18 (A) or ND-75 (B) was depleted from early development (D + A) vs from adulthood only (A-only). FDR, $P < 0.05$, $n = 4$ independent samples per experimental group. (C) Functional Enrichment Analysis using MetaboAnalyst MS Peaks to Pathways (KEGG) of the differential metabolic profiles of ND-18 and ND-75 KD D + A vs A-only, identified enrichment of the following categories. FDR, $P < 0.05$, $n = 4$ independent samples per experimental group. (D) Volcano plot of proteins with significantly different abundance between flies where ND-18 has been depleted from development (D+A) vs adulthood only (A-only). FDR, $P < 0.05$, $n = 5$ independent samples per experimental group. (E) Dot plot showing the STRING Functional Enrichment Analysis results of ND-18 KD D + A vs A-only flies. FDR, $P < 0.05$, $n = 5$ independent samples per experimental group. (F) Dot plot showing the most significant pathways according to KEGG altered at the protein level in ND-18 KD D + A flies. FDR, $P < 0.05$, $n = 4$ independent samples per experimental group. Source data are available online for this figure.

comparing D + A with A-only. In contrast, Granat et al (Granat et al, 2023) reported an up-regulation of genes regulated by ATF-4, which we did not detect. These discrepancies could be due to differences in the intensity of the knockdown during development and the choice to use whole flies for transcriptomics instead of heads in our study. Indeed, in a recent follow-up study, Granat et al, (Granat et al, 2024) did not observe ATF-4 activation when they reduced the intensity of CI depletion using a "weaker" RNAi line.

We chose to deplete CI ubiquitously, as is often the case in patients, but we know that mitochondria are enriched in tissues with higher energy demands, such as the skeletal muscle, heart, and brain. This is very clear when we look at the clinical phenotypes associated with mitochondrial dysfunction (Gorman et al, 2016). In the models presented here, we have observed alterations in the fat body in flies where CI dysfunction is induced by development (Figs. 2E and EV2E). We used tissue RNAseq enrichment data from Flyatlas2 (Leader et al, 2018) to see whether a specific tissue contributed to the shared transcriptional signature of D + A CI KD flies. We identified major contributions from three tissues: the fat body (Figs. 3E and EV3E), midgut (Figs. 3F and EV3F) and heart (Figs. 3G and EV3G). The fat body and midgut are essential to managing organismal energy balance. Several studies have demonstrated the role of mitochondrial metabolism in the proper differentiation and function of the intestine (Deng et al, 2018; Liu et al, 2016; Wisidagama and Thummel, 2019; Zhang et al, 2022), as well as its requirement in the fat body for long-range growth control (Banerjee et al, 2013; Song et al, 2017). In humans, the myocardium is the richest source of mitochondria (Page and McCallister, 1973), and in flies, proper mitochondrial dynamics are essential for cardiac function (Dorn et al, 2011). Our data imply that despite a ubiquitous decrease in CI function, the maladaptive response in D + A KD flies is most acute in the fat body, midgut, and heart. The fact that a significant part of the genes altered are highly expressed in the adult fat body aligns well with the results showing depletion in TAG, the absence of a fully developed adult fat body, and reduced resistance to starvation, as shown in the previous section.

## Metabolomic and proteomic analyses support transcriptomic data, indicating that an atrophied adult fat body likely accounts for the metabolic alterations in short-lived CI-depleted flies

We performed whole-fly metabolomics to investigate whether the distinct transcriptomic signature of D + A vs A-only CI KD flies extended to the metabolome. Treatment with RU-486 during

development resulted in a minor metabolomic response in control flies (control D + A vs A) (Fig. EV5A–C). As with our transcriptomic analysis, to remove any effects specific to RU-486 treatment, any metabolites found to be significantly altered in control flies were removed from our analysis of CI KD flies (ND-18 KD and ND-75 KD flies). A comparison of the metabolomes of D + A vs A-only revealed 418 significantly altered metabolites (260 downregulated and 158 upregulated) in ND-18 (Fig. 4A). This was notably escalated in D + A vs A-only ND-75 KD flies, where we detected 1475 significantly altered metabolites (919 downregulated and 556 upregulated) (Fig. 4B).

Using the MS Peaks to Pathways module (Pang et al, 2021) of MetaboAnalyst, we performed Functional Enrichment Analysis. We did this separately for ND-18 and ND-75 KD flies. Despite the disparity in the number of altered metabolites between ND-18 and ND-75 KD flies, there was a significant level of convergence amid the metabolic pathways significantly altered in D + A vs A-only KD flies. Most notably, like in our transcriptomic analysis, we found sugar, lipid, and amino acid metabolism to be most significantly affected (Fig. 4C) in short-lived D + A vs long-lived A-only CI KD flies, further supporting a role for metabolic plasticity in determining the effects of mitochondrial dysfunction and our hypothesis that this plasticity facilitates maladaptive responses that are detrimental to organismal fitness. Unsurprisingly, the metabolomic alterations found revolved around metabolic routes that preferentially occur in the fat body, such as fatty acid metabolism(Lehmann, 2018) or lysine degradation that in mammals occurs preferentially in the liver (Hallen et al, 2013), the functional equivalent of the fat body. Intriguingly, lysine degradation was also observed to be reduced in ND-75 KD neuronal-specific fly model, showing accumulation of lysine both when measured in brains (Granat et al, 2023) and in whole bodies (Fig. 4A–C). Lysine also accumulates in the muscle and olfactory bulb in mouse models of Leigh Syndrome (Terburgh et al, 2021; Terburgh et al, 2019). Therefore, alterations in this metabolic pathway are strong candidates for mediating pathogenic alterations associated with reduced survival in individuals with CI depletion since development.

Next, we asked if we could detect any changes in the proteome of our short-lived CI-depleted flies (Fig. EV5D) and whether these changes were correlated with the observed variations in the transcriptome and metabolome. In this case, due to the similarities observed between ND-75 and ND-18 models at the transcriptome and metabolome levels, we focused solely on the changes observed when ND-18 was targeted. Proteomic analysis of whole-fly homogenates confirmed the same depletion of CI subunits in short- and long-lived CI-depleted flies when compared to their respective controls, while subunits of CII, III, IV, and V were unaffected (Fig. EV5E). Similarly, knockout of *Ndufa9* in mouse liver only affects the levels of other CI subunits (Lesner et al, 2022).

This indicates a controlled response by mitochondrial proteases to degrade only unassembled OXPHOS components, a mechanism that seems to be evolutionarily conserved from flies to mice. Importantly, we detected no differences in the levels of any CI subunit (or in the components of any other respiratory complexes) when comparing short- (D + A) and long-lived (A-only) CI-depleted flies directly (Fig. EV5E). This further confirms that the differential effects on lifespan, gene expression, and metabolism are not attributable to varying levels of ETC function between D + A and A-only CI-depleted flies but instead are consequences of developmental maladaptation.

Using STRING (Szklarczyk et al, 2023), we performed a Functional Enrichment Analysis of proteins found to have significantly different abundances in D + A vs A-only in ND-18 KD (Figs. 4E,F and EV5F). The functional enrichment analysis revealed a distinct pattern of dysregulated metabolic processes, encompassing anabolic and catabolic lipid metabolism (GO:0006629, GO:0005811, GO:0016614), protein metabolism (GO:1901607, GO:0004867), and sugar metabolism (GO:005975, GO:0019203). These disruptions predominantly occur in the three tissues identified as most affected by our transcriptomic analysis: the fat body, midgut, and heart. This provides additional confirmation that moderate alterations in CI function during development lead to developmental maladaptations, preventing the proper differentiation of the adult fat body. The proteomic profiles of three different fly models of mitochondrial dysfunction—featuring mutations in clueless, Sod2, and Pink1—demonstrate a significant reduction in OXPHOS components, including CI subunits (Sen and Cox, 2022). However, the other pathways identified as potentially pathogenic in ND-18 KD flies remained unchanged in these different models, suggesting that the fat body alterations identified in this study could be specific to a dysfunctional ETC.

Our omics studies have revealed a recurring theme through GO/Enrichment/String analyses: dysregulation of energy homeostasis centered on an atrophied or dysfunctional adult fat body. Consequently, we conducted a manual inspection of the expression levels of genes, proteins and metabolites of pathways that are preferentially located in the adult fat body (Figs. 3A,B and 4A,B,D). Regarding lipids, we observed reduced concentrations of L-carnitine, L-palmitoyl carnitine, and dodecanedioic acid (Fig. 4A,B), alongside a decrease in triacylglyceride (TAG) levels (Fig. 2D) in short-lived CI-depleted flies. Accordingly, we observed a reduction in the mRNA and protein levels of FASN1 (involved in triglyceride biosynthesis) and whd and CPT (encoding two distinct carnitine palmitoyltransferases) among other enzymes involved in lipid metabolism (Fig. 4D). Concerning carbohydrates, we conducted a detailed inspection of trehalose metabolism. Trehalose is the principal circulating sugar in fly hemolymph, and its biosynthesis from glucose-6-phosphate occurs in the fat body (Matsushita and Nishimura, 2020). This process involves Tps1 (Trehalose-6-phosphate synthase 1), which catalyzes the first step, and CG5171, which dephosphorylates trehalose-6-phosphate to trehalose for export from the fat body. As expected from an atrophied adult fat body, we observed a decrease in the concentration of trehalose-6-phosphate and both Tps1 and CG5171 at the mRNA and protein levels.

Our omics data show several examples of alterations in amino acid metabolism. L-lysine degradation is a process that occurs preferentially in mammals' livers (Rao et al, 1992). In our omics studies, we detected a significant increase in L-lysine (Fig. 4A,B) alongside a reduction in the levels of the two primary enzymes involved in its degradation, LKRSDH and Aldh7A1 (Fig. 4D). Similar to the liver in mammals, the fat body is one of the tissues where glutathione concentration is higher (Lu, 2013). Accordingly, we observed a modest but significant reduction in the levels of glutathione in those flies with an atrophied adult fat body (Fig. 4A,B). However, this decrease was seen in only one of the enzymes involved in glutathione biosynthesis—Gclm—while the levels of other pathway components remained unaltered (Fig. 4A,B,D). Finally, we noticed a reduction in the levels of L-proline, which occurred in parallel with a decrease in the levels of the highly fat body expressed pyrroline-5-carboxylate reductase (P5cr-2) but not in the levels of P5cr, whose expression in the fat body is much lower according to FlyBase (Marygold et al, 2013). This further supports a specific role of the adult fat body in the metabolic dysregulation observed in short-lived CI-depleted flies.

In summary, the above results support the notion that the fat body is one of the primary tissues affected by mitochondrial dysfunction during development, impacting critical metabolic pathways at mRNA, protein, and metabolite levels.

## Conclusion

We have developed unique in vivo models in which we have temporally manipulated the function of CI by depletion of either ND-18 or ND-75. As expected, we observed a significant overlap in transcriptomic and metabolomic responses between both models. Over 50% of the genes altered in the ND-75 KD model changed in the same direction in the ND-18 KD model, with GO and KEGG analyses revealing high concordance in the pathways altered. Notably, more genes and metabolites were changed in the ND-75 KD than in the ND-18 KD model. Yet, the impact of ND-18 depletion on lifespan reduction was more pronounced, corresponding to a more significant decrease in mitochondrial respiration. A potential explanation, which requires experimental validation, suggests that a higher depletion in CI activity during development (27% in ND-18 knockdown versus 19% in ND-75) might restrict transcriptional and metabolic adaptability. Some developmental responses may reduce lifespan, while others could be adaptive, enhancing survival under conditions such as observed fat body atrophy. Furthermore, the distinct roles of ND-75 and ND-18 beyond CI assembly and function should be considered. For instance, ND-75's role in apoptosis induction (Elkholi et al, 2019) or ACAD9's dual role in fatty acid oxidation and CI biogenesis (Nouws et al, 2014) suggests that the absence of different CI subunits during development may prompt specific transcriptomic and metabolomic adaptations.

Our molecular and physiological characterization of short- and long-lived models of CI dysfunction establishes that development is the critical phase during which the degree of mitochondrial dysfunction determines lifespan and organismal fitness. Any significant alteration in the levels or function of CI during this period is likely to cause maladaptation, impacting adult lifespan in proportion to the severity of these changes. This clarification addresses significant contradictions in the field regarding the role of CI in longevity and mitochondrial diseases. However, the exact nature of the maladaptation remains to be established by future

investigations. These studies will need to determine why some tissues are more affected than others, whether the negative phenotypes observed have an epigenetic basis or another explanation, and if they can be totally or partially reversed by cellular reprogramming or similar interventions.

Concerning mitochondrial diseases, the intensity of mitochondrial maladaptation during development could potentially explain the varying ages at which mitochondrial disorders manifest, ranging from after birth to advanced ages (Gorman et al, 2015). Mutations that severely disrupt mitochondrial function during development would tend to manifest earlier. Thus, if such a critical period (as the one described in flies) exists in humans, it would imply that interventions aimed at correcting and restoring mitochondrial function must be implemented during this critical developmental window. For technical reasons, we were unable to perform a key experiment to restore CI levels in adults following disruption during development. The inherent leakiness and irreversibility of RNAi technology in *Drosophila* (Bosch et al, 2016; Poirier et al, 2008; Scialo et al, 2016b) prevents knockdown effects from being reversed simply by withdrawing RU-486 in adults previously exposed during development. While expressing human rescue constructs has successfully restored CI function in cases involving assembly factors such as Sicily (Zhang et al, 2013), this approach presents greater challenges when applied to subunits that form part of the final assembled complex, such as ND-75 or ND-18. This is likely due to insufficient conservation of amino acid sequences, which is inadequate to stabilize the assembled complex. A promising future strategy would involve combining RNAi with the overexpression of fly genes harboring silent mutations resistant to RNAi, offering a more effective and reliable rescue approach. Nonetheless, restoring CI expression in adults is unlikely to rectify developmental maladaptations fully. Evidence from brain-specific mouse models of Leigh disease indicates that postnatal CI rescue significantly extends lifespan but fails to restore health completely (Corrà et al, 2023; McElroy et al, 2020). Since these models are limited to brain tissue, it remains to be tested whether ubiquitous depletion of CI during development can be rescued in adults when other tissues, such as the heart, gut, or liver, which are particularly affected in fly models, are involved.

From the perspective of ageing, the fact that up to 75% depletion of CI does not shorten lifespan supports the idea that the age-related decline in CI observed in flies and humans (Cabre et al, 2017; Scialo et al, 2016a) could be an anti-ageing adaptation rather than a driver or accelerator of the ageing process. Accordingly, flies with reduced levels of CI induced by RNAi treatment (Copeland et al, 2009), as well as mice with lower levels of CI (Miwa et al, 2014), outlive individuals with higher levels of CI. In *Caenorhabditis elegans*, the effect of knocking down CI subunits contrasts with that observed in flies: during development, it extends lifespan, whereas in adulthood, it has no impact (Durieux et al, 2011). The reasons why flies and worms respond so differently and which adaptations derived from having low levels of CI may contribute to extending lifespan remain to be further investigated. Accordingly, future research must focus on understanding how CI level changes lead to development maladaptations and anti-ageing adaptations in adulthood. Identifying which of these can be targeted is crucial: addressing developmental alterations of CI is essential for treating mitochondrial diseases, while targeting adult changes could help delay ageing.

# Methods

## Reagents and tools table

| Reagent/resource | Reference or source | Identifier or catalog number |
|---|---|---|
| **Experimental models** | | |
| Tubulin-GeneSwitch | Gifted by Prof Scott Pletcher | |
| Daughterless-Gal4 | Bloomington Drosophila Stock Center | BL #55851 |
| Tubulin-GAL80ts | Bloomington Drosophila Stock | BL#7019 |
| UAS-KK library landing site | Vienna Drosophila Resource Center | VDRC#60100 |
| UAS-ND-18-IR | Vienna Drosophila Resource Center | VDRC#101489 |
| UAS-ND-75-IR | Vienna Drosophila Resource Center | VDRC#100733 |
| **Antibodies** | | |
| α-NDUFV2 | ThermoFisher | 15301-1-AP |
| α-tubulin | Abcam | ab179513 |
| α-mouse | Vector Labs | PI-2000 |
| α-rabbit | Vector Labs | PI-1000 |
| **Oligonucleotides and other sequence-based reagents** | | |
| Act88f F | agggtgtgatggtgggtatg | |
| Act88f R | cttctccatgtcgtcccagt | |
| ND-18 F | aagatcaccgtgccgactg | |
| ND-18 R | gacaatgggtcgccgctg | |
| ND-75 F | tgggagatccgtaaggtgag | |
| ND-75 R | ctcgttgacgtcctcgttct | |
| **Chemicals, enzymes, and other reagents** | | |
| mini EDTA-free protease inhibitor | Roche | 11836170001 |
| 2x Laemmli sample buffer | BioRad | 1610737 |
| Any kD Criterion TGX Stain-Free™ Protein Gels | BioRad | 5678124 |
| Clarity Western ECL Substrate | BioRad | 1705061 |
| Trizol | ThermoFisher | 15596026 |
| DNAse I | ThermoFisher | EN0521 |
| High-Capacity cDNA reverse transcription kit | Applied Biosystems | 4374967 |
| QuantiNova SYBR Green | Qiagen | 208052 |
| Quantitative Colorimetric Peptide Assay | Pierce | 23275 |
| TruSeq Stranded | Illumina | mRNA |
| LipidTOXred | Invitrogen | H34477 |
| Mifepristone (RU-486) | abcam | ab120356 |
| Bradford assay | ThermoFisher | A55866 |

| Reagent/resource | Reference or source | Identifier or catalog number |
|---|---|---|
| Resazurin | Sigma-Aldrich | R7017 |
| Alcohol dehydrogenase | Sigma-Aldrich | A3263 |
| Diaphorase | Sigma-Aldrich | D5540 |
| β-NAD | Sigma-Aldrich | N0632 |
| DC protein assay | BioRad | 5000116 |
| riboflavin 5'-monophosphate | Sigma-Aldrich | F8399 |
| Western blotting membranes, nitrocellulose | Amersham | GE10600002 |
| ProLong Diamond | ThermoFisher | P36965 |
| Triglycerides Reagent | ThermoFisher | TR22421 |
| **Software** | | |
| dplyr | https://dplyr.tidyverse.org/ | 1.0.10 |
| edgeR | https://bioconductor.org/packages/release/bioc/html/edgeR.html | 3.38.1 |
| extrafont | https://github.com/wch/extrafont | 0.18.0 |
| forcats | https://forcats.tidyverse.org/ | 0.5.2 |
| ggplot2 | https://ggplot2.tidyverse.org/ | 3.4.0 |
| ggrepel | https://ggrepel.slowkow.com/ | 0.9.2 |
| ggVennDiagram | https://cran.r-project.org/web/packages/ggVennDiagram/index.html | 1.2.2 |
| Glimma | https://github.com/Shians/Glimma | 2.6.0 |
| limma | https://bioconductor.org/packages/release/bioc/html/limma.html | 3.52.2 |
| pheatmap | https://cran.r-project.org/web/packages/pheatmap/ | 1.0.12 |
| purrr | https://purrr.tidyverse.org/ | 0.3.5 |
| RColorBrewer | https://cran.r-project.org/web/packages/RColorBrewer/ | 1.1-3 |
| readr | https://readr.tidyverse.org/ | 2.1.3 |
| sf | https://cran.r-project.org/web/packages/sf/ | 1.0-9 |
| stringer | https://stringr.tidyverse.org/ | 1.4.1 |
| tibble | https://tibble.tidyverse.org/ | 3.1.8 |
| tidyr | https://tidyr.tidyverse.org/ | 1.2.1 |
| tidyverse | https://www.tidyverse.org/ | 1.3.2 |
| MetaboAnalyst | https://www.metaboanalyst.ca/ | 5.0 |
| FlyEnrichr | https://maayanlab.cloud/FlyEnrichr | |
| Proteome Discoverer | ThermoFisher | v2.5 |
| String | https://string-db.org/ | |
| ImageJ | | v2.0.0 |
| GraphPad Prism | GraphPad | v9 |

| Reagent/resource | Reference or source | Identifier or catalog number |
|---|---|---|
| Oroboros DatLab | Oroboros | v7.0 |
| Applied Biosystems Step One software | Applied Biosystems | v2.3 |
| Flybase | https://flybase.org/ | |
| **Other** | | |
| Oxygraph 2-K | Oroboros | |
| FLUOstar Omega plate reader | BMG Labtech | |
| LAS-4000 CCD camera system. | FujiFilm | |
| Step One qPCR instrument | Applied Biosystems | |
| NextSeq2000 | Illumina | |
| Ultimate 3000 RSLC nano system/Orbitrap Eclipse | ThermoFisher | |
| Bead Ruptor | OMNI | |
| Multiskan FC | ThermoFisher | |
| Criterion Blotter wet transfer | BioRad | |
| TCS SP8 DLS LightSheet microscope | Leica | |

## Fly husbandry

Flies were maintained on standard media (1% agar, 1.5% sucrose, 3% glucose, 3.5% dried yeast, 1.5% maize, 1% wheat, 1% soya, 3% treacle, 0.5% propionic acid, 0.1% Nipagin) at 25 °C in a controlled 12 h light: dark cycle. Male flies were used in all experiments unless otherwise stated. To express the intended RNAi constructs, we employed two different expression systems: the GeneSwitch (GS) and the temperature-sensitive Gal80 (GAL80ts) coupled with GAL4. The GS system utilizes a modified GAL4 transcription factor that is activated by RU-486. In contrast, GAL80ts is a temperature-sensitive repressor of GAL4, which is active at 18 °C, preventing the expression of the UAS-regulated transgene. However, it degrades at temperatures between 29–32 °C, thereby allowing GAL4 to bind to UAS and enabling the expression of the transgene. For GS experiments, flies were placed on media supplemented with either vehicle (EtOH) or varying concentrations of RU-486. In experiments where the Gal4/Gal80$^{ts}$ system was utilized, flies were maintained at either 18 or 29 °C.

Experimental flies were collected within 48 h of eclosion using $CO_2$ anesthesia and kept at a density of 20 flies per vial.

## RU-486 feeding protocol

To activate *tubGS* from development, mating adults were flipped onto media containing titrated concentrations of RU-486 (up to 0.5 μM). To activate *tubGS* from adulthood, male progeny were collected and placed on standard media containing 500 μM RU-486.

## Survival analysis

Survival graphs were created using GraphPad Prism 9 with log-rank statistics applied. A minimum of 50 flies were used for each experimental group, and two-three replicate experiments were performed. Graphs presented depict pooled/representative results.

### Lifespan 25 °C and 29 °C

Flies were collected within 24 h of eclosion, maintained at a density of 20 flies per vial at the appropriate temperature (25 °C or 29 °C). They were transferred to fresh media every 2–3 days, and the number of dead flies was scored.

### Hydrogen peroxide treatment

In all, 3–5-day-old adult male flies were starved overnight and then transferred to vials containing 5 ml 1% agar, 5% Sucrose, 2.5% $H_2O_2$. The number of dead flies were scored at regular intervals.

### Thermal stress

Flies were collected within 24 h of eclosion, maintained at a density of 20 flies per vial at 32 °C. They were transferred to fresh media every 2–3 days, and the number of dead flies was scored.

### Starvation

To assay starvation, 3–5-day-old adult male flies were placed into vials containing 2 ml of 1% agar. The number of dead flies was scored at regular intervals.

## CI-linked respiration

Mitochondrial oxygen consumption was measured in homogenates of whole male L3 larvae and 3–5-day-old adults using a Oxygraph 2-K. 5 L3 larvae/10 flies were homogenized in isolation buffer (250 mM sucrose, 5 mM Tris–HCl, 2 mM EGTA, pH 7.4) and diluted 10× in assay buffer (120 mM KCl, 5 mM $KH_2PO_4$, 3 mM HEPES, 1 mM EGTA, 1 mM MgCl2, 0.2% (w.v.) BSA, pH 7.2). For L3 larvae, the following substrates were added, 5 mM Proline, 5 mM Pyruvate, 2.5 mM Glutamate, 2 mM Malate. For adults, 5 mM Proline and 5 mM Pyruvate were added. State 3 was initiated with the addition of 1 mM ADP. Raw $O_2$ Flux extracted using Oroboros DatLab 7.0 was normalized to the protein concentration (quantified via Bradford assay) of each homogenate (picomoles of $O_2$ per $min^{-1}$ per $mg^{-1}$).

## Western blot

In all, 3–5-day-old males of the indicated genotypes and experimental groups were homogenized in 150 μl of homogenization buffer (0.2% Triton X-100 supplemented with 1× complete mini EDTA-free protease inhibitor in PBS) using a motorized pestle homogenizer for ~1 min. After a 10 min incubation at 4 °C, samples were centrifuged at 13,000 × $g$ for 15 min at 4 °C. Protein concentration of the supernatants were quantified via the Bradford assay and a FLUOstar Omega plate reader/samples were mixed 1:1 with 2× Laemmli sample buffer supplemented with 5% β-ME and boiled at 95 °C for 5 min. 30 μg of protein was run on Any kD™ Criterion™ TGX Stain-Free™ Protein Gels and transferred to nitrocellulose membranes using the Criterion™ Blotter wet transfer system. Membranes were blocked in 5% Milk in PBS, 0.1% Tween-20 for 1 h at room temperature. They were then washed with PBS and incubated with primary antibodies (α-NDUFV2 (1:1000); α-tubulin (1:1000)) in 5% milk 0.1% Tween-20, overnight at 4 °C. Membranes were washed 3× for 10 min with PBS with 0.1% Tween-20. This was followed by incubation with HRP conjugated secondary antibodies (α-mouse (1:5000) and α-rabbit (1:5000)) in 5% Milk in PBS, 0.1% Tween-20 for 1 h at room temperature. Membranes were washed as above. Hybridization was visualized via Clarity Western ECL Substrate and using a LAS-4000 CCD camera system.

## qPCR gene expression analysis

Total RNA was extracted in quadruplicate from male L3 larvae and 3–5-day-old adults of the indicated genotypes and experimental groups using standard Trizol extraction. RNA was DNAse I treated and purified via ethanol precipitation. cDNA was synthesized using the High-Capacity cDNA reverse transcription kit according to the manufacturer's instructions. Relative expression of the target genes was quantified using QuantiNova SYBR green reagents and Applied Biosystems Step One qPCR instrument. Data were extracted and analyzed using Applied Biosystems Step One software. Relative expression of the target genes (ND-18 and ND-75) was normalized to levels of the housekeeping gene act88f. For each target, mean fold change with standard error mean is presented.

## RNA sequencing and differential expression analysis

For RNAseq, total RNA was extracted as outlined in qPCR gene expression analysis. RNA samples were submitted to the University of Glasgow Polyomics facility for QC (Agilent), PolyA selection, Library preparation (TruSeq Stranded mRNA Library Prep), and sequencing using the Illumina NextSeq platform (NextSeq2000 P3 Reagents, NextSeq 500/550 High Output Kit v2.5).

Raw reads were pre-processed through TrimGalore, which combines FastQC to determine quality and Cutadapt (Martin, 2011) to trim reads if necessary. Mapping of the trimmed reads was conducted in HISAT2 (Kim et al, 2019). All reads were aligned to the pre-built Hisat2 D. Melanogaster BDGP6 index (obtained through HISAT2's website—last accessed 14/9/22). The resultant Sam files were sorted and converted to Bam files using Samtools (Danecek et al, 2021), counts were then established using StringTie (Pertea et al, 2016). Each sample was annotated via StringTie using "dmel-all-r6.41.gtf" (available from Flybase) as a reference. Differential expression analysis was performed by utilizing the limma package (Ritchie et al, 2015).

For GeneOntology and Pathway Analysis, Gene names were retrieved from Flybase, before being sorted by FDR-adjusted P value (q value). Genes were determined to be significant if they had a q value less than 0.05. The significant gene names, ranked by their q value, were then submitted to FlyEnrichr (Chen et al, 2013) where the results for GO Biological Process, Cell Component, Molecular Function 2018 and KEGG 2019 were assessed and ranked by Combined Score to establish the most representative pathways in each comparison.

All graphics associated with the DE analysis were created in RStudio.

## Metabolomics

Metabolomic analysis via LC-MS was performed on quadruplicate samples of 3–5-day-old male adults of the indicated genotypes and

experimental groups. Anesthetized flies were homogenized in ice-cold extraction buffer (50% HPLC grade methanol (Fisher), 30% HPLC grade acetonitrile (Sigma), 20% water) and centrifuged at $13,000 \times g$ at 4 °C. Supernatants were stored at −80 °C and transferred to LC-MS vials before analysis. LC-MS and data analysis were performed as previously described (Maddocks et al, 2017). Analysis of peak data was performed using MetaboAnalyst 5.0 (Pang et al, 2021).

## Proteomics

Mass spectrometry analysis was performed at the Proteomics Facility of the Medical Research Council Toxicology Unit University of Cambridge, Cambridge, UK. Briefly, 50 5-day-old adult males were homogenized in 450 µL of 100 mM Triethylammonium bicarbonate (TEAB) for 5 cycles of 20 s at 5500 rpm with 10 s intervals at 4 °C. 1% RapiGest buffer and a further 450 µL of 100 mM TEAB buffer was added to the samples, followed by sonication and centrifugation at $2000 \times g$ for 5 min at 4 °C. In total, 850 µL of supernatant was collected for further processing. 50 µg of protein extract was digested in 50 µL 100 mM TEAB. Samples were denatured at 80 °C for 10 min. 2.8 µl of 72 mM DTT was added, and samples were heated at 60 °C for 10 min. 2.7 µl of 266 mM Iodoacetamide was added, and samples were incubated at room temperature for 30 min. DTT concentration was raised to 7 mM to quench alkylation. Samples were digested with 1 µg of trypsin (5 µL of 0.2 µg/µL stock) overnight at 37 °C. Digests were quantified using the Peptide quantification assay from Pierce™ Quantitative Colorimetric Peptide Assay according to manufacturer instructions. 2xTMT11 with pooled reference standards were used to multiplex samples. LC-MS/MS analysis was performed using synchronous precursor selection (SPSMS3), with MS3-based quantification triggered by real-time search of Drosophila MS2-identified peptides. Injected samples were analyzed using an Ultimate 3000 RSLC™ nano system coupled to an Orbitrap Eclipse™ mass spectrometer. Data were acquired using three FAIMS CVs (−45 v, −60 v, and −75 v) and each FAIMS experiment had a maximum cycle time of 2 s. For each FAIMS experiment the data-dependent SPSMS3 RTS program used for data acquisition consisted of a 120,000 resolution full-scan MS scan (AGC set to 50% (2e5 ions) with a maximum fill time of 30 ms) using a mass range of 415–1500 $m/z$. Raw data were imported and processed in Proteome Discoverer v2.5. The raw files were submitted to a database search using Proteome Discoverer with Sequest HT against the Uniprot UP000000803 Drosophila database (containing 21135 seq. accessed on 2022/08/04). Common contaminant proteins (human keratins, BSA and porcine trypsin) were added to the database. Statistical analyses for fold change and significance were performed similarly as previously described(Popovic et al, 2021). Proteins exhibiting significant changes were further subjected to Functional Enrichment analysis using the STRING database (Szklarczyk et al, 2023).

## Immunofluorescence and confocal microscopy

Adult fat bodies were dissected and fixed in 4% paraformaldehyde at room temperature for 30 min. Tissues were first rinsed in 1×PBS and then incubated with LipidTOXred 1:500 and DAPI (1:1000) in 1× PBS + 0.005% Saponin for 1 h at room temperature. Tissues

were then washed 3× for 15 min in 1× PBS + 0.005% Saponin and mounted with Prolong Diamond on polylysine-coated glass slides with 13 mm × 0.12 mm spacers. Confocal images were taken on a Leica SP8 DLS and processed using ImageJ.

## TAG assay

In all, 4 × 15 males adult flies per indicated genotype and condition were weighed and then homogenized in Tet buffer (10 mM Tris, 1 mM EDTA, pH 8.0 with 0.1% v/v Triton X-100) using glass beads and an OMNI Bead Ruptor. Samples were centrifuged at $13,000 \times g$ for 5 min at 4 °C. The supernatant was incubated at 72 °C for 15 min. 3 µl of sample and glycerol standards were added to 300 µl of ThermoFisher Infinity Triglyceride reagent and incubated for 15 m at 37 °C. Absorbance at 540 nm was measured on a Multiskan FC (ThermoFisher). A glycerol standard curve, weight and number of flies was used to calculate TAG/mg/fly.

## NAD/NADH measurements

Measurements of NAD+ and NADH in whole flies were performed as described in (Kanamori et al, 2018; Kataura et al, 2022). Briefly, NAD+ or NADH were extracted from triplicates of 10 (NAD + ) or 20 flies (NADH) homogenized with a motorized pestle using 20% trichloroacetic acid (TCA) (Sigma) or 0.5 M sodium hydroxide (NaOH, Sigma) and 5 mM EDTA (Sigma) respectively. The sample pH was adjusted to 8.0 with 1 M Tris. Levels of NAD+ and NADH were determined by levels of resorufin fluorescence produced from the enzymatic cycling reaction using resazurin, riboflavin 5'-monophosphate, alcohol dehydrogenase and diaphorase. Fluorescence intensity was measured once per minute for a total of 60 min using a FLUOstar Omega. NAD+ and NADH levels were determined by a β-NAD (Sigma) standard curve and adjusted to protein concentration determined by the DC protein assay.

## Statistical analysis

All bar graphs and survival curves were generated using GraphPad Prism 9. Statistical comparisons were conducted using Student's $t$ tests or the appropriate nonparametric Kolmogorov–Smirnov test, as specified in the figure legends. Survival curves were analyzed using a log-rank test. All other graphs were produced using RStudio. The ROUT method (Robust regression and Outlier removal), available in Prism, was employed to identify and remove outliers prior to performing the appropriate statistical significance tests. No blinding procedures were implemented in any of the experiments presented. All data are presented as means with SEM unless otherwise stated

## Schematic diagrams

All schematic diagrams were created using the biorender.com platform.

# Data availability

RNA-sequencing data are available under accession number GSE237015 on the NCBI Gene Expression Omnibus database.

The mass spectrometry proteomics data have been deposited in the ProteomeXchange Consortium via the PRIDE partner repository with the dataset identifier PXD043791.

The source data of this paper are collected in the following database record: biostudies:S-SCDT-10_1038-S44319-025-00416-6.

## Peer review information

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

## Acknowledgements

This research was supported by a Sir Henry Wellcome Postdoctoral Fellowship to R.S (204715/Z/16/Z), a Wellcome Senior Research Fellowship (212241/A/18/Z), and BBSRC grants (BB/R008167/1 & BB/W006774/1) to AS and fellowships from the Uehara Memorial Foundation and International Medical Research Foundation to TK. The work of the laboratory led by LMM is funded by the UK Medical Research Council intramural project MC_UU_00025/3 (no. RG94521). We thank the Vienna and Bloomington Drosophila Stock Centres for fly stocks. The authors thank D McGuinness and J Galbraith (University of Glasgow, Polyomics) for processing RNAseq samples. The authors thank Dr. Catarina Franco and Dr. Bini Ramachandran for their help with developing the proteomic sample processing pipeline, sample preparation, and data processing and for helpful discussions on sample analysis and experimental design.

## Author contributions

**Rhoda Stefanatos**: Conceptualization; Formal analysis; Investigation; Visualization; Methodology; Writing—original draft; Writing—review and editing. **Fiona Robertson**: Formal analysis. **Beatriz Castejon-Vega**: Investigation. **Yizhou Yu**: Formal analysis; Investigation; Methodology. **Alejandro Huerta Uribe**: Data curation; Formal analysis; Investigation. **Kevin Myers**: Formal analysis; Investigation; Methodology. **Tetsushi Kataura**: Formal analysis; Funding acquisition; Investigation. **Viktor I Korolchuk**: Formal analysis; Funding acquisition; Writing—original draft; Project administration. **Oliver D K Maddocks**: Formal analysis; Investigation. **L Miguel Martins**: Funding acquisition; Investigation; Writing—original draft; Project administration. **Alberto Sanz**: Conceptualization; Formal analysis; Supervision; Funding acquisition; Visualization; Methodology; Writing—original draft; Project administration; Writing—review and editing.

Source data underlying figure panels in this paper may have individual authorship assigned. Where available, figure panel/source data authorship is listed in the following database record: biostudies:S-SCDT-10_1038-S44319-025-00416-6.

## Disclosure and competing interests statement

VIK is a Scientific Advisor for Longaevus Technologies. The remaining authors declare no competing interests.

# Expanded View Figures

**Figure EV1.   Related to Figure 1.**

(A) mRNA quantification of KD in ND-18 KD (left) and ND-75 KD (right) adult males. Kolmogorov–Smirnov test, $P = 0.100$ (ND-18 KD) & $P = 0.100$ (ND-75 KD), $n = 3$ independent samples per experimental group. (B) mRNA quantification of KD in ND-18 KD (left) and ND-75 KD (right) 3rd instar larvae in the presence (drug) or absence (vehicle) of the GS inducer, RU-486. *T* test, $P = 0.0003$ (ND-18 KD) & $P = 0.0009$ (ND-75 KD), $n = 4$ independent samples per experimental group. (C) Western blot analysis of CI core subunit, NDUFV2 (ND-24) in control males and males where ND-18 or ND-75 has been depleted from early development (D + A) or from adulthood only (A-only), beta-actin was used as a loading control. ANOVA with Tukey's multiple comparison test ($P = 0.0059$ (ND-18 KD) & $P = 0.0069$ (ND-75 KD)). $n = 3$–6 independent samples per experimental group. (D) Quantification of ND-18 (upper) and ND-75 (lower) mRNA levels in adult males exposed to the inducer from early development (D + A) or from adulthood (A-only). *T* test, $P = 0.2852$ (ND-18 KD) & $P = 0.5807$ (ND-75 KD), $n = 3$ independent samples per experimental group. (E) Survival of male control flies exposed to the inducer, RU-486, from early development (D + A) or from adulthood (A-only). Log-rank test, $P = 0.1120$, $n = 223$–262 individuals per experimental group. (F) Survival of female flies where CI subunit, ND-18, has been depleted from early development (D + A) or from adulthood (A-only). Log-rank test, $P < 0.001$, $n = 50$–58 flies per experimental group. (G) Survival of female flies where CI subunit, ND-75, has been depleted from early development (D + A) or from adulthood (A-only). Log-rank test, $P < 0.001$, $n = 55$–57 flies per experimental group. (H) Levels of CI-linked respiration in control adult males exposed to the inducer from early development (D + A) or from adulthood (A-only). *T* test, $P = 0.2045$, $n = 10$–11 independent samples per group. (I) % Eclosion of flies where depletion of CI subunit, ND-75, has been induced at distinct developmental stages. $n = 5$ independent samples per experimental group. (J) Survival of male flies where depletion of CI subunit, ND-75, has been induced at distinct developmental stages. $n = 92$–100 flies per experimental group. (K) % Eclosion of control male flies moved from 18 °C to 29 °C at distinct developmental stages. $n = 4$–5 independent samples per experimental group. (L) Survival of control male flies moved from 18 °C to 29 °C at distinct developmental stages. $n = 26$–72 flies per experimental group. In bar graphs, means are presented with ± SEM. Significance is indicated as follows: *$P < 0.05$, **$P < 0.01$, ***$P < 0.001$; ns = not significant.

                                           

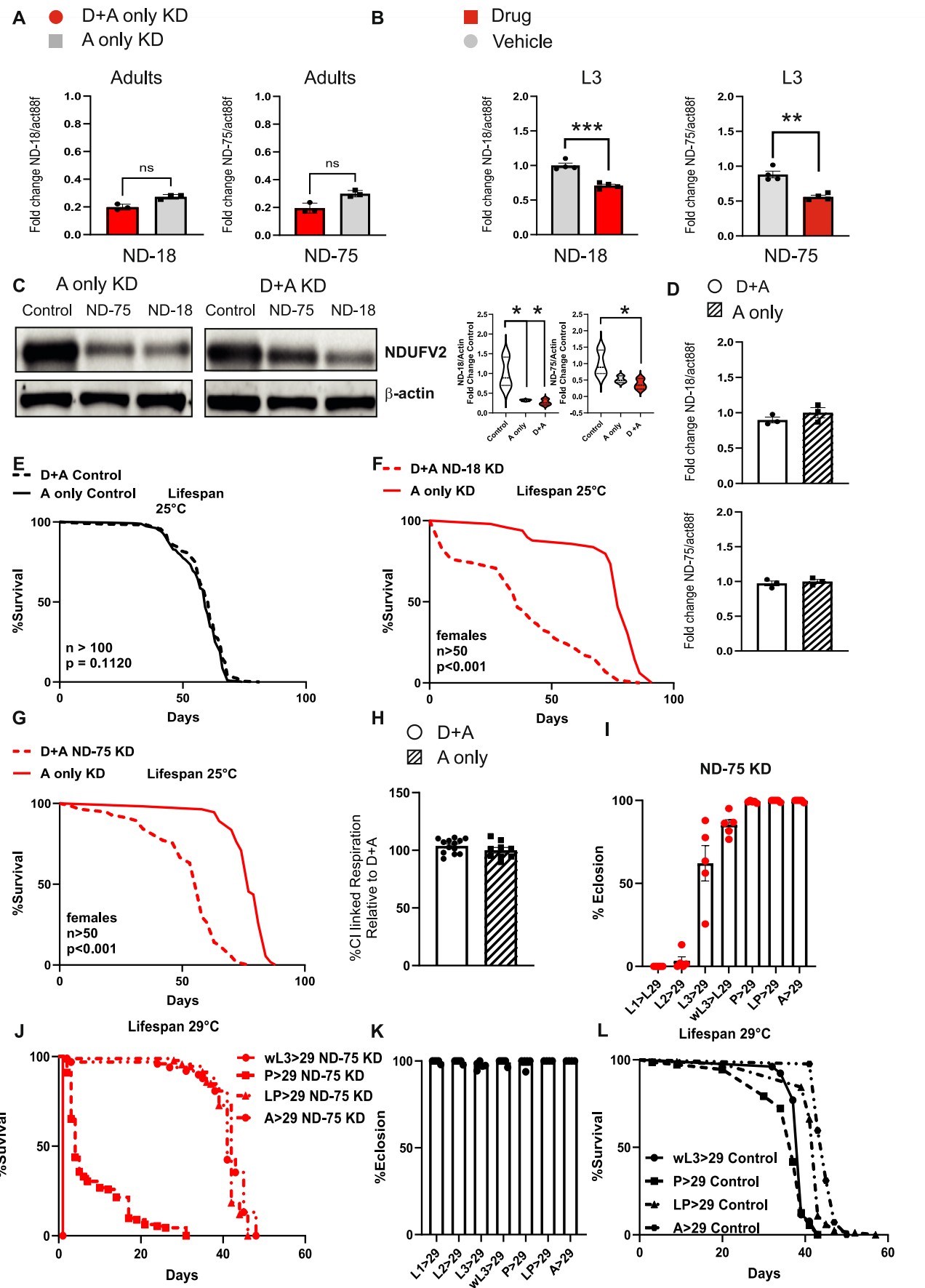

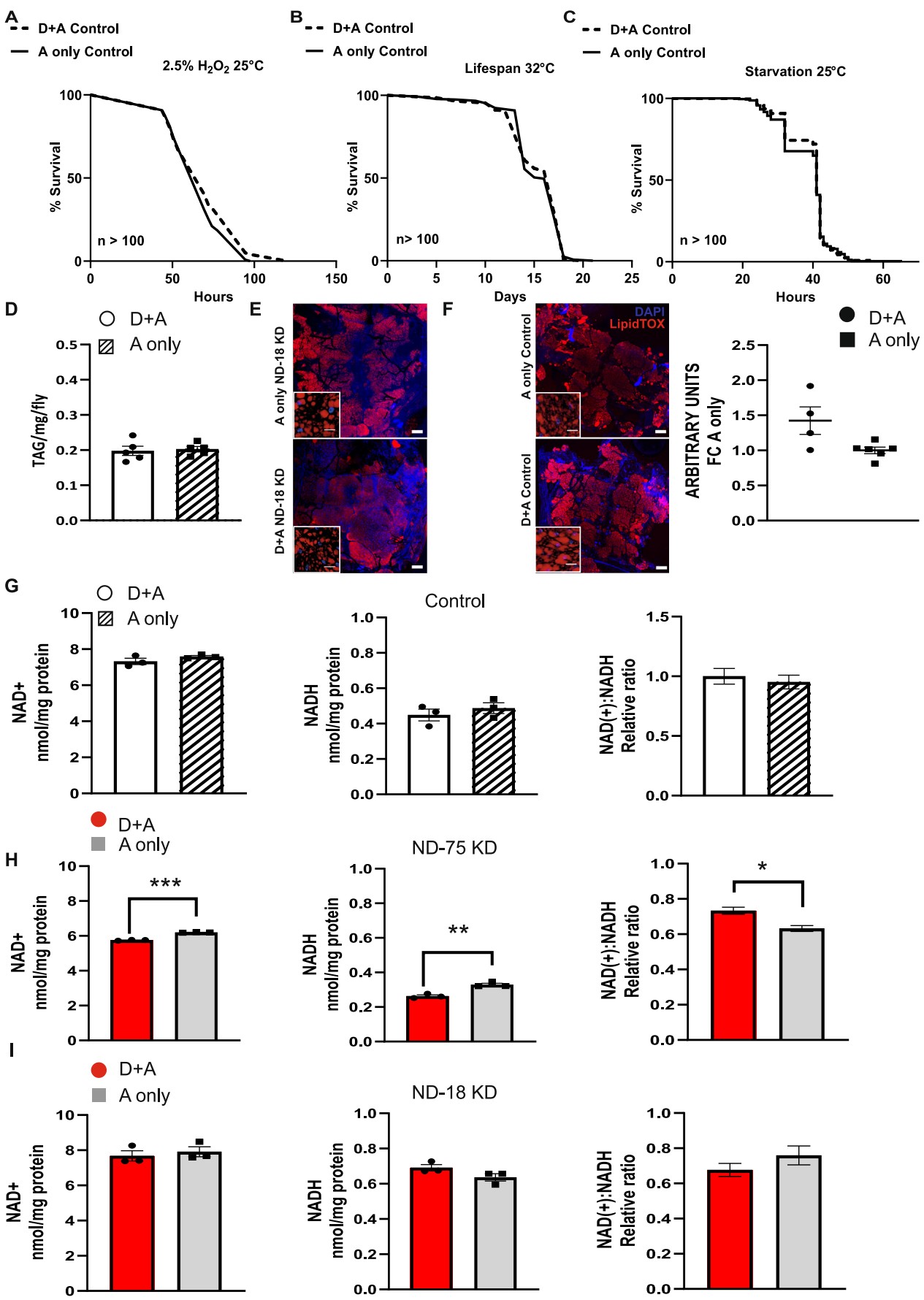

**Figure EV2. Related to Figure 2.**

Survival under oxidative stress (**A**), Log-rank test, $P = 0.0979$, $n = 156$–$159$ individuals per group), thermal stress (**B**), Log-rank test, $P = 0.5462$, $n = 274$–$281$ individuals per group), and starvation (**C**), Log-rank test, $P = 0.8445$, $n = 250$–$254$ individuals per group) conditions of control male flies exposed to the inducer, RU-486 from early development (D+A) or from adulthood (A-only). (**D**) Quantification of triacylglyceride levels in control male flies exposed to the inducer, RU-486 from early development (D+A) or from adulthood (A-only). T test, $P = 0.7662$. $n = 5$ independent samples per experimental group. (**E**) Confocal imaging of fat bodies from male flies where ND-18 has been depleted from D + A or A-only stained with LipidTOX Red and DAPI, scale bar (90 µM), inset scale bar (10 µM). $n = 2$ independent samples per group. (**F**) Confocal imaging of fat bodies from male flies exposed to the inducer, RU-486 from early development (D+A) or from adulthood (A-only) stained with LipidTOX Red and DAPI, scale bar (90 µM), inset scale bar (10 µM). T test, $P = 0.1168$, $n = 4$–$6$ independent samples per experimental group. (**G**) Levels of NAD+ (t test, $P = 0.2269$), NADH (t test, $P = 0.4408$), and the relative ratio of NAD(+):NADH (t test, $P = 0.6101$) in control flies. $n = 3$ independent samples per group. (**H**) Levels of NAD+ (t test, $P < 0.001$), NADH (t test, $P = 0.0041$), and the relative ratio of NAD(+):NADH (t test, $P = 0.0179$) in ND-75 KD flies. $n = 3$ independent samples per group. (**I**) Levels of NAD+ (t test, $P = 0.5993$), NADH (t test, $P = 0.1191$), and the relative ratio of NAD(+):NADH (t test, $P = 0.2759$) in ND-18 KD flies. $n = 3$ independent samples per group. In bar graphs, means are presented with ± SEM. Significance is indicated as follows: $*P < 0.05$, $**P < 0.01$, $***P < 0.001$; n.s. = not significant.

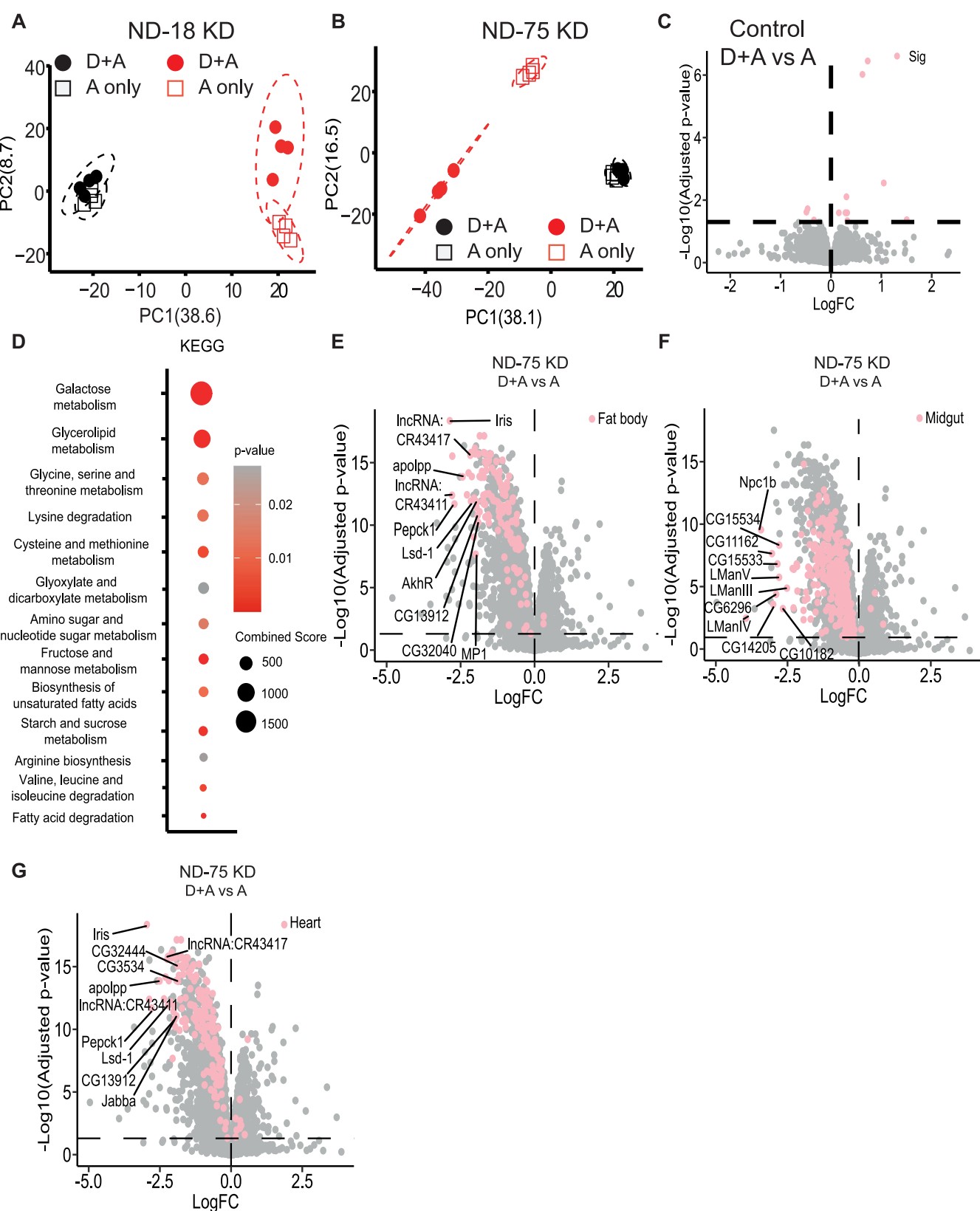

**Figure EV3.   Related to Figure 3.**

(A) PCA analysis of controls and ND-18 KD D + A and A-only conditions. $n = 4$ independent samples per group. (B) PCA analysis of controls and ND-75 KD D + A and A-only conditions. $n = 4$ independent samples per group. (C) Volcano plot of genes significantly differentially expressed between control D + A and control A. FDR, $P < 0.05$, $n = 4$ independent samples per group. (D) KEGG Enrichment Analysis (FlyEnricher) performed on those genes significantly altered in both ND-18 D + A vs A-only and ND-75 D + A vs A-only reveals significant enrichment in the following classifications. FDR, $P < 0.05$, $n = 4$ independent samples per group. (E–G) Volcano plots of genes significantly differentially expressed between ND-75 D + A and ND-75 A-only (gray), highlighted in pink, are those genes with highly enriched expression in the fat body (G), midgut (H) and heart (I). FDR, $P < 0.05$, $n = 4$ independent samples per group.

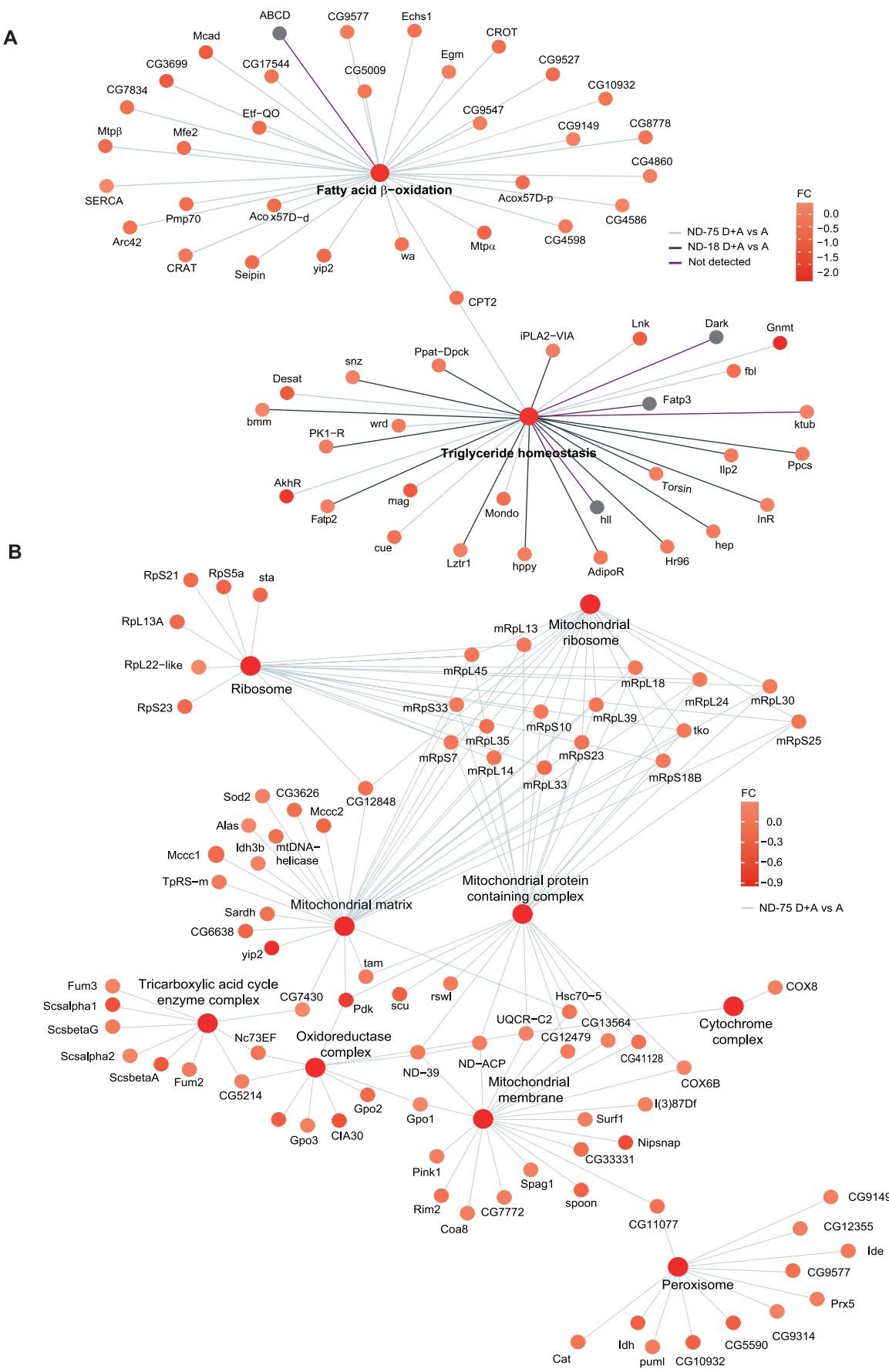

◄ **Figure EV4. Related to Figure 3.**

(A, B) Clustergrams of the indicated GO terms expressing the differential expression (FC) of genes associated with this term. All FCs are from ND-75 KD D + A vs A-only comparison unless otherwise indicated by a dark gray line (ND-18 KD D + A vs A-only) or a purple line (not detected in our study). $n = 4$ independent samples per group.

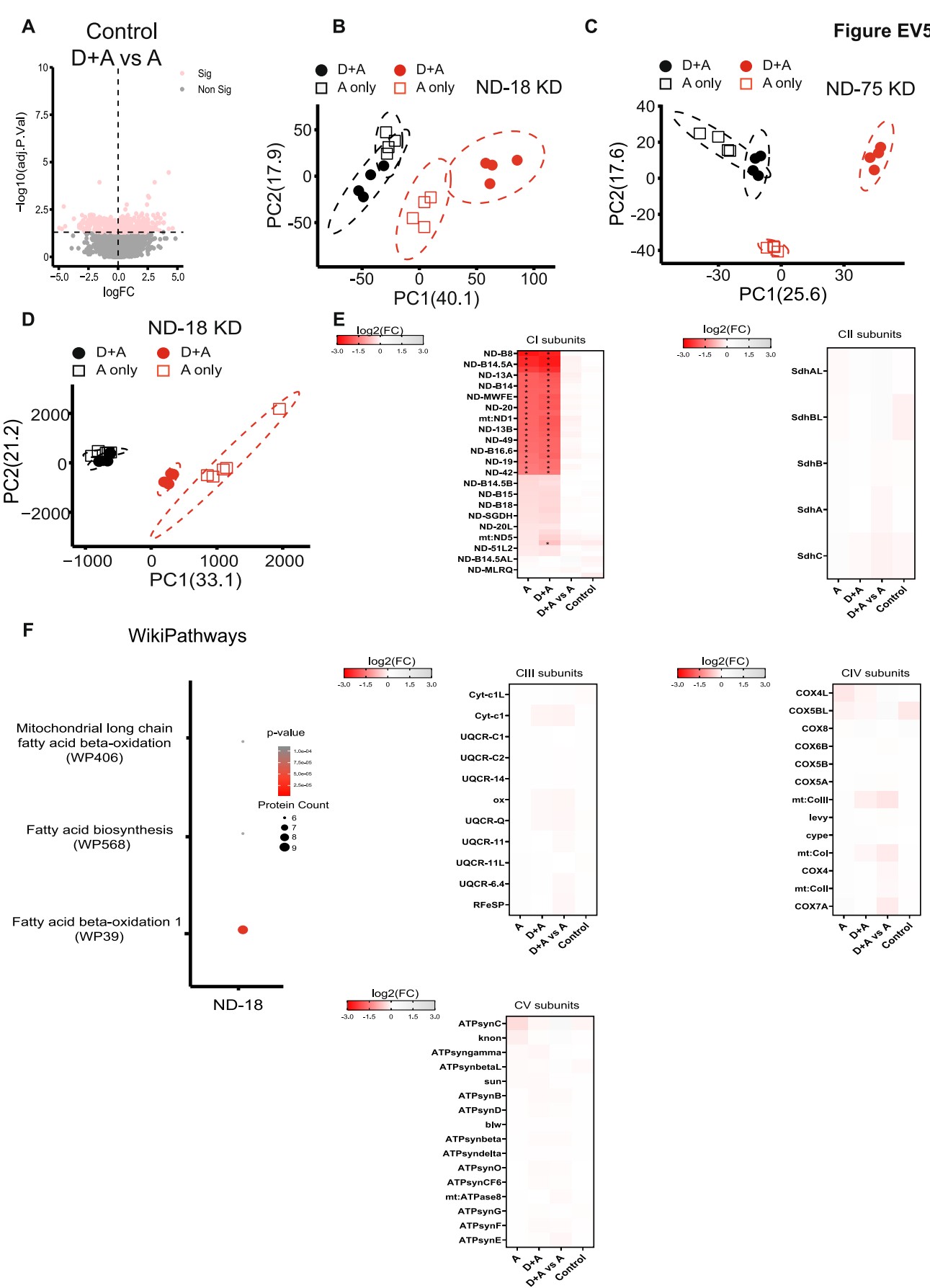

Figure EV5

**Figure EV5.** Related to Figure 4.

(A) Volcano plot of metabolites with significantly different abundance between control D + A and control A-only. FDR, $P < 0.05$, $n = 4$ independent samples per group. (B, C) PCA analysis of metabolomics data from controls and ND-18 KD (B) and controls and ND-75 KD (C) D + A vs A-only conditions. $n = 4$ independent samples per group. (D) PCA analysis of proteomics data from control and ND-18 KD D + A vs A-only conditions. $n = 5$ independent samples per group. (E) Heat maps depicting the expression of OXPHOS subunits (Complexes I–V). Asterisks (*) indicate significant differences in protein expression, FDR, $P < 0.05$. The comparisons are: (A) ND-18 A *versus* control A, (D + A) ND-18 D + A *versus* control D + A, (D + A vs A) ND-18 D + A *versus* ND-18 A, and (Control) control D + A versus control A. $n = 5$ independent samples per group. (F) Dot plot showing the most significant pathways according to STRING wikiPathways altered at the protein level in ND-18 KD D + A flies. FDR, $P < 0.05$. $n = 5$ independent samples per group.

