## [Peer Review File · EMBO Reports]

Developmental mitochondrial Complex I activity determines lifespan

Rhoda Stefanatos, Fiona Robertson, Beatriz Castejon-Vega, Yizhou Yu, Alejandro Huerta Uribe, Kevin Myers, Tetsushi Kataura, Viktor Korolchuk, Oliver Maddocks, L. Martins, and Alberto Sanz

Corresponding author(s): Alberto Sanz (Alberto.SanzMontero@glasgow.ac.uk) , Rhoda Stefanatos (rhoda.stefanatos@glasgow.ac.uk)

Review Timeline:

Transfer Date:	14th Oct 24
Editorial Decision:	9th Dec 24
Revision Received:	20th Jan 25
Accepted:	21st Feb 25

Editor: Deniz Senyilmaz Tiebe

Transaction Report: This manuscript was transferred to

EMBO reports following peer review at Review Commons.

**Review
COMMONS**

Review #1

1. Evidence, reproducibility and clarity:

Evidence, reproducibility and clarity (Required)

The manuscript by Stefanatos et al. reports interesting observations regarding the differential effects of knocking down two subunits of mitochondrial complex I (CI) at two different points in the life of the fruit fly, i.e., during larval stage vs. adult age. The data contained in the manuscript clearly show that disrupting CI during *D. melanogaster* larval development produces a shortening of the lifespan, whereas a similar CI deficiency but induced in adulthood actually increases the lifespan of the flies, reconciling the data available in the literature. The work in this manuscript also points out to a different metabolic adaptation in the different stages of the fly as the cause of the age-related differential effects.

However, there are certain points in the manuscript that deserve clarification in order to improve its readability and consistency in the interpretation of the results.

****Major points:****

1. It is not clear what the controls are in each of the experiments, i.e., whether it is the genetic control (RNAi control but non induced) or the non-RNAi flies but treated with the inducing drug (RU486). This distinction is important because RU486 is well known to affect gene expression (including expression of genes involved mitochondrial function: <https://www.nature.com/articles/s41514-019-0036-8>). For example, in figure S1 D, one would get the impression that the expression of ND-75 is increased 1.5-fold after RU486 administration in the adults.
2. To better understand and interpret the results, it would be much better to include all the data regarding the controls (lifespan plots, activity graphs, transcript levels, histology and EM images, etc.) together with the KD data in the main figures, instead of showing them in separate supplementary figures.
3. All relevant data mentioned in the manuscript, such as the eclosion rates, climbing assay results or the fat body findings, should be shown and not refer to them as "data not shown".
4. The extent of the defect in the CI-linked respiratory rates in Figure 1D seems to be comparable in the D+A and the A flies (around 25% for ND-18 and 50% for ND-75 KD), however the decrease in the mRNA levels is more profound in the D+A KDs (Figure S1A). This makes sense, as the duration of the KD induction is longer in the D+A than in the A only, but how can it be reconciled with the functional effects? Probably because of the lack of good antibodies for the target proteins, NDUFV2 protein levels were analyzed by western blot and immunodetection (Figure S1C). However, they do not seem to reflect the findings at the mRNA level, because the amounts of NDUFV2 in the D+A KDs appears higher than in the A only KDs. This is in any case difficult to assess properly as there is no densitometric quantification data and the two types of KDs seem to have been run in different gels. It would be convenient to quantify this properly and to run Blue-Native gels to determine the amounts of fully assembled CI in each of the samples, instead of the steady-state levels of the subunit, which could be present but not assembled.
5. The transcript and protein levels, as well as the respiratory activities in the adult flies were

assessed at 3-5 days of age, do the authors have any idea if these change at longer times?

6. In the methods section: 'For L3 larvae the following substrates were added, 5mM Proline, 5mM Pyruvate, 2.5mM Glutamate, 2mM Malate. For adults 5mM Proline, 5mM Pyruvate was added'. Is there any particular reason why the authors use different substrates for analyzing respiration at different developmental stages?

7. The most critical point is the fact that two previous reports exist in the literature in which constitutive KDs of the same two CI genes have been done from larval stage, with similar findings in terms of reduction of CI activity and life span

(<https://academic.oup.com/hmg/article/23/17/4686/741708> and

<https://journals.biologists.com/dmm/article/11/3/dmm032482/53188/Feeding-difficulties-a-key-feature-of-the>) . Both these papers claim that the reduction in life span is due to pathological findings, such as neurodegeneration or difficulties to feed. The authors should discuss their findings in the context of these previous papers and also discuss possible discrepancies between the previous reports and the present one.

8. Related to the previous point, in the images of the head section of the D+A ND-75 fly in Figure 2A, there seems to be pathological findings (unless for some reason that sample was damaged). Same applies to the EM images from the RNAi muscles, where there seems to be signs of vacuolization.

9. Again, it is not clear what kind of control is used in the -omics analyses, schematized in Figures S3A and S4A (see point #1) and how the genes that are differentially expressed due to the RU486 treatment are filtered out using a customized bioinformatics pipeline. This could introduce some bias in the analysis and could have been determined experimentally by performing an analysis comparing RU486-treated vs. untreated flies.

10. It is not clear why the authors carried out the metabolomics analyses in the whole fly instead of focusing on the tissues where they see the transcriptomic signature.

11. In the description of the findings in Figure 3, the authors state "ND-75 KD flies showed a greater response than ND-18 KD flies." But a few sentences later the text reads: "...the response to CI dysfunction in both models overlaps highly." However, in the results shown previously in the manuscript, ND-75 KD flies show a significantly less severe CI defect and less effect on lifespan. Please, discuss these apparently contradictory points.

12. There is a lack of depth in the analysis and description of the results of the -omics experiments at the end of the paper. Apart from the generic gene ontology and principal component analysis plots, it would be convenient to provide specific information about which are the actual transcripts, metabolites and proteins that change the most, and if the data from the transcriptomics, proteomics and metabolomics actually fit well with each other. The name of the most significant hits should be shown in the volcano plots shown in the figures, helping the reader to interpret and understand the final point of this report in a quicker way (it remains rather obscure in the current version of the manuscript).

****Minor point:****

1. Figures S2, S3 and S4 are too large.

2. Significance:

Significance (Required)

The strength of the work shown in this manuscript is that it clearly shows the differential effect of mitochondrial complex I deficiency in *D. melanogaster* depending on whether the deficiency occurs during larval development or after reaching the adult state, producing either lifespan shortening or expansion, respectively. The manuscript also demonstrates that the transcriptional and/or metabolic adaptive responses to complex I deficiency are common to the lack of two different subunits, but are different depending on whether the deficiency happened early or late in life.

One of the main limitations is the use of RU486 as an inducer of the KDs, which could give confounding results due to its effects on gene expression. In addition, it is not clear that inducing the CI deficiency at the larval stage (D+A KDs) does not have any effect on the development of the flies, or that it does not result in a mitochondrial disease-like phenotype, as it had been described previously by other laboratories studying constitutive KDs of the same two genes (see comment #7).

This manuscript contributes an important advancement in the knowledge of the organismal and metabolic consequences of mitochondrial respiratory chain deficiency at different life stages. It also resolves a long-standing controversy as to whether complex I deficiency induces either lifespan shortening or extension in *D. melanogaster*.

This work will be of interest for the mitochondrial biology/pathology field as well as for the ageing research community.

Expertise in mitochondrial respiratory chain biogenesis and (dys)function.

3. How much time do you estimate the authors will need to complete the suggested revisions:

Estimated time to Complete Revisions (Required)

(Decision Recommendation)

Less than 1 month

4. Review Commons values the work of reviewers and encourages them to get credit for their work. Select 'Yes' below to register your reviewing activity at Web of Science Reviewer Recognition Service (formerly Publons); note that the content of your review will not be visible on Web of Science.

Yes

Review #2

1. Evidence, reproducibility and clarity:

Evidence, reproducibility and clarity (Required)

This manuscript reports that inducing mitochondrial dysfunction during early development leads to severe reductions in survival and stress resistance in adulthood. By contrast, adult flies with similar levels of mitochondrial dysfunction (but with that dysfunction restricted only to adulthood) have an extended lifespan and enhanced stress resistance, despite up to a 75% reduction in CI activity.

After ruling out developmental defects as the cause, the researchers conducted molecular characterizations of both short-lived and long-lived flies in the context of CI dysfunction. The results show that CI-challenged flies have distinct transcriptomic and metabolomic responses, which overlap significantly with other models of CI dysfunction. This suggests that early mitochondrial dysfunction caused by CI depletion triggers a maladaptive response that significantly reduces survival.

The results of this study are intriguing, it is somewhat preliminary to draw definitive conclusions about mitochondrial function adaptation in the context of aging. A list of concerns are included below. Most of the major comments are in the realm of how the authors interpret their data. They are not necessarily requests for more experiments (though in some cases, additional data could help the authors to refine their conclusions).

Major points:

1. Perhaps the biggest piece of missing data is some sort of "D only" impairment of CI. Since the RU486-inducible GeneSwitch system can be turned on and off by presentation (or removal) of drug, then it should be possible to pinpoint if the observed effects are due to a critical time window - or if they are due to the combination of D+A CI inhibition.
2. Introduction, paragraph 3. The last segment of the introduction is not clear. If it exists, the authors should provide evidence how CI dysfunction at various developmental stages affects stress resistance.
3. Similarly, in the data discussion, the authors should comment on what might be happening adaptively in the adult stage despite CI loss, with supporting literature.
4. ND-75 vs. ND-18. Based on Figure 1, it seems like the ND-75 KD is a less severe manipulation. Yet ND-75 flies showed a significant decrease in NAD⁺:NADH ratio, while ND-18 did not. Is it possible that ND-75 loss could induce different mechanisms of NAD⁺:NADH regulation than ND-18 loss? What might be happening molecularly? This might be an issue to address in the discussion of the results.

Related: ND-18 and ND-75 are both Complex I subunits. However, when considering both knock downs, there are many differences in the levels of other metabolites. What could be some reasons for such differences? This is also discussion material.

5. The authors should discuss how metabolic plasticity determines maladaptive responses detrimental to organismal fitness.
6. It is difficult to understand if any mechanism is associated with changes in the metabolic and proteomic profile that determine lifespan and organismal fitness during development. Between A and D+A flies, the D+A flies showed altered metabolic or proteomic profiles. There are many

possibilities that could be consistent with this. One is that the animals require full complex I function during development or pupation - and those requirements are dispensable in the adult stage. That would predict that if the authors could somehow impair Complex I function early (but then restore it late - "D flies") that the animals would have just as many problems as the D+A flies. 7. The hypothesis for the aging process in the context of MCI is not clear. It came abruptly at the end of the Discussion section and needs to be expanded with supporting literature.

****Minor points:****

1. There are at least three instances of "data not shown." This is not allowed for many journals.
2. Figure 1F-H (and associated text): 29°C is a harsher environment for flies than 18°C. By switching from the GS system to the GAL80TS system for this experiment, it is possible that temperature-induced complications are being introduced. That would mean that the experiment is not a clean examination of developmental time windows.
3. Introduction, paragraph 2, line 1: The author should cite articles where CI function has been intensively studied in different model systems.
4. Materials and methods: The authors should briefly explain GeneSwitch (tubGS) and mifepristone (RU486) for readers who might not be familiar.
5. In the section titled, 'Developmental CI depletion increases stress sensitivity and alters metabolic homeostasis', paragraph 4 reads: "ND-75 KD flies had a slight but significant decrease..." It took a few readings to understand this sentence, so it might need editing.
6. The authors have discussed the proteome for the ND-18 KD flies. Did they conduct a similar experiment to test the ND-75 KD flies (and were the results similar)?
7. Figure S1: (A) Quantification of KD in ND-18 KD (left) and ND-75 KD (right) 5-7 day adult males. mRNA or protein quantification?
8. Figure 2 needs to be labeled correctly. C-F
9. The authors should denote the scale bar size for all the images.
10. Figure 2: The overall intensity of LipidTOX is reduced in D+A ND-75 KD. However, in the inset, it looks like it is saturated. Did the authors change intensity while processing the images? There should be a quantification panel of these images.
11. Figure S2: ND-18 KD flies either in A or in A+D condition did not show any change in NAD⁺ or NADH levels (this contrasts with ND-75 KD flies). Does this mean that ND-18 manipulation did not affect mitochondria and fitness in flies?

Similarly, there are quite a few differences in which genes are significantly altered in A vs. A+D when comparing the datasets from both ND-18 KD and ND-75 KD. Beforehand, one would assume a similar phenotype, as both ND-18 and ND-75 are components of MCI.

Likewise, ND-18 KD and ND-75 KD both disrupt MCI activity. However, looking at the MetaboAnalyst MS peaks, there is variability in the different biosynthetic pathways. It would be helpful to know the reason for such differences. The authors should elaborate on these points in the discussion section. It might provide valuable information to the readers.

12. Figure 3: The data presented in Figure 3 should mention A+D vs A in ND-18 and ND-75 in the

figure legend.

13. Figure S3: There seem to be some labeling errors in the legend. Specifically, F: GO enrichment analysis. G, H, I: Volcano plot of genes for the fat body, midgut, and heart. J, K: Clustergrams of the indicated GO terms expressing the differential expression (FC) of genes associated with this term.

2. Significance:

Significance (Required)

The results of this study are intriguing, and they could potentially close puzzling gaps in the literature about why some mitochondrial manipulations affect health and lifespan in one manner and others affect it in other ways. The specific knockdown experiments that the authors pursue with ND-75 and ND-18 offer possible mechanistic detail, though there are gaps in the approach. For this reviewer, one gap centered on pinpointing when in development the most profound effects of CI depletion happen (the critical window).

Another centered on actually understanding what is changed about the adult flies when comparing D+A vs. A only. The -omics approaches are a good way to start to get at this, but these are hypothesis generators, not specific tests themselves. It will be interesting in future studies to put some of those factors uncovered into specific tests.

This reviewer's broad areas of expertise are neuroscience and Drosophila genetics, including work on mitochondrial function.

3. How much time do you estimate the authors will need to complete the suggested revisions:

Estimated time to Complete Revisions (Required)

(Decision Recommendation)

Between 1 and 3 months

4. Review Commons values the work of reviewers and encourages them to get credit for their work. Select 'Yes' below to register your reviewing activity at Web of Science Reviewer Recognition Service (formerly Publons); note that the content of your review will not be visible on Web of Science.

Yes

Review #3

1. Evidence, reproducibility and clarity:

Evidence, reproducibility and clarity (Required)

In this study Stefanos et al show compelling evidence that the timing of mitochondrial dysfunction is a key determinant of the phenotype using a *Drosophila* model. By using inducible GeneSwitch models of ND-18 and ND-75 knockdown flies, they show that knockdown of either complex I subunit during development caused severe reduction in lifespan, while inducing knockdown after development in eclosed adult flies does not reduce lifespan. Developmentally induced ND-18 and ND-75 knockdown flies also show a reduction in complex I activity and moderate to severe change in lifespan on H₂O₂ supplemented food. Interestingly, transcriptional, metabolic and proteomic analyses between developmental and adulthood-induced knockdown show very different profiles in the genes, metabolites and proteins affected. These findings led the authors to hypothesise that the severe phenotypes caused by complex I inhibition during development are due to changes in plasticity occurring during development that are lost in adult flies.

I have the following comments:

1. In general, I would suggest including the control data shown in the supplemental together with the knockdown data in the main figures for all the experiments. It is very confusing to separate them.
2. Using round and square data points to label the D+A and A only conditions in the bar charts is very unclear. Using different patterns of colours of bars for the two conditions would be much clearer.
3. The lifespan data are very confusing. In the abstract and discussion it states that where mitochondrial dysfunction is perturbed in adulthood flies are long-lived. This suggests that ND-18 and ND-75 knockdown during adulthood causes flies to live longer than controls. However, it's not clear if this is the case. The control lifespan data in Figure S1E should be added to Figure 1C and statistical tests performed to compare the knockdown flies to controls. The third and fourth paragraphs of the Results are quite confusingly written, in part because the lifespan phenotypes are not compared to controls. This should be clarified.
4. The western blot data in Figure S1C should be quantified.
5. The second paragraph of the results states that the current data 'reconcile previous findings...'. Copeland et al observed a threshold effect for CG9172 (ND-20) and CG17856 (UQCR-14L), where moderate knockdown caused lifespan extension but strong knockdown caused developmental lethality. The current study therefore complements this work, rather than reconciling it.
6. Include data from Figure S1H in Figure 1D.
7. Eclosion analysis in Figure 1G should also be performed for ND-75 knockdown.
8. The developmental delay, negative geotaxis, fat body 'data not shown' data should be included.
9. The control data in Figure S2A should be included in Figure 2A. Same for figures S2B and 2B, S2C and 2C, S2D and 2D, S2E and 2E.
10. The lipid data in Figure 2G should be quantified.
11. When describing the transcriptomic, metabolomic and proteomic data, actual numbers should be described rather than writing 'many genes' etc. These should refer to supplemental tables showing lists of the genes, metabolites and proteins, which should be included.
12. The proteomic analysis should also be performed for ND-75 knockdown flies.
13. Were genes, metabolites or proteins indicating a switch to glycolytic metabolism upregulated in

the ND-18 or ND-75 knockdown flies? This would be interesting in the context of other CI deficiency models and patient studies showing increased glycolysis.

14. It would be interesting to know if the differentially expressed genes suggest the regulation by a transcription factor(s) known to regulate mitochondrial stress signalling, such as ATF4. Any evidence of this?

15. Are TCA cycle intermediates mis-regulated ND-18 or ND-75 knockdown flies?

16. How do the mis-regulated metabolites compare with those in other models of mitochondrial dysfunction such as serine, carnitine and 2-hydroxyglutarate (e.g. PMID: 27307216, PMID: 25681259, PMID: 31645461).

17. The terms 'adaptive response', 'plasticity', 'adaptive plasticity' and 'maladaptive' are used throughout the manuscript and are very confusing. Adaptive plasticity suggests a positive, beneficial effect. However, the data show that mitochondrial dysfunction causes negative phenotypes when it occurs during development. I would suggest only using the term 'maladaptive response' consistently throughout as this indicates a negative response to mitochondrial dysfunction during development.

18. There is very little actual discussion of the findings in the Results/Discussion section and the conclusion again confuses the idea of whether the study is focused on healthy lifespan extension (ageing), pathogenic mitochondrial dysfunction/disease, or both. It would be much better to include a full discussion section. Several interesting discussion points could be included. (1) Regarding mitochondrial disease, the relevance of early (childhood) onset versus adult-onset mitochondrial disease could be discussed. (2) Discussion of the omics changes in the context of other mitochondrial dysfunction/disease models. Are there any candidate molecules/pathways that might be contributing to the maladaptive response to developmental CI deficiency? If so, could these be targeted therapeutically. (3) Discussion of the potential lifespan extension caused by adult only CI inhibition in the context of other related Drosophila models e.g., Copeland 2009. (4) Regarding lifespan extension, there is a large literature in *C. elegans* on developmental timing, mitochondrial dysfunction and lifespan extension that should be discussed in light of the current findings. (5) Expand on the possible explanations for the finding that adult flies can function normally with an 80% reduction in CI activity. Are there other examples of this? How does this relate to adult-onset mitochondrial disease?

****Minor comments:****

1. Supplemental tables showing the transcriptomic, metabolomic and proteomic data should be included and these data should be deposited in publicly accessible databases (GEO etc).
2. Descriptions of what the charts represent (mean, SEM etc), sample numbers and details of statistical tests used should be included in all figure legends.
3. Details of scale bars should be included in the figure legends.
4. 'Figure S2H and I' on p.6 should be S2I and J.
5. 'Data' is plural, so do not write 'This data...'
6. Typo in title to Figure 3D.
7. The abbreviations used for different units changes throughout the methods section (e.g minutes/min/m; g/rcf). In the case of centrifugal speeds, in some cases rpm was used to describe the centrifugal spin. Please give the rcf, as rpm differs between centrifuges.

8. Include the dilutions of the antibodies using in the methods.
9. The manuscript should be read carefully to correct grammatical errors and typos.

2. Significance:

Significance (Required)

The study is well designed, well executed and the data are largely convincing. However, it generally not clear whether the study is focusing on the pathogenic consequences of mitochondrial dysfunction/disease, mitochondrial dysfunction in healthy ageing (lifespan), or both. The interpretations of the results are quite confusing in places, the figures could be much better presented, data not shown needs to be included, and a proper discussion section added to fully discuss the findings in the context of the literature. With these improvements, then the manuscript with would provide an important advance in the field and will be of interest to researchers in ageing and mitochondrial biology/disease.

I have expertise in mitochondrial biology/metabolism, Drosophila and mouse models.

3. How much time do you estimate the authors will need to complete the suggested revisions:

Estimated time to Complete Revisions (Required)

(Decision Recommendation)

Between 1 and 3 months

Yes

Full Revision

Manuscript number: RC-2023-02211R

Corresponding author(s): Sanz, Alberto

[Please use this template only if the submitted manuscript should be considered by the affiliate journal as a full revision in response to the points raised by the reviewers.]

*If you wish to submit a preliminary revision with a revision plan, please use our "Revision Plan" template. **It is important to use the appropriate template to clearly inform the editors of your intentions.**]*

1. General Statements [optional]

Dear Editor,

We want to express our gratitude for the time and effort you and the reviewers have dedicated to revising our work. We are encouraged by the positive feedback on our research and have made a significant effort to address all the comments and incorporate the reviewers' suggestions wherever feasible.

The revisions undertaken have substantially improved the manuscript. We hope that this updated version effectively addresses the reviewers' queries.

Enclosed below is a point-by-point response to the reviewers' comments.

Kind regards,

Alberto Sanz

Reviewer #1

The manuscript by Stefanatos et al. reports interesting observations regarding the differential effects of knocking down two subunits of mitochondrial complex I (CI) at two different points in the life of the fruit fly, i.e., during larval stage vs. adult age. The data contained in the manuscript clearly show that disrupting CI during *D. melanogaster* larval development produces a shortening of the lifespan, whereas a similar CI deficiency but induced in adulthood actually increases the lifespan of the flies, reconciling the data available in the literature. The work in this manuscript also points out to a different metabolic adaptation in the different stages of the fly as the cause of the age-related differential effects.

Full Revision

General comment: We concur with the reviewer's observation that our data demonstrate, for the first time, the significance of the timing of mitochondrial function disruption. This is as important, if not more so, than the intensity of the dysfunction. Furthermore, as noted by the reviewer, our data reconcile contradictory findings in the existing literature.

However, there are certain points in the manuscript that deserve clarification in order to improve its readability and consistency in the interpretation of the results.

Major points:

1. It is not clear what the controls are in each of the experiments, i.e., whether it is the genetic control (RNAi control but non induced) or the non-RNAi flies but treated with the inducing drug (RU486). This distinction is important because RU486 is well known to affect gene expression (including expression of genes involved mitochondrial function:

<https://www.nature.com/articles/s41514-019-0036-8>). For example, in figure S1 D, one would get the impression that the expression of ND-75 is increased 1.5-fold after RU486 administration in the adults.

Response 1: In our experiments, the control group consists of flies that lack the RNAi construct but possess the same genomic landing position, thereby serving as a control for the insertion of the RNAi transgene. Specifically, the D+A group comprises flies that were fed RU-486 during both their developmental and adult stages, while the A-only group comprises flies that received RU-486 only after eclosion. This experimental design applies to flies subjected to RNAi targeting either ND-75 or ND-18, as well as the previously mentioned control flies. It is important to emphasise that our experimental setup differs from the study cited, which observed differences between flies fed and those not fed with RU-486 during adulthood. In our system, all flies utilised in our experiments are administered RU-486 during adulthood. Accordingly, our comprehensive omics analyses indicate no discernible differences associated with RU-486 feeding during development in the control group, given that the concentrations of RU-486 employed in our experiments are within the nanomolar range.

The initial version did not correctly normalise Figure S1D to the levels of the control without RU-486, leading to an apparent increase compared to the panel above. However, this issue has been rectified in the revised version of the manuscript.

2. To better understand and interpret the results, it would be much better to include all the data regarding the controls (lifespan plots, activity graphs, transcript levels, histology and EM images, etc.) together with the KD data in the main figures, instead of showing them in separate supplementary figures.

Response 2: We respectfully disagree with the reviewer's suggestion. While all experimental groups share the w1118 genetic background, it's crucial to consider the genetic relatedness within these groups. Specifically, the RNAi-expressing groups are more genetically similar to each other, as they share the same parentage, than they are to the control groups, which have different parents. Similarly, the control groups have the same parents and are more similar among themselves than to the RNAi groups. Comparing groups with the same genetic background aligns with the standard practice when using the GeneSwitch system¹⁻⁷, designed initially to account for background effects.

In our study, the essence lies in revealing that flies with identical mitochondrial defects in adulthood exhibit significantly different lifespans, transcriptomes, proteomes, and metabolomes.

Full Revision

This can only be effectively demonstrated by directly comparing the D+A (short-lived) group with the A-only (long-lived) group, while controls, both treated and untreated with RU-486, confirm that the activation of the GeneSwitch (GS) system itself does not induce the observed phenotypes.

Nonetheless, we acknowledge the reviewer's interest in seeing data from control and experimental groups together. To address this, we have included a supplementary file where data from control and experimental groups are compared side by side for a more comprehensive perspective.

3. All relevant data mentioned in the manuscript, such as the nd rates, climbing assay results or the fat body findings, should be shown and not refer to them as "data not shown".

Response 1.3: We have included all the mentioned data as requested or remove them if they are not essential in supporting the main conclusions of the paper.

4. The extent of the defect in the CI-linked respiratory rates in Figure 1D seems to be comparable in the D+A and the A flies (around 25% for ND-18 and 50% for ND-75 KD), however the decrease in the mRNA levels is more profound in the D+A KDs (Figure S1A). This makes sense, as the duration of the KD induction is longer in the D+A than in the A only, but how can it be reconciled with the functional effects? Probably because of the lack of good antibodies for the target proteins, NDUFV2 protein levels were analysed by western blot and immunodetection (Figure S1C). However, they do not seem to reflect the findings at the mRNA level, because the amounts of NDUFV2 in the D+A KDs appears higher than in the A only KDs. This is in any case difficult to assess properly as there is no densitometric quantification data and the two types of KDs seem to have been run in different gels. It would be convenient to quantify this properly and to run Blue-Native gels to determine the amounts of fully assembled CI in each of the samples, instead of the steady-state levels of the subunit, which could be present but not assembled.

Response 1.4: The mRNA levels in Figure S1A do not show statistically significant differences between D+A and A-only groups, as indicated by the absence of asterisks denoting such a difference in the panel. Correspondingly, we have added the quantification of NDUFV2 levels, revealing no significant differences between the D+A and A-only groups (Figure S1C). Importantly, there is a strong and highly significant correlation between mRNA levels and respiration rates when all data points are collectively analysed, as shown in Rebuttal Figure 1. This suggests that mRNA depletion reliably predicts the impact on respiration. Additionally, our proteomic data in the ND-18 model indicate a significant depletion of ND-18 and most CI subunits compared to controls in both D+A and A-only groups, but no difference between the ND-18 D+A and A-only groups themselves. These data, already included in the paper, have been re-presented as heat maps instead of the original volcano plots for enhanced clarity (Figure S4H).

Considering all alternative CI measurements, our data conclusively demonstrate that the differences observed between short- and long-lived mitochondrially compromised flies are not due to variations in CI levels during adulthood. Therefore, we deem further semi-quantitative analyses, such as Blue-Native (BNE) gel electrophoresis, unnecessary for this study. Moreover, our findings regarding CI activity depletion, measured through high-resolution respirometry, show that a 50% reduction resulting from ND-75 knock-down during both development and

Full Revision

adulthood has a more significant impact on longevity than a 75% reduction caused by ND-18 knock-down exclusively in adulthood (Figure 1B & 1C). This underscores the notion that the timing of mitochondrial dysfunction plays a more pivotal role than the magnitude of dysfunction in determining lifespan outcomes.

5. The transcript and protein levels, as well as the respiratory activities in the adult flies were assessed at 3-5 days of age, do the authors have any idea if these change at longer times?

Response 1.5: While we have not systematically collected data for flies older than ten days, it's essential to consider the irreversible nature of RNAi in *Drosophila*, as it has been shown before^{8,9}. Given this irreversibility, we do not anticipate significant changes in older flies when compared to younger ones.

6. In the methods section: 'For L3 larvae the following substrates were added, 5mM Proline, 5mM Pyruvate, 2.5mM Glutamate, 2mM Malate. For adults 5mM Proline, 5mM Pyruvate was added'. Is there any particular reason why the authors use different substrates for analysing respiration at different developmental stages?

Response 1.6: The utilisation of different substrates for analysing respiration at different developmental stages stems from the inherent metabolic differences between larvae and adults, particularly in the context of mitochondrial function¹⁰. Through our experiments, we observed contrasting adult and larval mitochondria responses to various substrates. To ensure our data's reproducibility and enhance respiration rates in larval mitochondria, we introduced glutamate and malate as additional substrates. This adjustment was made based on empirical evidence that these substrates were necessary to achieve consistent results due to the dissimilarities in larval and adult mitochondrial metabolism.

7. The most critical point is the fact that two previous reports exist in the literature in which constitutive KDs of the same two CI genes have been done from larval stage, with similar findings in terms of reduction of CI activity and life span

(<https://academic.oup.com/hmg/article/23/17/4686/741708> and

<https://journals.biologists.com/dmm/article/11/3/dmm032482/53188/Feeding-difficulties-a-key-feature-of-the>). Both these papers claim that the reduction in life span is due to pathological findings, such as neurodegeneration or difficulties to feed. The authors should discuss their findings in the context of these previous papers and also discuss possible discrepancies between the previous reports and the present one.

Response 1.7: The studies referenced support our conclusions about the essential role of optimal mitochondrial function during development for a normal adult lifespan. These papers, which also conducted CI gene knock-downs from the larval stage, reported reduced CI activity and lifespan. They linked the reduction in lifespan to pathological findings like neurodegeneration and feeding difficulties. Our study aligns with these observations, showing that even minor CI depletion during development, such as the one in our study, significantly reduces lifespan. Our approach to CI disruption minimally impacts proper tissue development. This distinction sets our study apart from the aforementioned ones, underscoring the critical role of mitochondrial function even without gross tissue alterations. Besides, our work demonstrates an exceptional tolerance of flies to adult CI dysfunction, provided mitochondrial integrity is preserved during development. We appreciate the reviewer's suggestions and have incorporated this discussion into our manuscript.

Full Revision

8. Related to the previous point, in the images of the head section of the D+A ND-75 fly in Figure 2A, there seems to be pathological findings (unless for some reason that sample was damaged). Same applies to the EM images from the RNAi muscles, where there seems to be signs of vacuolisation.

Response 1.8: Upon close inspection of all images displayed in panels 2A & B and S2A & B, we identified technical artefacts that could be misconstrued as neurological lesions or mitochondrial structural alterations. These artefacts are present in both the experimental and control groups. It is important to emphasise that there are no observable differences between the D+A and A-only groups, nor between the KD groups and the controls. However, we have decided to remove these images from the manuscript, as they do not meet the quality standards upheld by the other figures presented.

9. Again, it is not clear what kind of control is used in the -omics analyses, schematised in Figures S3A and S4A (see point #1) and how the genes that are differentially expressed due to the RU486 treatment are filtered out using a customised bioinformatics pipeline. This could introduce some bias in the analysis and could have been determined experimentally by performing an analysis comparing RU486-treated vs. untreated flies.

Response 1.9: In all the -omics analyses conducted, we use the same four groups: (1) Control A (empty control without RU-486 during development), (2) Control D+A (empty control fed RU-486), (3) ND-18 KD or ND-75 KD A-only (RNAi against ND-18 or ND-75 without RU-486 during development) and (4) ND-18 or ND-75 D+A (RNAi against ND-18 or ND-75 fed with RU-486 during development).

To address any potential bias, we implemented a rigorous analysis strategy. Initially, we performed principal component analysis (PCA) and generated volcano plots comparing D+A versus A-only while distinguishing between controls and RNAi lines. This approach was employed to eliminate any background effects from the analysis. Subsequently, in the downstream analysis, which includes volcano plots and the identification of pathway/enrichment genes, metabolites, or proteins, we removed those elements that exhibited statistically significant differences in the controls when comparing D+A versus A-only. This step was taken to control for any unintended effects arising from the presence of RU-486 in the fly food during development. It is worth noting that all adult groups were subjected to RU-486 treatment during adulthood, which discards any unwanted effect of RU-486 during this period.

Notably, the comparison of controls (D+A versus A-only) revealed only minor differences in a few genes and proteins. Interestingly, although there were variations in the concentration of different metabolites (represented by distinct peaks in the metabolomics analysis) between controls (D+A vs. A-only), these samples did not cluster differently in the PCA analysis (Figure S4C). In contrast, the RNAi lines (both ND-18 and ND-75) exhibited clear distinctions in the PCA analysis, underscoring the specificity of the RNAi-induced effects (Figure S4D).

10. It is not clear why the authors carried out the metabolomics analyses in the whole fly instead of focusing on the tissues where they see the transcriptomic signature.

Response 1.10: The decision to conduct metabolomics analyses on whole flies, rather than focusing on specific tissues where the transcriptomic signature is observed, was made with careful consideration. We chose this approach primarily to minimise disruptions to the

Full Revision

metabolome, as, e.g., dissecting the adult fly fat body can be highly invasive and time-consuming.

Although using whole flies may result in a loss of resolution compared to analysing specific tissues, it allows us to preserve a larger portion of the metabolome, closely resembling the *in vivo* conditions. Therefore, we deemed gaining a comprehensive view of the fly metabolome in this initial study advantageous. It's important to note that changes in the transcriptome and metabolome do not always exhibit a direct correlation, and by analysing whole flies, we aimed to capture a more holistic understanding of metabolic changes. We consider this approach important as the present study represents the first in-depth one addressing the developmental and adult effects of CI dysfunction.

11. In the description of the findings in Figure 3, the authors state "ND-75 KD flies showed a greater response than ND-18 KD flies." But a few sentences later the text reads: "...the response to CI dysfunction in both models overlaps highly." However, in the results shown previously in the manuscript, ND-75 KD flies show a significantly less severe CI defect and less effect on lifespan. Please, discuss these apparently contradictory points.

Response 1.11: Figures 3 and 4, alongside their supplementary data, demonstrate significant overlaps in transcriptomic and metabolomic responses between ND-75 KD and ND-18 KD models. This is particularly highlighted in the GO and KEGG analyses, which pinpoint identical pathways in both models of CI dysfunction. Over 50% of genes altered in the ND-75 KD model change in the same direction in the ND-18 KD model, with minimal instances of opposing trends, as delineated in Figure 3. This underscores a unified response to mitochondrial CI dysfunction.

The disparity between lifespan variations in the ND-18 KD model and pronounced transcriptomic and metabolomic differences in the ND-75 KD model is indeed striking. A plausible explanation lies in the differential impact of ND-18 and ND-75 knock-downs on respiration during larval stages (27% vs. 19% depletion, respectively) and its subsequent effect on lifespan. This suggests a significant role of CI activity in developmental stages in shaping adult lifespan, highlighting a correlation between CI activity during development and adult lifespan.

The observation that ND-75 depletion, with less pronounced effects on CI respiration and lifespan, elicits a more substantial transcriptional and metabolic response compared to ND-18 remains to be fully elucidated. Our current hypothesis, pending further investigation, suggests that higher depletion in CI activity during development might limit transcriptional and metabolic adaptability. While some developmental responses may contribute to reduced lifespan, others could be adaptive, enhancing survival in conditions like the observed fat body atrophy. Additionally, distinct roles of ND-75 and ND-18 beyond CI assembly and function must be considered. For instance, ND-75's involvement in cytosolic apoptosis induction¹¹ or ACAD9's dual role in fatty acid oxidation and CI biogenesis¹² indicate that their absence during development may trigger specific transcriptomic and metabolomic adaptations. This complexity adds a fascinating dimension to CI dysfunction studies, although a thorough examination of these adaptations falls outside this work's current scope. In this manuscript, we acknowledge the differences between the two KD models; however, our focus is primarily on the similarities observed between both ND-18 and ND-75 KD flies. We posit that the genes, proteins, and

Full Revision

metabolites consistently altered in D+A flies across both models provide critical insights into the pathological mechanisms underlying CI dysfunction.

12. There is a lack of depth in the analysis and description of the results of the -omics experiments at the end of the paper. Apart from the generic gene ontology and principal component analysis plots, it would be convenient to provide specific information about which are the actual transcripts, metabolites and proteins that change the most, and if the data from the transcriptomics, proteomics and metabolomics actually fit well with each other. The name of the most significant hits should be shown in the volcano plots shown in the figures, helping the reader to interpret and understand the final point of this report in a quicker way (it remains rather obscure in the current version of the manuscript).

Response 1.12: The information extracted from omics data is extensive, and its comprehensive analysis and interpretation are ongoing processes without a definite end. Nevertheless, we have taken steps to address the reviewer's concerns. Firstly, we have uploaded the raw data to the appropriate repositories (as outlined in the Materials and Methods section). Secondly, we have included an Excel file containing our analyses, allowing other researchers to explore and analyse the data using alternative approaches.

Thirdly, in response to the reviewer's valuable feedback, we have implemented the suggested modifications to enhance the manuscript's clarity. Specifically, we have added essential information to the volcano plots, including the names of the most significant hits, which aids in facilitating a quicker and more straightforward interpretation of the results. Moreover, we have conducted a more in-depth analysis to elucidate the relationship between the transcriptomics, proteomics, and metabolomics changes observed, providing a more comprehensive understanding of how these datasets align with each other.

Minor point:

1. Figures S2, S3 and S4 are too large.

Response 1.13: We acknowledge the reviewer's observation that Figures S2, S3, and S4 contain extensive information, and we have removed the images displayed in S2A & B. Nevertheless, the rest of the information is crucial to comprehensively understanding this study's novel concepts and results.

Reviewer #1 (Significance (Required)):

The strength of the work shown in this manuscript is that it clearly shows the differential effect of mitochondrial complex I deficiency in *D. melanogaster* depending on whether the deficiency occurs during larval development or after reaching the adult state, producing either lifespan shortening or expansion, respectively. The manuscript also demonstrates that the transcriptional and/or metabolic adaptive responses to complex I deficiency are common to the lack of two different subunits, but are different depending on whether the deficiency happened early or late in life.

One of the main limitations is the use of RU486 as an inducer of the KDs, which could give confounding results due to its effects on gene expression. In addition, it is not clear that inducing the CI deficiency at the larval stage (D+A KDs) does not have any effect on the development of the flies, or that it does not result in a mitochondrial disease-like phenotype, as it had been described previously by other laboratories studying constitutive KDs of the same two genes (see comment #7).

Full Revision

This manuscript contributes an important advancement in the knowledge of the organismal and metabolic consequences of mitochondrial respiratory chain deficiency at different life stages. It also resolves a longstanding controversy as to whether complex I deficiency induces either lifespan shortening or extension in *D. melanogaster*.

This work will be of interest for the mitochondrial biology/pathology field as well as for the ageing research community.

Expertise in mitochondrial respiratory chain biogenesis and (dys)function.

Summary: We appreciate the reviewer's positive assessment of our work. We acknowledge the relevance of our findings, which demonstrate that the timing of mitochondrial dysfunction is more critical for its impact on lifespan than the degree of dysfunction per se. Since all adult groups in our study were treated with RU-486, and the transcriptomic/metabolomic/proteomic analyses were conducted in adults, the potential effects of RU-486 are expected to be consistent across all groups, and this does not significantly affect the conclusions of our study. Furthermore, we agree with the reviewer's observation that the short-lived flies (where mitochondrial dysfunction is induced during development) serve as a model of mitochondrial disease, as indicated by previous research. We have emphasised this aspect more clearly in the revised manuscript. As a result, our work highlights the importance of minimising and controlling developmental effects when studying the impact of mitochondria on ageing and age-related diseases.

Reviewer #2 (Evidence, reproducibility and clarity (Required)):

This manuscript reports that inducing mitochondrial dysfunction during early development leads to severe reductions in survival and stress resistance in adulthood. By contrast, adult flies with similar levels of mitochondrial dysfunction (but with that dysfunction restricted only to adulthood) have an extended lifespan and enhanced stress resistance, despite up to a 75% reduction in CI activity.

After ruling out developmental defects as the cause, the researchers conducted molecular characterisations of both short-lived and long-lived flies in the context of CI dysfunction. The results show that CI-challenged flies have distinct transcriptomic and metabolomic responses, which overlap significantly with other models of CI dysfunction. This suggests that early mitochondrial dysfunction caused by CI depletion triggers a maladaptive response that significantly reduces survival.

The results of this study are intriguing, it is somewhat preliminary to draw definitive conclusions about mitochondrial function adaptation in the context of ageing. A list of concerns are included below. Most of the major comments are in the realm of how the authors interpret their data.

They are not necessarily requests for more experiments (though in some cases, additional data could help the authors to refine their conclusions).

Summary: We appreciate the positive feedback from the reviewer. While we acknowledge that our study alone may not be sufficient to draw definitive conclusions about the role of mitochondrial function in regulating ageing, it does emphasise the importance of controlling for developmental effects when investigating the role of mitochondria in ageing. Furthermore, our findings suggest that therapies targeting mitochondrial diseases may be more effective if implemented early during development. We believe that the publication of these results marks the first step in reshaping how mitochondrial research on ageing and diseases is conducted.

Full Revision

This includes replicating and expanding our findings independently in fly models and exploring the translatability of our conclusions to mammals.

Major points:

1. Perhaps the biggest piece of missing data is some sort of "D only" impairment of CI. Since the RU486-inducible GeneSwitch system can be turned on and off by presentation (or removal) of drug, then it should be possible to pinpoint if the observed effects are due to a critical time window - or if they are due to the combination of D+A CI inhibition.

Response 2.1: We concur with the reviewer's suggestion that an experiment designed to specifically induce mitochondrial dysfunction during the developmental phase and restore mitochondrial function in adults would be highly informative. However, we have encountered significant challenges in creating such a model. These challenges stem from the GS system's inherent leakiness and RNAi's irreversible nature in fruit flies, leading to substantial depletion of the target gene even in the absence of RU-486 induction, as both we and others have demonstrated previously^{8,9,13}. Although we can regulate the levels of a transgene such as GFP using GS -where exposure to RU-486 increases its levels, and removing RU-486 from the fly food returns these levels to baseline—our efforts to counteract CI depletion by expressing a human orthologue of the target fly gene have been unsuccessful so far. We suspect that the limited similarity between fly and human proteins may hinder the stabilisation of CI assembly, which relies on multiple proteins' interactions.

We are currently developing constructs in which the endogenous fly gene incorporates silent mutations to render it resistant to RNAi while enabling the expression of a fully functional protein. Nevertheless, creating and validating these new fly models is expected to consume considerable time. Furthermore, the new models will likely yield substantial data, potentially meriting a separate publication.

Given these challenges and the importance of our current results, we contend that it is imperative to publish our findings promptly.

2. Introduction, paragraph 3. The last segment of the introduction is not clear. If it exists, the authors should provide evidence how CI dysfunction at various developmental stages affects stress resistance.

Response 2.2. In response to the reviewer's feedback, we have revised the referred part of the introduction and the results for enhanced clarity. The revised text now explicitly references the experiments depicted in Figure 2A-C. These experiments demonstrate a clear correlation: mitochondrial CI dysfunction during developmental stages leads to diminished resistance to oxidative stress, starvation, and thermal stress. This provides empirical evidence supporting our discussion on the impact of CI dysfunction at various developmental stages.

3. Similarly, in the data discussion, the authors should comment on what might be happening adaptively in the adult stage despite CI loss, with supporting literature.

Response 2.3: As per the reviewer's request, we have incorporated a paragraph into the discussion section.

4. ND-75 vs. ND-18. Based on Figure 1, it seems like the ND-75 KD is a less severe manipulation. Yet ND-75 flies showed a significant decrease in NAD⁺:NADH ratio, while ND-18 did not. Is it possible that ND-75 loss could induce different mechanisms of NAD⁺:NADH

Full Revision

regulation than ND-18 loss? What might be happening molecularly? This might be an issue to address in the discussion of the results.

Response 2.4: As previously discussed in response 1.11, the knock-down of ND-75 has a more pronounced impact on the transcriptome and metabolome than the knock-down of ND-18.

Despite this, the latter exhibits a more significant effect on lifespan. However, changes in NADH and NAD⁺ levels in relation to the controls are similar, and the apparent confusion arises from the presentation of the data. In both ND-75 and ND-18 models, there is a significant reduction in the NAD⁺/NADH ratio when comparing RNAi lines with control groups, as depicted in Rebuttal Figure 2. Specifically, in the case of ND-75, additional differences are noted between the D+A and A-only groups. However, these differences are relatively minor, especially when juxtaposed with control groups. Crucially, results across respirometry, Western blotting, and proteomics consistently indicate that both models strongly deplete CI relative to controls. This depletion accounts for the shared transcriptomic and metabolomic signatures observed in both models.

Related: ND-18 and ND-75 are both Complex I subunits. However, when considering both knock downs, there are many differences in the levels of other metabolites. What could be some reasons for such differences? This is also discussion material.

Response 2.5 In response 1.11, we addressed the differences in the transcriptomic and metabolomic responses between ND-18 and ND-75.

5. The authors should discuss how metabolic plasticity determines maladaptive responses detrimental to organismal fitness.

Response 2.5. We have done this in the manuscript.

6. It is difficult to understand if any mechanism is associated with changes in the metabolic and proteomic profile that determine lifespan and organismal fitness during development. Between A and D+A flies, the D+A flies showed altered metabolic or proteomic profiles. There are many possibilities that could be consistent with this. One is that the animals require full complex I function during development or pupation - and those requirements are dispensable in the adult stage. That would predict that if the authors could somehow impair Complex I function early (but then restore it late - "D flies") that the animals would have just as many problems as the D+A flies.

Response 2.6: This issue has been previously addressed, outlining the technical challenges of conducting the suggested experiments. Nevertheless, it is imperative to highlight that the outcomes of such an approach, while potentially shedding light on the reversibility of certain effects, would not modify the two fundamental conclusions of our manuscript: 1) CI is crucial during development, and even minimal disruptions can lead to a marked reduction in adult lifespan; 2) Adult flies demonstrate a significant tolerance to mitochondrial dysfunction, evidenced by their lifespan remaining unaffected despite up to a 75% decrease in CI activity.

7. The hypothesis for the aging process in the context of MCI is not clear. It came abruptly at the end of the Discussion section and needs to be expanded with supporting literature.

Response 2.7 The text has been revised as suggested by the reviewer.

Minor points:

1. There are at least three instances of "data not shown." This is not allowed for many journals.

Full Revision

Response 2.8 We have incorporated the relevant data into the manuscript and omitted references to "data not shown" in instances where it is not crucial to the study, considering the extensive amount of data already presented.

2. Figure 1F-H (and associated text): 29°C is a harsher environment for flies than 18°C. By switching from the GS system to the GAL80TS system for this experiment, it is possible that temperature-induced complications are being introduced. That would mean that the experiment is not a clean examination of developmental time windows.

Response 2.9: We acknowledge the reviewer's concerns regarding the potential impact of temperature on our experiments using the GAL80ts system. While 29°C presents a more challenging environment for flies than 18°C, this is currently the only feasible method for our experimental design. Alternative inducible systems, such as those reliant on feeding compounds like RU-486, are not viable, as they do not permit exclusive feeding to pupae.

3. Introduction, paragraph 2, line 1: The author should cite articles where CI function has been intensively studied in different model systems.

Response 2.10: We have included appropriate references in the introduction, citing studies that study the effects of CI dysfunction in different model organisms.

4. Materials and methods: The authors should briefly explain GeneSwitch (tubGS) and mifepristone (RU486) for readers who might not be familiar.

Response 2.11: We have added a concise explanation of the GeneSwitch system in the materials and methods section.

5. In the section titled, 'Developmental CI depletion increases stress sensitivity and alters metabolic homeostasis', paragraph 4 reads: "ND-75 KD flies had a slight but significant decrease..." It took a few readings to understand this sentence, so it might need editing.

Response 2.12. We have revised the sentence as requested.

6. The authors have discussed the proteome for the ND-18 KD flies. Did they conduct a similar experiment to test the ND-75 KD flies (and were the results similar)?

We did not conduct the proteomics experiment on ND-75 KD flies. Our initial focus was on ND-18 KD flies, where we observed proteomic changes consistent with the transcriptomic and metabolomic data. Moreover, proteomic analysis reveals that RNAi targeting ND-18 caused more than 50% depletion in ND-75 (see Rebuttal Figure 3). Given the similarity in the main transcriptomic and metabolomic changes between both fly models, we decided it was not necessary to allocate further time and resources to ND-75 proteomics. Therefore, we have clarified in the text that proteomic analysis was performed solely on the ND-18 model.

7. Figure S1: (A) Quantification of KD in ND-18 KD (left) and ND-75 KD (right) 5-7 day adult males. mRNA or protein quantification?

Response 2.14. The figure shows mRNA quantification, and this has now been clearly indicated in the figure legend.

8. Figure 2 needs to be labelled correctly. C-F

Response 2.15. The labelling of Figure 2 has been corrected.

9. The authors should denote the scale bar size for all the images.

Response 2.16. We have added scale bar sizes to all images as requested.

Full Revision

10. Figure 2: The overall intensity of LipidTOX is reduced in D+A ND-75 KD. However, in the inset, it looks like it is saturated. Did the authors change intensity while processing the images? There should be a quantification panel of these images.

Response 2.17. All images were processed using consistent settings. We have included the requested quantification.

11. Figure S2: ND-18 KD flies either in A or in A+D condition did not show any change in NAD⁺ or NADH levels (this contrasts with ND-75 KD flies). Does this mean that ND-18 manipulation did not affect mitochondria and fitness in flies?

Similarly, there are quite a few differences in which genes are significantly altered in A vs. A+D when comparing the datasets from both ND-18 KD and ND-75 KD. Beforehand, one would assume a similar phenotype, as both ND-18 and ND-75 are components of MCI.

Likewise, ND-18 KD and ND-75 KD both disrupt MCI activity. However, looking at the MetaboAnalyst MS peaks, there is variability in the different biosynthetic pathways. It would be helpful to know the reason for such differences. The authors should elaborate on these points in the discussion section. It might provide valuable information to the readers.

Response 2.18: We have previously addressed similar comments in responses 1.1, 2.4 and 2.5.

12. Figure 3: The data presented in Figure 3 should mention A+D vs A in ND-18 and ND-75 in the figure legend.

Response 2.19. We have amended the figure legend as suggested.

13. Figure S3: There seem to be some labeling errors in the legend. Specifically, F: GO enrichment analysis. G, H, I: Volcano plot of genes for the fat body, midgut, and heart. J, K: Clustergrams of the indicated GO terms expressing the differential expression (FC) of genes associated with this term.

Response 2.20. The labelling errors in the legend have been rectified. We appreciate the reviewer for pointing them out.

Reviewer #2 (Significance (Required)):

The results of this study are intriguing, and they could potentially close puzzling gaps in the literature about why some mitochondrial manipulations affect health and lifespan in one manner and others affect it in other ways. The specific knock-down experiments that the authors pursue with ND-75 and ND-18 offer possible mechanistic detail, though there are gaps in the approach. For this reviewer, one gap centered on pinpointing when in development the most profound effects of CI depletion happen (the critical window).

Another centered on actually understanding what is changed about the adult flies when comparing D+A vs. A only. The -omics approaches are a good way to start to get at this, but these are hypothesis generators, not specific tests themselves. It will be interesting in future studies to put some of those factors uncovered into specific tests.

This reviewer's broad areas of expertise are neuroscience and *Drosophila* genetics, including work on mitochondrial function.

Summary: We agree with the reviewer's evaluation that our study significantly contributes to understanding the dual role of mitochondria in lifespan regulation, a longstanding question in the field of ageing. For the first time, this paper demonstrates both lifespan extension and reduction in two different models of CI dysfunction in the same genetic background, showing no differences in mitochondrial activity during adulthood between short- and long-lived individuals.

Full Revision

This was accomplished using comprehensive methods such as high-resolution respirometry, Western blotting, and TMT-proteomics, in contrast to previous assessments of CI dysfunction, which primarily focused on measuring the mRNA levels of the target subunit.

Our results underscore the essential role of fully functional mitochondria during the larva-to-pupa transition in fruit flies. Owing to technical limitations, these experiments could not employ systems based on feeding specific compounds for RNAi activation, such as RU-486, since feeding halts during this transition phase. Consequently, the use of GAL80ts was deemed the only feasible approach.

We also concur that our study, with its extensive omics data, is set to open new avenues in mitochondrial and ageing research. While our findings pose new questions for future exploration by our team and others with diverse expertise, we believe that publishing our study in its current form is pivotal for advancing the understanding of mitochondria's role in health and disease. Emphasising the timing of mitochondrial dysfunction as a crucial factor is a key step forward.

Reviewer #3 (Evidence, reproducibility and clarity (Required)):

In this study Stefanos et al show compelling evidence that the timing of mitochondrial dysfunction is a key determinant of the phenotype using a *Drosophila* model. By using inducible GeneSwitch models of ND-18 and ND-75 knock-down flies, they show that knock-down of either complex I subunit during development caused severe reduction in lifespan, while inducing knock-down after development in eclosed adult flies does not reduce lifespan. Developmentally induced ND-18 and ND-75 knock-down flies also show a reduction in complex I activity and moderate to severe change in lifespan on H₂O₂ supplemented food. Interestingly, transcriptional, metabolic and proteomic analyses between developmental and adulthood-induced knock-down show very different profiles in the genes, metabolites and proteins affected. These findings led the authors to hypothesise that the severe phenotypes caused by complex I inhibition during development are due to changes in plasticity occurring during development that are lost in adult flies.

We concur with the reviewer's observation that our study presents "compelling evidence" demonstrating that the timing of mitochondrial dysfunction is crucial in determining adult survival. Furthermore, our data show that even minor disruptions of mitochondrial function during development can lead to significant metabolic, proteomic, and transcriptomic alterations. These findings contribute to understanding the devastating phenotypes observed in patients with mitochondrial diseases. Below, we address each of the reviewer's comments individually.

I have the following comments:

1. In general, I would suggest including the control data shown in the supplemental together with the knock-down data in the main figures for all the experiments. It is very confusing to separate them.

Response 3.1: As previously mentioned, we compare data within the same genetic background wherever possible, which is the standard approach when employing the GeneSwitch system^{1,2,7,14-17}. Controls (that do not express the RNAi) are essential to demonstrate that RU-486 is not the cause of observed differences, and they provide a benchmark for the extent of mitochondrial activity depletion and its impact on lifespan compared to flies with normal mitochondrial function. To address the concern raised, we have now added a summary Excel file, where both control and RNAi group data are presented for easier comparison and clarity.

Full Revision

2. Using round and square data points to label the D+A and A only conditions in the bar charts is very unclear. Using different patterns of colours of bars for the two conditions would be much clearer.

Response 3.2. Following the reviewer's suggestion, we have updated the colour scheme in our bar charts to represent the D+A and A conditions distinctly.

3. The lifespan data are very confusing. In the abstract and discussion it states that where mitochondrial dysfunction is perturbed in adulthood flies are long-lived. This suggests that ND-18 and ND-75 knock-down during adulthood causes flies to live longer than controls. However, it's not clear if this is the case. The control lifespan data in Figure S1E should be added to Figure 1C and statistical tests performed to compare the knock-down flies to controls. The third and fourth paragraphs of the Results are quite confusingly written, in part because the lifespan phenotypes are not compared to controls. This should be clarified.

Response 3.3. The terms 'short-lived' and 'long-lived' are used in relative terms to compare different fly groups: short-lived refers to flies with compromised mitochondrial function during both development and adulthood (D+A), while long-lived denotes those where mitochondrial dysfunction occurs only in adulthood (A-only). It's important to note that the 'long-lived' flies do indeed live longer than the control group. However, the emphasis in our study is on the relative differences between the experimental groups rather than the comparison with controls, considering the similar backgrounds (w1118) between control and RNAi flies, but the absence of an extensive backcrossing process. We have clarified this point in the text and included comparisons between all groups in the aforementioned supplemental file for comprehensive understanding.

4. The western blot data in Figure S1C should be quantified.

Response 3.4. We have quantified the blots and added a panel to the supplementary figure.

5. The second paragraph of the results states that the current data 'reconcile previous findings...'. Copeland et al observed a threshold effect for CG9172 (ND-20) and CG17856 (UQCR-14L), where moderate knock-down caused lifespan extension but strong knock-down caused developmental lethality. The current study therefore complements this work, rather than reconciling it.

Response 3.5: The study by Copeland et al. was a landmark contribution to our understanding of mitochondria's role in ageing. However, it's crucial to note that their research did not comprehensively quantify CI activity or overall mitochondrial function in either adults or larvae, as BNE was the primary method used. Furthermore, the abstract of their study states that "long-lived flies with reduced expression of electron transport chain (ETC) genes did not consistently show reduced assembly of respiratory complexes or decreased ATP levels." As such, lifespan differences observed in their study cannot be directly attributed to variations in CI levels during development or adulthood. Moreover, the specific timing of mitochondrial function disruption was not discussed at all in their manuscript.

In contrast, our study replicates both the short- and long-lived phenotypes documented in previous literature, utilising the same CI subunit and genetic background. This methodology is vital to robustly assess whether the timing of mitochondrial dysfunction is critical. Additionally, using different subunits that cause various extents of adult CI depletion, our study indicates that the degree of CI reduction during adulthood is less significant in determining adult lifespan than

Full Revision

the timing of the dysfunction. A notable finding of our research is a fly model where a 75% reduction in CI activity does not shorten lifespan. This unique aspect sets our work apart from previous publications and introduces a novel perspective in understanding mitochondrial dysfunction in ageing.

6. Include data from Figure S1H in Figure 1D.

Response 3.6: Data encompassing both controls and RNAi lines, as referenced in Figure S1H, have been incorporated into a supplementary Excel file to facilitate easier comparison.

7. Eclosion analysis in Figure 1G should also be performed for ND-75 knock-down.

Response 3.7: We have performed the experiment and added the results to the new version of the manuscript as requested by the reviewer.

8. The developmental delay, negative geotaxis, fat body 'data not shown' data should be included.

Response 3.8. We have addressed this in response 1.3.

9. The control data in Figure S2A should be included in Figure 2A. Same for figures S2B and 2B, S2C and 2C, S2D and 2D, S2E and 2E.

Response 3.9: As previously stated, we have created an Excel file that includes all data, combining controls and RNAi lines.

10. The lipid data in Figure 2G should be quantified.

Response 3.10. We have added the requested quantification.

11. When describing the transcriptomic, metabolomic and proteomic data, actual numbers should be described rather than writing 'many genes' etc. These should refer to supplemental tables showing lists of the genes, metabolites and proteins, which should be included.

Response. 3.11: We have implemented the recommended changes in the manuscript, providing specific numbers for the transcriptomic, metabolomic, and proteomic data rather than using general terms like 'many genes'. Additionally, we have included a supplementary Excel file listing the genes, metabolites, and proteins for detailed reference.

12. The proteomic analysis should also be performed for ND-75 knock-down flies.

Response 3.12: Since similar transcriptomic and metabolomic responses were observed in both models, proteomic analysis was conducted solely on the ND-18 knock-down model. Given that the knock-down of ND-18 leads to a substantial decrease in ND-75 protein levels (by more than 50%) (Rebuttal Figure 3), it is expected that the proteomic rearrangements in response to ND-75 knock-down will align with those observed in the ND-18 model, as is the case with the transcriptomic and metabolomic profiles.

13. Were genes, metabolites or proteins indicating a switch to glycolytic metabolism upregulated in the ND-18 or ND-75 knock-down flies? This would be interesting in the context of other CI deficiency models and patient studies showing increased glycolysis.

Response 3.13: In this work, we focus on understanding how the timing of mitochondrial dysfunction influences the organism's response. We demonstrate that a moderate reduction in the activity of CI during development significantly shortens adult survival, whereas adult flies are remarkably resilient to substantial reductions in CI levels. Notably, short- and long-lived individuals with mitochondrial compromise exhibit marked differences in their transcriptomes, proteomes, and metabolomes. One or more of these identified differences could explain the observed disparity in lifespan. Conversely, if specific genes, metabolites, or pathways do not

Full Revision

differ between short- and long-lived individuals, it is very unlikely that they can explain the aforementioned differences. For example, our data did not reveal any consistent changes in the expression levels of glycolytic genes or metabolites between short- and long-lived CI-depleted flies. The sole exception was Hexokinase C (Hex-C), which exhibited reduced expression at both mRNA and protein levels (Rebutal Figure 4). Interestingly, we observed a clear reduction in the levels of many glycolytic intermediates; however, these striking differences were observed only in the ND-75 model, leading us to conclude that alterations in glycolysis are not required for lifespan shortening associated with CI.

14. It would be interesting to know if the differentially expressed genes suggest the regulation by a transcription factor(s) known to regulate mitochondrial stress signalling, such as ATF4. Any evidence of this?

Response 3.14. We utilised the TF2DNA database (https://www.fiserlab.org/tf2dna_db/) to identify genes regulated by the *Drosophila* orthologue of ATF4, namely *crc*. This analysis yielded 405 genes. Within this cohort, only two genes (~0.5%: one upregulated and one downregulated) exhibited significant differential expression in the comparison between short- and long-lived ND-18 flies, while 21 genes (~5%: 16 downregulated and 5 upregulated) showed differential expression in ND-75 flies under the same comparison.

Additionally, we evaluated a list of *Drosophila* homologues of mammalian ATF4 target genes reported to exhibit differential expression in a neuronal-specific model of CI dysfunction targeting ND-75 with RNAi¹⁸. No significant up- or downregulation was observed when comparing short- and long-lived ND-18 flies, with only two genes showing downregulation in ND-75 flies.

In summary, our findings do not suggest a significant role for the ATF4-mediated UPR in our *Drosophila* models of CI dysfunction. Across both ND-18 and ND-75 flies, the expression of only four genes was significantly altered, with *CG43175* and *Pepck2* being upregulated and *CG34040* and *Muc30E* downregulated. The discrepancies observed compared to the findings of Granat et al.¹⁸ could potentially be attributed to differences in experimental methodologies, including the neuronal-specific knock-down in ND-75, the developmental stage and intensity at which the knock-down was initiated, and the utilisation of heads for transcriptomic analysis as opposed to whole flies in our study.

15. Are TCA cycle intermediates mis-regulated ND-18 or ND-75 knock-down flies?

Response 3.15. While we observed changes in two metabolites associated with the TCA cycle, these alterations were confined to the ND-75 model and not replicated in ND-18 flies. Specifically, ND-75 short-lived individuals displayed increased malate and fumarate concentrations compared to long-lived ND-75 individuals. No statistically significant changes were detected in any TCA cycle intermediates within the ND-18 knock-down models. This finding suggests that modifications to the TCA cycle are not necessarily prerequisites for lifespan variations associated with altered CI levels.

16. How do the mis-regulated metabolites compare with those in other models of mitochondrial dysfunction such as serine, carnitine and 2-hydroxyglutarate (e.g. PMID: 27307216, PMID: 25681259, PMID: 31645461).

Response 3.16: Serine levels were upregulated, while 2-hydroxyglutarate levels were downregulated in short-lived ND-75 compared to long-lived ND-75. However, no differences

Full Revision

were observed in the ND-18 model. L-carnitine levels decreased in both models, as indicated in Figures 4A and 4B.

Therefore, our data only support a potential pathological role for carnitine in CI deficiency when analysing whole fly homogenates. However, we cannot rule out the possibility that other metabolites mentioned by the reviewer, such as serine or 2-hydroxyglutarate, may also be significant. Differences in these metabolites could become apparent when analyses are conducted on dissected tissues. Accordingly, metabolomics analyses performed exclusively on fly brains^{18,19} have highlighted a significant pathological role for 2-hydroxyglutarate in CI deficiency.

17. The terms 'adaptive response', 'plasticity', 'adaptive plasticity' and 'maladaptive' are used throughout the manuscript and are very confusing. Adaptive plasticity suggests a positive, beneficial effect. However, the data show that mitochondrial dysfunction causes negative phenotypes when it occurs during development. I would suggest only using the term 'maladaptive response' consistently throughout as this indicates a negative response to mitochondrial dysfunction during development.

Response 3.17: Following the reviewer's advice, we have primarily used the term "maladaptive response" throughout the manuscript to denote the detrimental changes associated with CI dysfunction during development.

18. There is very little actual discussion of the findings in the Results/Discussion section and the conclusion again confuses the idea of whether the study is focused on healthy lifespan extension (ageing), pathogenic mitochondrial dysfunction/disease, or both. It would be much better to include a full discussion section. Several interesting discussion points could be included. (1) Regarding mitochondrial disease, the relevance of early (childhood) onset versus adult-onset mitochondrial disease could be discussed. (2) Discussion of the omics changes in the context of other mitochondrial dysfunction/disease models. Are there any candidate molecules/pathways that might be contributing to the maladaptive response to developmental CI deficiency? If so, could these be targeted therapeutically. (3) Discussion of the potential lifespan extension caused by adult only CI inhibition in the context of other related *Drosophila* models e.g., Copeland 2009. (4) Regarding lifespan extension, there is a large literature in *C. elegans* on developmental timing, mitochondrial dysfunction and lifespan extension that should be discussed in light of the current findings. (5) Expand on the possible explanations for the finding that adult flies can function normally with an 80% reduction in CI activity. Are there other examples of this? How does this relate to adult-onset mitochondrial disease?

Response 3.18: We have revised the results/discussion/conclusions sections to address the reviewer's recommendations, providing a deeper analysis of our findings and their implications:

1. We delve into the impact of our research in the context of mitochondrial disease, especially focusing on the diversity of mitochondrial symptom manifestations, with some cases presenting in childhood and others only in adulthood.
2. The discussion now includes a comparison of the omics changes observed in our study with those in previously published models, with a particular emphasis on identifying factors that may contribute to the maladaptive response associated with CI dysfunction during development.

Full Revision

3. We explore the findings that individuals with up to 75% CI dysfunction exhibit longer lifespans than controls with normal CI levels and discuss the potential implications of this phenomenon.
4. The discussion also contrasts our findings with those in *C. elegans*, where developmental dysfunction of CI extends lifespan, but adult dysfunction does not produce a similar effect, highlighting both the similarities and differences in these models.

Minor comments:

1. Supplemental tables showing the transcriptomic, metabolomic and proteomic data should be included and these data should be deposited in publicly accessible databases (GEO etc).

Response 3.19: As recommended by the reviewer, we have included supplemental tables showcasing the transcriptomic, metabolomic, and proteomic data in our study. These data have also been deposited in publicly accessible databases as detailed in materials and methods.

2. Descriptions of what the charts represent (mean, SEM etc), sample numbers and details of statistical tests used should be included in all figure legends.

Response 3.20: We have revised all figure legends to include an appropriate description of what is being represented.

3. Details of scale bars should be included in the figure legends.

Response 3.21: Details of the scale bars have been included in the figure legends as requested.

4. 'Figure S2H and I' on p.6 should be S2I and J.

Response 3.22: The corrections have been made as indicated.

5. 'Data' is plural, so do not write 'This data...'

Response 3.23: While it is acknowledged that 'data' is technically a plural noun, in contemporary usage it is often paired with both plural and singular verbs. However, for consistency in our manuscript, we have chosen to treat 'data' as plural throughout.

6. Typo in title to Figure 3D.

Response 3.24. The typo has been corrected.

7. The abbreviations used for different units changes throughout the methods section (e.g minutes/min/m; g/rcf). In the case of centrifugal speeds, in some cases rpm was used to describe the centrifugal spin. Please give the rcf, as rpm differs between centrifuges.

Response 3.25. Abbreviations are now in a consistent format.

8. Include the dilutions of the antibodies using in the methods.

Response 3.26. We have added the information as requested.

9. The manuscript should be read carefully to correct grammatical errors and typos.

Response 3.27: We have thoroughly revised the manuscript to correct any grammatical errors and typos.

Reviewer #3 (Significance (Required)):

The study is well designed, well executed and the data are largely convincing. However, it generally not clear whether the study is focusing on the pathogenic consequences of mitochondrial dysfunction/disease, mitochondrial dysfunction in healthy ageing (lifespan), or both. The interpretations of the results are quite confusing in places, the figures could be much better presented, data not shown needs to be included, and a proper discussion section added to fully discuss the findings in the context of the literature. With these improvements, then the

Full Revision

manuscript with would provide an important advance in the field and will be of interest to researchers in ageing and mitochondrial biology/disease.

I have expertise in mitochondrial biology/metabolism, Drosophila and mouse models.

Summary: We thank the reviewer for their positive comments on the design and execution of our experiments. Following their suggestions, we have made modifications to clarify the focus of our study, highlighting that mitochondrial dysfunction during development significantly impacts adult survival, while adult flies exhibit remarkable tolerance to such dysfunction. As we did not undertake extensive backcrossing, and our experiments predominantly compared short- versus long-lived CI-depleted flies, our study does not centre on the potential role of disrupting CI as a strategy to extend lifespan.

Full Revision

Figures

Rebuttal Figure 1: Mitochondrial oxygen consumption rate (OCR) was plotted against mRNA levels, revealing a robust and statistically significant correlation. This correlation is characterised by a clear distinction between control groups (upper section) and RNAi groups (lower section).

Full Revision

Rebuttal Figure 2: Relative ratio of NAD(+):NADH in the genotypes indicate. Both CI deficient models have a lower ratio when compared to controls, a small but significant differences is observed between D+A and A in the ND-18 RNAi model. * = $p < 0.05$, ** = $p < 0.01$, *** = $p < 0.001$, **** = $p < 0.0001$.

Full Revision

Rebuttal Figure 3: Relative abundance of ND-75 as detected in proteomic analysis. RNA interference targeting ND-18 results in a reduction of ND-75 levels by over 50% relative to control groups. No significant differences were detected between the D+A and A groups within the same genotype. **** = $p < 0.0001$.

Full Revision

Rebuttal Figure 4. Expression of genes and metabolites involved in glycolysis shows no consistent changes across both CI KD models. (Top) Volcano plots display changes in gene expression in ND-18 KD (left) and ND-75 KD (right) comparing D+A versus A-only. Genes involved in glycolysis are indicated in red when significantly altered and in blue when not. (Bottom) Volcano plots illustrate changes in metabolite expression in ND-18 KD (left) and ND-75 KD (right) comparing D+A versus A-only. Metabolites involved in glycolysis are shown in red when significantly altered and in blue for those that are not significantly different between the D+A and A-only groups within the same genotype.

Full Revision

References

- 1 Aparicio, R., Rana, A. & Walker, D. W. Upregulation of the Autophagy Adaptor p62/SQSTM1 Prolongs Health and Lifespan in Middle-Aged *Drosophila*. *Cell reports* 28, 1029-1040 e1025 (2019). <https://doi.org/10.1016/j.celrep.2019.06.070>
- 2 Rana, A. et al. Promoting Drp1-mediated mitochondrial fission in midlife prolongs healthy lifespan of *Drosophila melanogaster*. *Nature communications* 8, 448 (2017). <https://doi.org/10.1038/s41467-017-00525-4>
10.1038/s41467-017-00525-4 [pii]
- 3 Rana, A., Rera, M. & Walker, D. W. Parkin overexpression during aging reduces proteotoxicity, alters mitochondrial dynamics, and extends lifespan. *Proc Natl Acad Sci U S A* 110, 8638-8643 (2013). <https://doi.org/10.1073/pnas.1216197110>
- 4 Copeland, J. M. et al. Extension of *Drosophila* life span by RNAi of the mitochondrial respiratory chain. *Curr Biol* 19, 1591-1598 (2009). [https://doi.org/S0960-9822\(09\)01586-3](https://doi.org/S0960-9822(09)01586-3) [pii]
10.1016/j.cub.2009.08.016
- 5 Urena, E. et al. Trametinib ameliorates aging-associated gut pathology in *Drosophila* females by reducing Pol III activity in intestinal stem cells. *Proc Natl Acad Sci U S A* 121, e2311313121 (2024). <https://doi.org/10.1073/pnas.2311313121>
- 6 Regan, J. C. et al. Sexual identity of enterocytes regulates autophagy to determine intestinal health, lifespan and responses to rapamycin. *Nat Aging* 2, 1145-1158 (2022). <https://doi.org/10.1038/s43587-022-00308-7>
- 7 Lu, Y. X. et al. A TORC1-histone axis regulates chromatin organisation and non-canonical induction of autophagy to ameliorate ageing. *eLife* 10 (2021). <https://doi.org/10.7554/eLife.62233>
- 8 Scialo, F., Sriram, A., Stefanatos, R. & Sanz, A. Practical Recommendations for the Use of the GeneSwitch Gal4 System to Knock-Down Genes in *Drosophila melanogaster*. *PLoS One* 11, e0161817 (2016). <https://doi.org/10.1371/journal.pone.0161817>
- 9 Bosch, J. A., Sumabat, T. M. & Hariharan, I. K. Persistence of RNAi-Mediated Knock-down in *Drosophila* Complicates Mosaic Analysis Yet Enables Highly Sensitive Lineage Tracing. *Genetics* 203, 109-118 (2016). <https://doi.org/10.1534/genetics.116.187062>
- 10 Jacobs, H. T., George, J. & Kemppainen, E. Regulation of growth in *Drosophila melanogaster*: the roles of mitochondrial metabolism. *J Biochem* 167, 267-277 (2020). <https://doi.org/10.1093/jb/mvaa002>
- 11 Elkholi, R. et al. MDM2 Integrates Cellular Respiration and Apoptotic Signaling through NDUFS1 and the Mitochondrial Network. *Mol Cell* 74, 452-465 e457 (2019). <https://doi.org/10.1016/j.molcel.2019.02.012>
- 12 Nouws, J., Te Brinke, H., Nijtmans, L. G. & Houten, S. M. ACAD9, a complex I assembly factor with a moonlighting function in fatty acid oxidation deficiencies. *Hum Mol Genet* 23, 1311-1319 (2014). <https://doi.org/10.1093/hmg/ddt521>
- 13 Poirier, L., Shane, A., Zheng, J. & Seroude, L. Characterization of the *Drosophila* gene-switch system in aging studies: a cautionary tale. *Aging Cell* 7, 758-770 (2008). <https://doi.org/10.1111/j.1474-9726.2008.00421.x>

Full Revision

- 14 Schmid, E. T., Pyo, J. H. & Walker, D. W. Neuronal induction of BNIP3-mediated mitophagy slows systemic aging in *Drosophila*. *Nat Aging* 2, 494-507 (2022).
<https://doi.org/10.1038/s43587-022-00214-y>
- 15 Juricic, P. et al. Long-lasting geroprotection from brief rapamycin treatment in early adulthood by persistently increased intestinal autophagy. *Nat Aging* 2, 824-836 (2022).
<https://doi.org/10.1038/s43587-022-00278-w>
- 16 Bolukbasi, E. et al. Cell type-specific modulation of healthspan by Forkhead family transcription factors in the nervous system. *Proc Natl Acad Sci U S A* 118 (2021).
<https://doi.org/10.1073/pnas.2011491118>
- 17 Filer, D. et al. RNA polymerase III limits longevity downstream of TORC1. *Nature* 552, 263-267 (2017). <https://doi.org/10.1038/nature25007>
- 18 Granat, L. et al. Yeast NDI1 reconfigures neuronal metabolism and prevents the unfolded protein response in mitochondrial complex I deficiency. *PLoS Genet* 19, e1010793 (2023). <https://doi.org/10.1371/journal.pgen.1010793>
- 19 Hunt, R. J. et al. Mitochondrial stress causes neuronal dysfunction via an ATF4-dependent increase in L-2-hydroxyglutarate. *J Cell Biol* 218, 4007-4016 (2019).
<https://doi.org/10.1083/jcb.201904148>

Dear Alberto,

Thank you for submitting your revised manuscript, which was previously peer-reviewed at Review Commons. It has now been seen by all of the original referees.

As you can see, the referees find that the study is significantly improved during revision and recommend publication. However, I need you to address the points below before I can accept the manuscript. I apologize in beforehand for the lengthiness of the list.

- We note that there are outstanding referee concerns. Please address them as outlined below and provide a point-by-point response:
 - o Please address the remaining concern (1) of referee #3 by clarifying the control lines used in respective figures and the backcrossing status.
 - o We note that main point 1 of referee #2, regarding the absence of a condition where the CI is inactivated only during the development, but reactivated during adulthood, was also raised in the previous round of peer-review and you responded to it by pointing out the technical challenges preventing from addressing this concern. Please add a discussion point into the manuscript text acknowledging the lack of this condition and the aforementioned technical challenges.
 - o Please address main point 2 of referee #2 by making the required textual changes.
 - o We note that main points 3 and 4 of referee #2 regarding the data presentation (i.e. controls were presented separately not put onto the same graph as the experimental conditions) was also raised in the previous round (referee #1, major point 1 of the first round). We appreciate that you provided the numerical data as a part of the source data in response to these comments. However, we agree with referees #1 and #2 that, as is, this comparison is not sufficiently accessible for the readers. Therefore, we would like to ask you to present the controls and the knockdown data on the same graph and include it in an Appendix file (containing a table of contents with page numbers on the title page; the nomenclature and manuscript callouts of the figures would be Appendix Figure S1, etc.)
 - o Please address main point 6 of referee #2 textually by discussing the potential reasons underlying the differences in phenotypic strengths of ND-18 and ND-75 knockdowns.
 - o Please address all other remaining (minor) concerns.
- Please make the dataset PXD043791 publicly available.
- As per our guidelines, please add a 'Data Availability Section', where you list datasets that were generated in the study - i.e. PXD043791, GSE237015, and provide URLs that directly resolve to the dataset directly. Please see <https://www.embopress.org/page/journal/14693178/authorguide#dataavailability> for further information.
- Please rename the Ethical Statement/Conflict of Interest section as Disclosure Statement and Competing Interests.
- Please add email addresses of the corresponding authors to the title page of the manuscript.
- Please remove the Author Contributions section from the manuscript text.
- As per our format requirements, in the reference list, citations should be listed in alphabetical order and then chronologically, with the authors' surnames and initials inverted; where there are more than 10 authors on a paper, 10 will be listed, followed by 'et al.'. Please see <https://www.embopress.org/page/journal/14693178/authorguide#referencesformat>
- Please fill out and include an author checklist as listed in our online guidelines (<https://www.embopress.org/page/journal/14693178/authorguide>)
- Funding information needs to be fully reported both in the manuscript file and the manuscript tracking system. We note that fellowships from the Uehara Memorial Foundation and International Medical Research Foundation are currently missing from the manuscript tracking system.
- We note that you have 4 supplemental figures, for which our nomenclature is Expanded View figures (Figure EV1, Figure EV2 etc). Please update their source file names, titles in the manuscript tracking system, figure legends in the manuscript, callouts in the manuscript. (Please see <https://www.embopress.org/page/journal/14693178/authorguide#expandedview>).
- Related to the point above, Suppl. Figures 2-4 span across multiple pages, which is not allowed as per journal format requirements. Please rearrange them in a way that one EV Figure is presented on a single page. Of note, we can accommodate up to 5 EV Figures in case you would like to split EV Figures.
- We note that there is an excel file named as Supplemental Data, which contains source data as well. Please refer to the email sent by our Source Data Coordinator Dr. Hannah Sonntag (dated 06.12.2024) regarding restructuring it as one file per figure. Of note, I realize that some tabs only present the mean and SEM values used to construct the graphs. Individual data points need to be reported as well in the source data file. The datasets other than source data need to be resubmitted as Dataset EV1 containing its title and description.
- We note that there is a file entitled as Supplementary Information. The content entitled Materials and Methods should be included into the main manuscript as follows. Supplementary Tables 1, 2 and 3 should be moved into the Reagents & Tools table (please see the below point). The section entitled as In-house RNA sequencing pipeline should be moved to the Methods section of the manuscript. References should also be moved to the main text.
- All research articles submitted as revised versions must include a structured methods section that includes a Reagents and Tools Table followed by a Methods and Protocols section. Please see <https://www.embopress.org/page/journal/14693178/authorguide#structuredmethods> for further information.
- The manuscript sections should be in the following order: Title page - Abstract & Keywords - Introduction - Results -

Discussion - Methods - Data Availability - Acknowledgments - Disclosure Statement & Competing Interests - References - Figure Legends - (Main Tables with legends if applicable) - Expanded View Figure Legends.

- Our production/data editors have asked you to clarify several points in the figure legends:

- o Please note that the exact p values are not provided in the legends of figures 1B-E, H; 2A-E; supplementary figures 1B, C, F, G; 2H.

- o Please indicate the statistical test used for data analysis in the legends of figures 1B-E, H; 2A-E; 3A, B, D, E-G; 4A-F; supplementary figures 1A-C, E-G; 2H, 3E, F, G, H, I; 4B, I.

- o Please note that information related to n is missing in the legends of figures 1E, G; 2D, E; 3E-G; 4A, B, D; supplementary figures 1A-D, H-I, K; 2 D, F, G-I; 3E, G, H, I; 4B.

- Papers published in EMBO Reports include a 'synopsis' and 'bullet points' to further enhance discoverability. Both are displayed on the html version of the paper and are freely accessible to all readers. The synopsis includes a short standfirst summarizing the study in 1 or 2 sentences (max 35 words) that summarize the paper and are provided by the authors and streamlined by the handling editor. I would therefore ask you to include your synopsis blurb and 3-5 bullet points listing the key experimental findings.

- In addition, please provide an image for the synopsis. This image should provide a rapid overview of the question addressed in the study but still needs to be kept fairly modest since the image size cannot exceed 550 (width) x 300-600 (height) pixels.

Thank you again for giving us to consider your manuscript for EMBO Reports, I look forward to your minor revision.

Kind regards,

Deniz

--

Deniz Senyilmaz Tiebe, PhD
Senior Scientific Editor
EMBO Reports

Referee #1:

After the first review through Review Commons, in this revised manuscript, the authors have addressed correctly the concerns raised by the Reviewers. Therefore, the current version is already suitable for publication in EMBO Reports.

Referee #2:

In this paper, Stefanatos et al. impair mitochondrial complex I (CI) subunits in *Drosophila melanogaster* at various stages of development. They examine the behavioral and molecular consequences of those manipulations. The logic motivating the study is that mitochondrial functions are well conserved. By studying the molecular consequences of CI loss in *Drosophila*, it might be possible to learn more about human conditions where CI is lost. This makes sense on a molecular level.

The authors use the GAL4/UAS expression system to impair two *Drosophila* CI subunits by RNAi, ND-18 or ND-75. They temporally control the expression of the UAS-RNAi transgenes using the RU486-based Gene Switch system or the GAL80TS-based TARGET system. The main finding is that if CI is impaired throughout life (post-hatching, Development + Adulthood, D+A), there are severe adult survival phenotypes (Figs. 1 and 2), but there is not impaired survival if CI is impaired only during adulthood (A). The D+A impairments of Complex I also render the animals less resilient to challenges like 2.5% hydrogen peroxide, starvation, or heat exposure (Fig. 2).

To understand potential mechanisms that may be mediating these effects, the authors go on to conduct transcriptomic (Fig. 3) and metabolomic (Fig. 4) analyses for driver controls, D+A losses of CI, and A-only losses of CI. In general, the transcriptomic and metabolomic analyses show concordance between ND-18 and ND-75 losses of function. Based on gene ontology meta-analyses of the transcriptomic data (Fig. 3), there seem to be significant changes in genetic factors controlling processes like proteolysis, hexose metabolism, triglyceride homeostasis, fatty acid beta-oxidation, oligosaccharide catabolism, and lipid transport. Based on meta-analyses of the metabolomic data (Fig. 4), there seem to be significant changes in factors controlling lysine degradation, arginine/proline metabolism, fatty acid degradation, starch and sucrose metabolism, cysteine and methionine metabolism, butanoate metabolism, glycine/serine/threonine metabolism, and amino sugar/nucleotide sugar metabolism. These omics-level analyses could be starting points or hypothesis-generating ideas for follow-up studies to understand what might happen when CI is impaired during critical developmental periods.

The data in this paper are generally good quality, and there is considerable evidence for some sort of larval-specific critical period for CI function. For this reviewer, there are several experimental-level and interpretation-level issues that need to be tied up.

MAIN POINTS

1. Developmental critical period determining lifespan: Comparing D+A knockdown vs. A knockdown conditions, the authors conclude that there is a critical period for Complex I function during development. They narrow the period down with temperature shift experiments in Figure 1F and conclude that the earlier that CI is knocked down, the more severe the phenotypes. The interpretation in the paper is that CI knockdown is not consequential during adulthood (at least for lifespan) because even when CI function is decreased by 50-75% in adults (Fig. 1D), those A-only adults have normal lifespans (Figs. 1B, C).

The data make sense to rule in larval stages as important. But it is not possible with the current data to rule out adulthood as also being important for these phenotypes. That is because there is no D-only condition attempted. Every knockdown condition attempted also knocked down CI during adulthood. Even if CI function were knocked down during the larval period, it is possible that the lifespan phenotype could have nevertheless been rescued if CI were resupplied after eclosion.

In the absence of any data testing the developmental (larval) period only, this experimental shortcoming would need to be highlighted in a revised Discussion. It would also necessitate a title change ("contributes to" instead of "determines"), as well as several changes of phrase in the paper.

One way to execute a D-only condition would be to use the Gene Switch system (feeding larvae RU486 and then adults non-RU486 food). The GAL80TS TARGET system is probably not ideal because it prevents eclosion after enough larval time at 29°C. But even with the TARGET system, it might be possible to refine conclusions with temperature shifts back to 18°C after spending time at 29°C for L1-L2. All of this assumes that these RNAi-transgenes can be shut off once turned on.

2. Several times the authors invoke a "maladaptive" response - or a "maladaptation" that results from losing CI function in the larval stages. This reviewer is not certain that it is warranted to invoke maladaptation.

A maladaptive response would be occurring if a system were challenged with a perturbation (larval CI loss in this case), and then the system subsequently attempted to suppress the effects of that perturbation with a homeostatic response (unknown). But because of the homeostatic response, there could be a new, downstream physiological problem that is caused, and the long-term result for the animal would be maladaptive (short lifespan, observed) instead of adaptive (normal lifespan despite early CI loss, not observed).

To be clear, all of that is possible. But the difficulty with concluding maladaptation is the absence of a definable homeostatic response. As things stand, the data could mean that there is an early critical period for CI function, nothing more. The downstream problems might simply be a result of that early deficit.

This might seem like a semantic point, but in terms of how the data are ultimately interpreted (lifespan, transcriptomic changes, metabolomic changes, etc.), it is important because what is happening is not necessarily maladaptive. The fact that one sees differences in the transcriptomes or the metabolomes does not constitute evidence of a maladaptation.

3. Controls Missing from Main Figure Graphs: The lifespan graphs compare D+A or A-only knockdowns of CI components. In terms of knowing whether the A-only conditions approximate a normal lifespan - or if they themselves have a defect - it would be helpful to include side-by-side control conditions on the main graphs (EtOH). Right now, those control data are housed in the Supplemental information. This applies to a lot of panels: Figs. 1B, C, H, and Figs. 2A, B, C.

4. Relatedly, the abstract states that mitochondrial dysfunction in adulthood results in flies that are "long-lived" and "stress resistant." An interpretation of that sentence is that the A-only knockdowns are healthier than wild-type/control flies. Is that true, or are they just statistically indistinguishable from controls? It is tricky to compare back and forth between the main figures and the supplemental.

5. The omics-level analyses are interesting, but somewhat unsatisfying. Figures 3 and 4 are good in terms of being hypothesis-generating mechanisms, but there is no accompanying functional analysis. Gene Ontology (or similar) is a starting point for organizing omics data, but it does not substitute for functional analysis, which is a strength of the *Drosophila* system.

6. Based on the early data (Figure 1), knockdown of ND-75 is a less severe manipulation than knockdown of ND-18. But based on the transcriptomic and metabolomic data (Figure 3), it seems that ND-75 KD yields far more changes. Some discussion about the differences between ND-18 and ND-75 knockdowns is warranted, even though they show some concordance in the types of factors that are upregulated or downregulated.

MINOR POINTS

1. Several of the Figure panels have unusual fonts or text that appears stretched or squished from resizing pictures of graphs.

2. Abstract: "discrete" is the term intended.

3. p. 3 Introduction: "...a mountain of evidence has accumulated, suggesting that the severity of pathology induced by mitochondrial/CI dysfunction depends on a factor unrelated to the dysfunction itself."

As written, this conclusion does not seem to make sense, given that the primary defect is to CI gene function. I understand that the authors are trying to draw a distinction between an initial insult to mitochondria vs. the ability of the organism to deal with that. But it does not make the "factor" unrelated to the insult.

4. "...more adaptive yet also more sensitive" - this phrasing does not make sense. More sensitive would seem to imply less adaptive.

5. p. 11: "Our data, reconcile previously contradictory findings, and indicate that fully functional CI is necessary during development but not adulthood." For the reasons described in Main Point 1 above, the current data do not rule out a function during adulthood. Moreover, the text needs to be more specific about what is being reconciled (as opposed to referring briefly to "conflicting published reports"). If it is a lengthy explanation, then the Discussion would be a better place than the Introduction.

6. p. 12: "These data suggest that...loss in CI levels...may not contribute in itself to ageing or the onset of age-related diseases." The adult *Drosophila* data for Complex I cannot be extrapolated to these other situations. This kind of extrapolation occurs in the Discussion too (e.g., p. 20 - "if our discoveries could be translated to humans..."). Without evidence of a critical period in mammals, the extrapolation is not warranted.

7. pp. 16-17, Fig. 2, and Fig. S2: The idea of an "underdeveloped fat body" is not clear from the images shown.

8. Fig. 1: The differences between the Gene Switch RU486-based system for knockdown and the TARGET GAL80TS system are curious. It is difficult to tell which one is stronger. The Gene Switch system seems to be a more severe effect on CI-linked respiration, yet the overall effects for the organism are mild enough that one gets eclosed adults to evaluate. By contrast, the TARGET system only subtly knocks down CI-linked respiration but precludes eclosion of adults.

Referee #3:

The authors mostly addressed my previous comments, and the manuscript is much improved, but I have a few of additional comments on the revised manuscript:

(1) The authors state in response 3.3, about including the control data in the main figures, that they have not extensively backcrossed the control and RNAi flies. In the results the controls are described as 'UAS-Empty/Control' but there are no 'UAS-empty' or control RNAi lines listed in table S1 or mentioned in the figure legends. Moreover, below Table S1 it says 'Lines were backcrossed into w1118 background for 11 generations'. If this included a UAS-empty or control RNAi line, then this would contradict the statement in response 3.3. Can the authors please include the details of the control line used and state whether this line was backcrossed into the same background as the ND-75 and ND-18 RNAi lines. They should also state which control line was used in the figure legends.

(2) Please include catalogue numbers and supplier details for all the antibodies used. This is important so others can reproduce the experiments using the same reagents.

(3) Since the original submission of this manuscript a paper has been published (PMID 38304969) that models CI deficiency in *Drosophila* using two different RNAi lines against ND-75, one of which is the same as the line used by the authors (100733/KK). The authors should mention this paper and that the strength of CI inhibition can be controlled by using RNAi lines with different efficiencies. This is important for researchers wishing to use RNAi to model CI deficiency in *Drosophila*.

Referee #1:

After the first review through Review Commons, in this revised manuscript, the authors have addressed correctly the concerns raised by the Reviewers. Therefore, the current version is already suitable for publication in EMBO Reports.

Referee #2:

In this paper, Stefanatos et al. impair mitochondrial complex I (CI) subunits in *Drosophila melanogaster* at various stages of

development. They examine the behavioral and molecular consequences of those manipulations. The logic motivating the study is that mitochondrial functions are well conserved. By studying the molecular consequences of CI loss in *Drosophila*, it might be possible to learn more about human conditions where CI is lost. This makes sense on a molecular level.

The authors use the GAL4/UAS expression system to impair two *Drosophila* CI subunits by RNAi, ND-18 or ND-75. They temporally control the expression of the UAS-RNAi transgenes using the RU486-based Gene Switch system or the GAL80TS-based TARGET system. The main finding is that if CI is impaired throughout life (post-hatching, Development + Adulthood, D+A), there are severe adult survival phenotypes (Figs. 1 and 2), but there is not impaired survival if CI is impaired only during adulthood (A). The D+A impairments of Complex I also render the animals less resilient to challenges like 2.5% hydrogen peroxide, starvation, or heat exposure (Fig. 2).

To understand potential mechanisms that may be mediating these effects, the authors go on to conduct transcriptomic (Fig. 3) and metabolomic (Fig. 4) analyses for driver controls, D+A losses of CI, and A-only losses of CI. In general, the transcriptomic and metabolomic analyses show concordance between ND-18 and ND-75 losses of function. Based on gene ontology meta-analyses of the transcriptomic data (Fig. 3), there seem to be significant changes in genetic factors controlling processes like proteolysis, hexose metabolism, triglyceride homeostasis, fatty acid beta-oxidation, oligosaccharide catabolism, and lipid transport. Based on meta-analyses of the metabolomic data (Fig. 4), there seem to be significant changes in factors controlling lysine degradation, arginine/proline metabolism, fatty acid degradation, starch and sucrose metabolism, cysteine and methionine metabolism, butanoate metabolism, glycine/serine/threonine metabolism, and amino sugar/nucleotide sugar metabolism. These omics-level analyses could be starting points or hypothesis-generating ideas for follow-up studies to understand what might happen when CI is impaired during critical developmental periods.

The data in this paper are generally good quality, and there is considerable evidence for some sort of larval-specific critical period for CI function. For this reviewer, there are several experimental-level and interpretation-level issues that need to be tied up.

MAIN POINTS

1. Developmental critical period determining lifespan: Comparing D+A knockdown vs. A knockdown conditions, the authors conclude that there is a critical period for Complex I function during development. They narrow the period down with temperature shift experiments in Figure 1F and conclude that the earlier that CI is knocked down, the more severe the phenotypes. The interpretation in the paper is that CI knockdown is not consequential during adulthood (at least for lifespan) because even when CI function is decreased by 50-75% in adults (Fig. 1D), those A-only adults have normal lifespans (Figs. 1B, C).

The data make sense to rule in larval stages as important. But it is not possible with the current data to rule out adulthood as also being important for these phenotypes. That is because there is no D-only condition attempted. Every knockdown condition attempted also knocked down CI during adulthood. Even if CI function were knocked down during the larval period, it is possible that the lifespan phenotype could have nevertheless been rescued if CI were resupplied after eclosion.

In the absence of any data testing the developmental (larval) period only, this experimental shortcoming would need to be highlighted in a revised Discussion. It would also necessitate a title change ("contributes to" instead of "determines"), as well as several changes of phrase in the paper.

One way to execute a D-only condition would be to use the Gene Switch system (feeding larvae RU486 and then adults non-RU486 food). The GAL80TS TARGET system is probably not ideal because it prevents eclosion after enough larval time at 29°C. But even with the TARGET system, it might be possible to refine conclusions with temperature shifts back to 18°C after spending time at 29°C for L1-L2. All of this assumes that these RNAi-transgenes can be shut off once turned on.

2. Several times the authors invoke a "maladaptive" response - or a "maladaptation" that results from losing CI function in the larval stages. This reviewer is not certain that it is warranted to invoke maladaptation.

A maladaptive response would be occurring if a system were challenged with a perturbation (larval CI loss in this case), and then the system subsequently attempted to suppress the effects of that perturbation with a homeostatic response (unknown). But because of the homeostatic response, there could be a new, downstream physiological problem that is caused, and the long-term result for the animal would be maladaptive (short lifespan, observed) instead of adaptive (normal lifespan despite early CI loss, not observed).

To be clear, all of that is possible. But the difficulty with concluding maladaptation is the absence of a definable homeostatic response. As things stand, the data could mean that there is an early critical period for CI function, nothing more. The downstream problems might simply be a result of that early deficit.

This might seem like a semantic point, but in terms of how the data are ultimately interpreted (lifespan, transcriptomic changes, metabolomic changes, etc.), it is important because what is happening is not necessarily maladaptive. The fact that one sees differences in the transcriptomes or the metabolomes does not constitute evidence of a maladaptation.

3. Controls Missing from Main Figure Graphs: The lifespan graphs compare D+A or A-only knockdowns of CI components. In terms of knowing whether the A-only conditions approximate a normal lifespan - or if they themselves have a defect - it would be helpful to include side-by-side control conditions on the main graphs (EtOH). Right now, those control data are housed in the Supplemental information. This applies to a lot of panels: Figs. 1B, C, H, and Figs. 2A, B, C.

4. Relatedly, the abstract states that mitochondrial dysfunction in adulthood results in flies that are "long-lived" and "stress resistant." An interpretation of that sentence is that the A-only knockdowns are healthier than wild-type/control flies. Is that true, or are they just statistically indistinguishable from controls? It is tricky to compare back and forth between the main figures and the supplemental.

5. The omics-level analyses are interesting, but somewhat unsatisfying. Figures 3 and 4 are good in terms of being hypothesis-generating mechanisms, but there is no accompanying functional analysis. Gene Ontology (or similar) is a starting point for organizing omics data, but it does not substitute for functional analysis, which is a strength of the *Drosophila* system.

6. Based on the early data (Figure 1), knockdown of ND-75 is a less severe manipulation than knockdown of ND-18. But based on the transcriptomic and metabolomic data (Figure 3), it seems that ND-75 KD yields far more changes. Some discussion about the differences between ND-18 and ND-75 knockdowns is warranted, even though they show some concordance in the types of factors that are upregulated or downregulated.

MINOR POINTS

1. Several of the Figure panels have unusual fonts or text that appears stretched or squished from resizing pictures of graphs.

2. Abstract: "discrete" is the term intended.

3. p. 3 Introduction: "...a mountain of evidence has accumulated, suggesting that the severity of pathology induced by mitochondrial/CI dysfunction depends on a factor unrelated to the dysfunction itself."

As written, this conclusion does not seem to make sense, given that the primary defect is to CI gene function. I understand that the authors are trying to draw a distinction between an initial insult to mitochondria vs. the ability of the organism to deal with that. But it does not make the "factor" unrelated to the insult.

4. "...more adaptive yet also more sensitive" - this phrasing does not make sense. More sensitive would seem to imply less adaptive.

5. p. 11: "Our data, reconcile previously contradictory findings, and indicate that fully functional CI is necessary during development but not adulthood." For the reasons described in Main Point 1 above, the current data do not rule out a function during adulthood. Moreover, the text needs to be more specific about what is being reconciled (as opposed to referring briefly to "conflicting published reports"). If it is a lengthy explanation, then the Discussion would be a better place than the Introduction.

6. p. 12: "These data suggest that...loss in CI levels...may not contribute in itself to ageing or the onset of age-related diseases." The adult *Drosophila* data for Complex I cannot be extrapolated to these other situations. This kind of extrapolation occurs in the Discussion too (e.g., p. 20 - "if our discoveries could be translated to humans..."). Without evidence of a critical period in mammals, the extrapolation is not warranted.

7. pp. 16-17, Fig. 2, and Fig. S2: The idea of an "underdeveloped fat body" is not clear from the images shown.

8. Fig. 1: The differences between the Gene Switch RU486-based system for knockdown and the TARGET GAL80TS system are curious. It is difficult to tell which one is stronger. The Gene Switch system seems to be a more severe effect on CI-linked respiration, yet the overall effects for the organism are mild enough that one gets eclosed adults to evaluate. By contrast, the TARGET system only subtly knocks down CI-linked respiration but precludes eclosion of adults.

Referee #3:

The authors mostly addressed my previous comments, and the manuscript is much improved, but I have a few of additional comments on the revised manuscript:

(1) The authors state in response 3.3, about including the control data in the main figures, that they have not extensively backcrossed the control and RNAi flies. In the results the controls are described as 'UAS-Empty/Control' but there are no 'UAS-empty' or control RNAi lines listed in table S1 or mentioned in the figure legends. Moreover, below Table S1 it says 'Lines were backcrossed into w1118 background for 11 generations'. If this included a UAS-empty or control RNAi line, then this would

contradict the statement in response 3.3. Can the authors please include the details of the control line used and state whether this line was backcrossed into the same background as the ND-75 and ND-18 RNAi lines. They should also state which control line was used in the figure legends.

(2) Please include catalogue numbers and supplier details for all the antibodies used. This is important so others can reproduce the experiments using the same reagents.

(3) Since the original submission of this manuscript a paper has been published (PMID 38304969) that models CI deficiency in *Drosophila* using two different RNAi lines against ND-75, one of which is the same as the line used by the authors (100733/KK). The authors should mention this paper and that the strength of CI inhibition can be controlled by using RNAi lines with different efficiencies. This is important for researchers wishing to use RNAi to model CI deficiency in *Drosophila*.

Rev_Com_number: RC-2023-02211

New_manu_number: EMBOR-2024-60577V1-T

Corr_author: Sanz

Title: Developmental mitochondrial Complex I activity determines lifespan

Editorial comments

1. Please address the remaining concern (1) of referee #3 by clarifying the control lines used in respective figures and the backcrossing status.

Response 1.1. The control group was used to account for any unintended effects of RU-486 during development and to provide a baseline for CI levels in specific experiments (e.g., mitochondrial respiration). This group comprised progeny from the cross between virgin females *tubulinGeneSwitch* (tubGS) and males from the Vienna Stock Center control line 60100, which is used to control for the insertion of the RNAi transgene. Our initial response to reviewers noted that the concentration of RU-486 used causes negligible transcriptomic or proteomic alterations in controls during development. Information about line 60100 has been included in the relevant methods section.

Other control lines, such as 60101 and 60102, have been used in separate projects in our lab to assess the effects of RU-486 by measuring parameters such as mitochondrial respiration, starvation resistance, and transcriptomic profiles of other mitochondrial-deficient flies. These experiments consistently showed no significant effects from using RU-486 at low concentrations during development.

We backcrossed lines such as tubGS, tubGal80ts, and daGal4 that were not originally in the w1118 background used for the RNAi KK collection by VDRC. These lines were backcrossed upon receipt in the lab, though not simultaneously or extensively, which is why they are described as "not extensively backcrossed."

The RU-486 system minimises background effects by enabling direct comparisons between RU-486-fed and non-fed flies. Our experiments were designed to compare control and RNAi lines under both conditions during development, rather than directly comparing controls and RNAi lines, thereby reducing the need for continuous simultaneous backcrossing to control for background effects in complex experiments such as lifespan or survival studies.

As requested, we have provided data for all groups in an appendix file.

2. We note that main point 1 of referee #2, regarding the absence of a condition where the CI is inactivated only during the development, but reactivated during adulthood, was also raised in the previous round of peer-review and you responded to it by pointing out the technical challenges preventing from addressing this concern. Please add a discussion point into the manuscript text acknowledging the lack of this condition and the aforementioned technical challenges.

Response 1.2: We have incorporated the requested information into the paper's discussion section.

3. Please address main point 2 of referee #2 by making the required textual changes.

Response 1.3: The definition (Cambridge Dictionary) of "maladaptive" within the context of biology is twofold: (1) "Not having the ability to change to suit different conditions." (2) "Having an adaptation (=changed feature) that is not suitable for particular conditions." We contend that the term "maladaptive" is appropriately used in the text, as it describes flies that exhibit impaired stress adaptation and shortened lifespan due to a lack of CI during development. These characteristics are likely caused by some of the transcriptomic, proteomic, and metabolomic changes we described in our paper. We agree with the reviewer that a fully functional CI is crucial during the critical developmental period, and its deficiency results in maladaptation, consistent with the definitions provided. We have incorporated a brief note to clarify that the specific nature of this maladaptation remains unidentified at present and will require future exploration.

4. We note that main points 3 and 4 of referee #2 regarding the data presentation (i.e. controls were presented separately not put onto the same graph as the experimental conditions) was also raised in the previous round (referee #1, major point 1 of the first round). We appreciate that you provided the numerical data as a part of the source data in response to these comments. However, we agree with referees #1 and #2 that, as is, this comparison is not sufficiently accessible for the readers. Therefore, we would like to ask you to present the controls and the knockdown data on the same graph and include it in an Appendix file (containing a table of contents with page numbers on the title page; the nomenclature and manuscript callouts of the figures would be Appendix Figure S1, etc.)

Response 1.4: We have prepared the requested appendix, which is included as supplementary information.

5. Please address main point 6 of referee #2 textually by discussing the potential reasons underlying the differences in phenotypic strengths of ND-18 and ND-75 knockdowns.

Response 1.5: We have added a paragraph in the discussion as requested.

6. Please address all other remaining (minor) concerns.

Response 1.6: We have addressed the other concerns of the reviewers.

7. Please make the dataset PXD043791 publicly available.
• As per our guidelines, please add a 'Data Availability Section', where you list datasets that were generated in the study - i.e. PXD043791, GSE237015, and provide URLs that directly resolve to the dataset directly. Please see <https://www.embopress.org/page/journal/14693178/authorguide#dataavailability> for further information.

Response 1.7: We have done as requested.

8. Please rename the Ethical Statement/Conflict of Interest section as Disclosure Statement and Competing Interests.

Response 1.8: We have done as requested.

9. Please add email addresses of the corresponding authors to the title page of the manuscript.

Response 1.9: We have added the emails.

10. Please remove the Author Contributions section from the manuscript text.

Response 1.10: We have removed the author contribution section.

11. As per our format requirements, in the reference list, citations should be listed in alphabetical order and then chronologically, with the authors' surnames and initials inverted; where there are more than 10 authors on a paper, 10 will be listed, followed by 'et al.'. Please see <https://www.embopress.org/page/journal/14693178/authorguide#referencesformat>

Response 1.11: We have updated the reference format to comply with the requirements of EMBO Reports.

12. Please fill out and include an author checklist as listed in our online guidelines (<https://www.embopress.org/page/journal/14693178/authorguide>)

Response 1.12: We have filled out the checklist as requested.

13. Funding information needs to be fully reported both in the manuscript file and the manuscript tracking system. We note that fellowships from the Uehara Memorial Foundation and International Medical Research Foundation are currently missing from the manuscript tracking system.

Response 1.13: We have added the required information to the tracking system.

- We note that you have 4 supplemental figures, for which our nomenclature is Expanded View figures (Figure EV1, Figure EV2 etc). Please update their source file names, titles in the manuscript tracking system, figure legends in the manuscript, callouts in the manuscript. (Please see <https://www.embopress.org/page/journal/14693178/authorguide#expandedview>).

Response 1.14: We have implemented the requested changes.

- Related to the point above, Suppl. Figures 2-4 span across multiple pages, which is not allowed as per journal format requirements. Please rearrange them in a way that one EV Figure is presented on a single page. Of note, we can accommodate up to 5 EV Figures in case you would like to split EV Figures.

Response 1.15: We have modified the EV figures as requested, resulting in a total of five EV figures.

- We note that there is an excel file named as Supplemental Data, which contains source data as well. Please refer to the email sent by our Source Data Coordinator Dr. Hannah Sonntag (dated 06.12.2024) regarding restructuring it as one file per figure. Of note, I realise that some tabs only present the mean and SEM values used to construct the graphs. Individual data points need to be reported as well in the source data file. The datasets other than source data need to be resubmitted as Dataset EV1 containing its title and description.

Response 1.16: We have provided the requested information in the format specified by Dr Sonntag, including individual data points.

- We note that there is a file entitled as Supplementary Information. The content entitled Materials and Methods should be included into the main manuscript as follows. Supplementary Tables 1, 2 and 3 should be moved into the Reagents & Tools table (please see the below point). The section entitled as In-house RNA sequencing pipeline should be moved to the

Methods section of the manuscript. References should also be moved to the main text.
Response:

Response 1.17: We have implemented the requested changes.

- All research articles submitted as revised versions must include a structured methods section that includes a Reagents and Tools Table followed by a Methods and Protocols section. Please see <https://www.embopress.org/page/journal/14693178/authorguide#structuredmethods> for further information.

Response 1.18: We have added the required Reagents and Tools table.

- The manuscript sections should be in the following order: Title page - Abstract & Keywords - Introduction - Results - Discussion - Methods - Data Availability - Acknowledgments - Disclosure Statement & Competing Interests - References - Figure Legends - (Main Tables with legends if applicable) - Expanded View Figure Legends.

Response 1.18: We have reformatted the manuscript in accordance with the instructions provided above.

- Our production/data editors have asked you to clarify several points in the figure legends:
 - o Please note that the exact p values are not provided in the legends of figures 1B-E, H; 2A-E; supplementary figures 1B, C, F, G; 2H.

Response 1.19: We have added the P values, as obtained in GraphPad Prism, to the revised version. Please note that for very small P values, GraphPad reports them as \$P < 0.001\$ or similar.

- o Please indicate the statistical test used for data analysis in the legends of figures 1B-E, H; 2A-E; 3A, B, D, E-G; 4A-F; supplementary figures 1A-C, E-G; 2H, 3E, F, G, H, I; 4B, I.

Response: 1.20: We have added the statistical test as requested.

- o Please note that information related to n is missing in the legends of figures 1E, G; 2D, E; 3E-G; 4A, B, D; supplementary figures 1A-D, H-I, K; 2 D, F, G-I; 3E, G, H, I; 4B.

Response 1.21: We have added the information about the number of samples/experiments as requested.

- Papers published in EMBO Reports include a 'synopsis' and 'bullet points' to further enhance discoverability. Both are displayed on the html version of the paper and are freely accessible to all readers. The synopsis includes a short standfirst summarising the study in 1 or 2 sentences (max 35 words) that summarise the paper and are provided by the authors and streamlined by the handling editor. I would therefore ask you to include your synopsis blurb and 3-5 bullet points listing the key experimental findings.
- In addition, please provide an image for the synopsis. This image should provide a rapid overview of the question addressed in the study but still needs to be kept fairly modest since the image size cannot exceed 550 (width) x 300-600 (height) pixels.

Response 1.22: We have created the synopsis/bullet points and the requested image.

Referees' comments.

Referee #1:

1. After the first review through Review Commons, in this revised manuscript, the authors have addressed correctly the concerns raised by the Reviewers. Therefore, the current version is already suitable for publication in EMBO Reports.

Response 2.1: We are pleased that the referee is satisfied with our resubmission, and we thank them for their valuable contributions to improving our manuscript.

Referee #2:

1. Developmental critical period determining lifespan: Comparing D+A knockdown vs. A knockdown conditions, the authors conclude that there is a critical period for Complex I function during development. They narrow the period down with temperature shift experiments in Figure 1F and conclude that the earlier that CI is knocked down, the more severe the phenotypes. The interpretation in the paper is that CI knockdown is not consequential during adulthood (at least for lifespan) because even when CI function is decreased by 50-75% in adults (Fig. 1D), those A-only adults have normal lifespans (Figs. 1B, C).

The data make sense to rule in larval stages as important. But it is not possible with the current data to rule out adulthood as also being important for these phenotypes. That is because there is no D-only condition attempted. Every knockdown condition attempted also knocked down CI during adulthood. Even if CI function were knocked down during the larval period, it is possible that the lifespan phenotype could have nevertheless been rescued if CI were resupplied after eclosion.

In the absence of any data testing the developmental (larval) period only, this experimental shortcoming would need to be highlighted in a revised Discussion. It would also necessitate a title change ("contributes to" instead of "determines"), as well as several changes of phrase in the paper.

One way to execute a D-only condition would be to use the Gene Switch system (feeding larvae RU486 and then adults non-RU486 food). The GAL80TS TARGET system is probably not ideal because it prevents eclosion after enough larval time at 29°C. But even with the TARGET system, it might be possible to refine conclusions with temperature shifts back to 18°C after spending time at 29°C for L1-L2. All of this assumes that these RNAi-transgenes can be shut off once turned on.

Response 3.1: We have added a paragraph (in the discussion) addressing the study's limitations, specifically regarding the inability to restore CI activity during adulthood after depleting its activity during development.

In the first rebuttal letter, we explained extensively, with appropriate citations, why the experiments proposed by the reviewer cannot be performed. It is not feasible to restore the expression of the target gene simply by removing RU-486 from the food or returning the flies to the restrictive temperature when using the GAL80ts system. This is because RNAi activation in flies seems irreversible once initiated (please see reference in the original rebuttal and the discussion in the new version of the manuscript). The only viable solution at present would involve the use of rescue constructs resistant to the RNAi targeting the gene. We have already discussed (both in the rebuttal and in the revised manuscript) that while expressing an orthologue from another species (e.g., the human orthologue) is not optimal in this context;

it may be technically feasible to use a construct carrying silent mutations. We are actively working on the latest approach.

Finally, we respectfully disagree with the suggestion to change "determines" to "contributes" in the title. Even if it were possible to restore the lifespan of the flies by reinstating CI levels, in real-world biological contexts where CI mutations exist but genetic engineering does not (at least until recently and only with human intervention), having a dysfunctional CI during development leads to a shorter lifespan. Therefore, from a biological perspective, we demonstrate that low CI activity during development determines adult lifespan, whereas reduced CI activity during adulthood does not (in flies). This conclusion is supported by evidence from other animal models and patient data, where depletions of CI during development consistently shorten adult survival. Furthermore, although anecdotal, as it is based on a single study, it has been shown that "knocking out" CI in the adult liver of a mouse model (recently published in eLife¹) is completely asymptomatic.

2. Several times the authors invoke a "maladaptive" response - or a "maladaptation" that results from losing CI function in the larval stages. This reviewer is not certain that it is warranted to invoke maladaptation.

A maladaptive response would be occurring if a system were challenged with a perturbation (larval CI loss in this case), and then the system subsequently attempted to suppress the effects of that perturbation with a homeostatic response (unknown). But because of the homeostatic response, there could be a new, downstream physiological problem that is caused, and the long-term result for the animal would be maladaptive (short lifespan, observed) instead of adaptive (normal lifespan despite early CI loss, not observed).

To be clear, all of that is possible. But the difficulty with concluding maladaptation is the absence of a definable homeostatic response. As things stand, the data could mean that there is an early critical period for CI function, nothing more. The downstream problems might simply be a result of that early deficit.

This might seem like a semantic point, but in terms of how the data are ultimately interpreted (lifespan, transcriptomic changes, metabolomic changes, etc.), it is important because what is happening is not necessarily maladaptive. The fact that one sees differences in the transcriptomes or the metabolomes does not constitute evidence of a maladaptation.

Response 3.2: We have addressed this issue above (Response 1. 3).

3. Controls Missing from Main Figure Graphs: The lifespan graphs compare D+A or A-only knockdowns of CI components. In terms of knowing whether the A-only conditions approximate a normal lifespan - or if they themselves have a defect - it would be helpful to include side-by-side control conditions on the main graphs (EtOH). Right now, those control data are housed in the Supplemental information. This applies to a lot of panels: Figs. 1B, C, H, and Figs. 2A, B, C.

Response 3.3: As suggested by the editor, we have included the requested information in an Appendix.

4. Relatedly, the abstract states that mitochondrial dysfunction in adulthood results in flies that are "long-lived" and "stress resistant." An interpretation of that sentence is that the A-only knockdowns are healthier than wild-type/control flies. Is that true, or are they just

statistically indistinguishable from controls? It is tricky to compare back and forth between the main figures and the supplemental.

Response 3.4: We addressed this in our original rebuttal and clarified it in the resubmitted manuscript. The A-only group with CI depletion lives longer than any of the controls. However, the terms "short-lived" (D+A) and "long-lived" (A) are used in relative terms when comparing groups with depleted CI, rather than in comparison to the controls. It is crucial to note that flies with the same background and identical levels of CI during adulthood are short- or long-lived, depending on whether CI was depleted during development.

5. The omics-level analyses are interesting, but somewhat unsatisfying. Figures 3 and 4 are good in terms of being hypothesis-generating mechanisms, but there is no accompanying functional analysis. Gene Ontology (or similar) is a starting point for organising omics data, but it does not substitute for functional analysis, which is a strength of the *Drosophila* system.

Response 3.5: We agree with the reviewer's observations regarding the omics-level experiments as a foundation for further experiments. This is precisely why we released a pre-print and are keen to publish this research promptly, aware that not everyone reads pre-prints (unfortunately). We hope these new models will enable more biologically meaningful experiments to understand better the roles of Complex I and mitochondria in development and ageing.

6. Based on the early data (Figure 1), knockdown of ND-75 is a less severe manipulation than knockdown of ND-18. But based on the transcriptomic and metabolomic data (Figure 3), it seems that ND-75 KD yields far more changes. Some discussion about the differences between ND-18 and ND-75 knockdowns is warranted, even though they show some concordance in the types of factors that are upregulated or downregulated.

Response 3.6: We have included a paragraph discussing the differences between both models.

7. Several of the Figure panels have unusual fonts or text that appears stretched or squished from resizing pictures of graphs.

Response 3.7: We have corrected the formatting in the affected panels.

2. Abstract: "discrete" is the term intended.

Response 3.8: We appreciate the reviewer pointing out this typo and have corrected it.

3. p. 3 Introduction: "...a mountain of evidence has accumulated, suggesting that the severity of pathology induced by mitochondrial/CI dysfunction depends on a factor unrelated to the dysfunction itself."

As written, this conclusion does not seem to make sense, given that the primary defect is to CI gene function. I understand that the authors are trying to draw a distinction between an initial insult to mitochondria vs. the ability of the organism to deal with that. But it does not make the "factor" unrelated to the insult.

Response 3.9: We have revised the phrase to clarify our findings better: "Accordingly, during the course of our research, we have gathered substantial evidence indicating that the severity of pathology induced by mitochondrial or CI dysfunction depends on factors beyond mere

energy generation or the overproduction of reactive oxygen species." This adjustment highlights that the commonly accepted phenotypes associated with mitochondrial dysfunction are not solely a consequence of an energy crisis.

4. "...more adaptive yet also more sensitive" - this phrasing does not make sense. More sensitive would seem to imply less adaptive.

Response 3.10: We have rephrased it: "During development, organisms are more adaptable and, therefore, more vulnerable to changes in their internal and external environments. Consequently, any maladaptations that occur can persist into adulthood".

5. p. 11: "Our data, reconcile previously contradictory findings, and indicate that fully functional CI is necessary during development but not adulthood." For the reasons described in Main Point 1 above, the current data do not rule out a function during adulthood. Moreover, the text needs to be more specific about what is being reconciled (as opposed to referring briefly to "conflicting published reports"). If it is a lengthy explanation, then the discussion would be a better place than the Introduction.

Response 3.11: We have rephrased to: "Our data reconcile the previously contradictory findings that reported both lifespan shortening and extension when CI is depleted. We demonstrate that a fully functional CI is crucial during development but not necessarily in adulthood. This provides a more specific context to the differing outcomes noted in prior studies."

6. p. 12: "These data suggest that...loss in CI levels...may not contribute in itself to ageing or the onset of age-related diseases." The adult *Drosophila* data for Complex I cannot be extrapolated to these other situations. This kind of extrapolation occurs in the discussion too (e.g., p. 20 - "if our discoveries could be translated to humans..."). Without evidence of a critical period in mammals, the extrapolation is not warranted.

Response 3.12: We have amended the text to clarify that while our findings apply to *Drosophila*, they necessitate further validation in other organisms.

7. pp. 16-17, Fig. 2, and Fig. S2: The idea of an "underdeveloped fat body" is not clear from the images shown.

Response 3.13: We have replaced "underdeveloped" with "atrophied."

8. Fig. 1: The differences between the Gene Switch RU486-based system for knockdown and the TARGET GAL80TS system are curious. It is difficult to tell which one is stronger. The Gene Switch system seems to be a more severe effect on CI-linked respiration, yet the overall effects for the organism are mild enough that one gets eclosed adults to evaluate. By contrast, the TARGET system only subtly knocks down CI-linked respiration but precludes eclosion of adults.

Response 3.14: We do not have CI-linked respiration data in our manuscript using the TARGET GAL80ts system, so such a comparison cannot be made.

Referee #3:

(1) The authors state in response 3.3, about including the control data in the main figures, that they have not extensively backcrossed the control and RNAi flies. In the results the controls are described as 'UAS-Empty/Control' but there are no 'UAS-empty' or control RNAi lines listed

in table S1 or mentioned in the figure legends. Moreover, below Table S1 is says 'Lines were backcrossed into w1118 background for 11 generations'. If this included a UAS-empty or control RNAi line, then this would contradict the statement in response 3.3. Can the authors please include the details of the control line used and state whether this line was backcrossed into the same background as the ND-75 and ND-18 RNAi lines. They should also state which control line was used in the figure legends.

Response 4.1: We have clarified the issue with the control and the backcrossing in response 1.1.

(2) Please include catalogue numbers and supplier details for all the antibodies used. This is important so others can reproduce the experiments using the same reagents.

Response 4.2: We have addressed the issue with the control and backcrossing in Response 1.1.

(3) Since the original submission of this manuscript a paper has been published (PMID 38304969) that models CI deficiency in *Drosophila* using two different RNAi lines against ND-75, one of which is the same as the line used by the authors (100733/KK). The authors should mention this paper and that the strength of CI inhibition can be controlled by using RNAi lines with different efficiencies. This is important for researchers wishing to use RNAi to model CI deficiency in *Drosophila*.

Response 4.3: We have included the reference.

References

- 1 Lesner, N. P. *et al.* Differential requirements for mitochondrial electron transport chain components in the adult murine liver. *eLife* **11** (2022). <https://doi.org:10.7554/eLife.80919>

Dr. Alberto Sanz
University of Glasgow
Campus Avenue
United Kingdom

Dear Alberto,

Thank you for submitting your revised manuscript. I have now looked at everything and all is fine. Therefore, I am very pleased to accept your manuscript for publication in EMBO Reports.

Congratulations on a nice work!

Kind regards,

Deniz
--
Deniz Senyilmaz Tiebe, PhD
Senior Scientific Editor
EMBO Reports

--

Rev_Com_number: RC-2023-02211
New_manu_number: EMBOR-2024-60577V2
Corr_author: Sanz
Title: Developmental mitochondrial Complex I activity determines lifespan